# Developmental change in prefrontal cortex recruitment supports the emergence of value-guided memory

**Kate Nussenbaum, Catherine A Hartley***

New York University, New York City, United States

**Abstract** Prioritizing memory for valuable information can promote adaptive behavior across the lifespan, but it is unclear how the neurocognitive mechanisms that enable the selective acquisition of useful knowledge develop. Here, using a novel task coupled with functional magnetic resonance imaging, we examined how children, adolescents, and adults (N = 90) learn from experience what information is likely to be rewarding, and modulate encoding and retrieval processes accordingly. We found that the ability to use learned value signals to selectively enhance memory for useful information strengthened throughout childhood and into adolescence. Encoding and retrieval of high- vs. low-value information was associated with increased activation in striatal and prefrontal regions implicated in value processing and cognitive control. Age-related increases in value-based lateral prefrontal cortex modulation mediated the relation between age and memory selectivity. Our findings demonstrate that developmental increases in the strategic engagement of the prefrontal cortex support the emergence of adaptive memory.

## Introduction

Memories of past experiences guide our behavior, promoting adaptive action selection throughout our lives (*Biderman et al., 2020*). But not all experiences are equally useful to remember — the information we encounter varies in its utility in helping us gain future reward. By adulthood, individuals demonstrate the ability to prioritize memory for information that is likely to be most rewarding in the future (*Adcock et al., 2006*; *Cohen et al., 2019b*; *Cohen et al., 2014*; *Hennessee et al., 2019*; *Shigemune et al., 2014*; *Shohamy and Adcock, 2010*; *Wittmann et al., 2005*). Children, however, demonstrate weaker memory selectivity, often remembering relatively inconsequential information at the expense of higher value items or associations (*Castel et al., 2011*; *Hanten et al., 2007*; *Nussenbaum et al., 2020*). Behavioral studies have found that the use of value to guide encoding and retrieval processes emerges and strengthens gradually throughout childhood and adolescence, promoting more efficient acquisition of useful knowledge with increasing age (*Castel et al., 2011*; *Hanten et al., 2007*; *Nussenbaum et al., 2020*). It is unclear, however, how changes in brain activity support this observed emergence of motivated memory. Although a large literature has examined developmental change in the neural mechanisms that support memory from early childhood to young adulthood (*Ghetti and Fandakova, 2020*; *Ofen, 2012*; *Shing et al., 2010*), no prior studies have investigated how the developing brain prioritizes memories based on their relative utility.

Prioritizing valuable information in memory requires both determining the value of information and strategically modulating encoding accordingly. The vast majority of adult studies have focused only on the strategic *use* of value to guide memory — in most studies of motivated memory, *determining* the value of information is trivial for participants because experimenters label to-be-remembered information with explicit value cues (e.g. dollar signs, stars, point amounts) (*Adcock et al., 2006*; *Castel et al., 2011*; *Cohen et al., 2014*; *Murty et al., 2017*). However, in real-world contexts, individuals must derive the value of information from the statistics of their environments. In a recent

*For correspondence:
cate@nyu.edu

Competing interest: See
page 20

Reviewing editor: Thorsten
Kahnt, Northwestern University,
United States

behavioral study (*Nussenbaum et al., 2020*), we demonstrated that young adults could use naturalistic value signals to prioritize memory for useful information. Specifically, we manipulated information value via item *frequency.* Across many environments, the frequency of encountering something in the past predicts the frequency of encountering it in the future. In this way, frequency can signal information utility (*Anderson and Milson, 1989*; *Anderson and Schooler, 1991*; *Liu et al., 2021*; *Pachur et al., 2014*; *Rich and Gureckis, 2018*; *Stevens et al., 2016*) — if individuals are likely to encounter something often in the future, encoding information about it is likely to be valuable. For example, remembering the supermarket aisle of an ingredient with which one often cooks is likely to facilitate greater reward gain than remembering the aisle of an ingredient one almost never uses. In our prior study, we translated this feature of real-world environments to a laboratory task in which individuals could learn the potential reward value of associative information by first learning the relative frequency of items in their environments. We found that individuals exploited these naturalistic value signals and demonstrated better memory for information associated with high- relative to low-frequency items. Critically, this pattern of results varied with age; the strategic prioritization of high-value information in memory increased from age 7 to age 25 (*Nussenbaum et al., 2020*). It is unclear, however, if developmental improvements in memory prioritization stemmed from differences in learning the relative value of information based on environmental statistics or *using* learned value signals to strategically prioritize memory. Each of these processes likely engages separable neural systems.

Deriving value from the structure of the environment first requires the learning of statistical regularities. In the case of learning the frequency with which one might need to use information, individuals must differentiate novel occurrences (e.g. cooking with a rare ingredient) from oft-repeated experiences (e.g. cooking with a common food). Neurally, medial temporal lobe regions may support sensitivity to item repetitions. The parahippocampal cortex in particular demonstrates reduced responsivity to repeated relative to novel presentations of items (i.e. 'repetition suppression') (*Gonsalves et al., 2005*; *Kirchhoff et al., 2000*; *Köhler et al., 2005*; *O'Kane et al., 2005*; *Turk-Browne et al., 2006*). Although some accounts of repetition suppression suggest that attenuated responses simply indicate neural 'fatigue,' the phenomenon has also been shown to be sensitive to the statistical context of the environment, suggesting that suppression may reflect stimulus expectation and index learning of environmental regularities (*Auksztulewicz and Friston, 2016*). Repetition suppression has also been shown to relate to implicit memory for repeated items (*Ward et al., 2013*). Paralleling their robust implicit learning abilities (*Amso and Davidow, 2012*; *Finn et al., 2016*; *Meulemans et al., 1998*), children and adolescents also demonstrate neural repetition suppression effects (*Nordt et al., 2016*; *Scherf et al., 2011*; *Turi et al., 2015*), although repetition suppression — and the ability to learn the statistical structure of the environment — may increase throughout childhood (*Scherf et al., 2011*). When individuals need to remember information associated with previously encountered stimuli (e.g. the grocery store aisle where an ingredient is located), frequency knowledge may be instantiated as value signals, engaging regions along the mesolimbic dopamine pathway that have been implicated in reward anticipation and the encoding of stimulus and action values. These areas include the ventral tegmental area (VTA) and the ventral and dorsal striatum (*Adcock et al., 2006*; *Liljeholm and O'Doherty, 2012*; *Shigemune et al., 2014*).

*Using* these learned value signals to guide memory likely requires cognitive control (*Castel et al., 2007*; *Cohen et al., 2014*). Value responses in the striatum may signal the need for increased engagement of the dorsolateral prefrontal cortex (dlPFC) (*Botvinick and Braver, 2015*), which supports the implementation of strategic control. Enhanced recruitment of control processes promotes the use of deeper and more elaborative encoding strategies (*Cohen et al., 2019b*; *Cohen et al., 2014*; *Miotto et al., 2006*; *Uncapher and Wagner, 2009*) as well as the selection and maintenance of effective retrieval and post-retrieval monitoring strategies (*Libby and Lipe, 1992*; *Scimeca and Badre, 2012*), which may contribute to better memory for high-value information. The use of value to proactively upregulate cognitive control responses improves throughout development, though the specific trajectory of improvement may relate to the control demands of a given task (*Davidow et al., 2018*). Selectively enhancing the use of encoding and retrieval strategies requires not only tight coordination between subcortical regions involved in value processing and prefrontal areas implicated in control (*Murty and Adcock, 2014*), but also an available repertoire of memory strategies to implement. Even in the absence of value cues, children and adolescents demonstrate reduced use of strategic control (*Bjorkland et al., 2009*) and reduced lateral prefrontal engagement

during encoding (*Ghetti et al., 2010*; *Ghetti and Fandakova, 2020*; *Shing et al., 2016*; *Tang et al., 2018*), suggesting that the availability of mnemonic control strategies may increase with age.

Taken together, prior work suggests that adaptive memory requires the recruitment and coordination of multiple neural systems, including mechanisms for learning environmental structure, representing value, and engaging strategic control, all of which may undergo marked changes from childhood to adulthood. Here, we examined how the development of these neurocognitive processes supports the emergence and strengthening of value-guided memory from childhood to young adulthood. To pinpoint loci of developmental differences in adaptive memory prioritization, we combined our novel motivated memory experiment (*Nussenbaum et al., 2020*) with functional neuroimaging. During the task, participants first learned the frequency of items in their environments, and then learned information associated with each item. Importantly, we structured our task such that the frequency with which participants first experienced each item indicated the frequency with which they would be asked to report the information associated with it, and therefore, the number of points they could earn by remembering the association. Immediately following encoding, we administered a memory test in which participants had to select each item's correct associate. Because frequency of exposure to an item may facilitate subsequent associative memory even when it does *not* signal the value of information (*Popov and Reder, 2020*; *Reder et al., 2016*), in our prior behavioral study (*Nussenbaum et al., 2020*), we examined the effects of item frequency on subsequent associative memory in two contexts: one in which item frequency signaled information value and one in which it did not. Critically, we found that with our experimental design, frequency only facilitated memory when it signaled the value of remembering information — increased item exposure did not in and of itself enhance subsequent associative memory. Thus, in the present fMRI study, we focused only on the condition in which item frequency *did* indicate the potential reward that could be earned for remembering associations.

We examined neural activation during the learning of item frequency, and when participants were asked to encode and retrieve information associated with high- vs. low-frequency items. We hypothesized that while participants across our entire age range would demonstrate sensitivity to the frequency of items in their environments, with increasing age, participants would show improvements in transforming this experiential learning into value signals and modulating the engagement of strategic control processes during encoding. Neurally, we expected that at encoding and retrieval, the recognition of information value would be reflected in increased striatal activation in response to associations involving high- vs. low-frequency items, while the engagement of strategic control would be reflected in increased activation in lateral prefrontal cortex. Further, we hypothesized that increased recruitment of the striatum and prefrontal cortex during encoding and retrieval of high- vs. low-value information would underpin the strengthening of adaptive memory prioritization from childhood to early adulthood.

## Results

### Approach

Participants ages 8–25 years (N = 90; 30 children ages 8–12 years; 30 adolescents ages 13–17 years; 30 adults ages 18–25 years) completed two blocks of three tasks (*Figure 1*) while undergoing functional magnetic resonance imaging. In the first, *frequency-learning* task, participants viewed a continuous stream of 24 unique postcards, one at a time. Twelve of the postcards only appeared once, while 12 repeated five times. Participants indicated whether each postcard they viewed was old or new. In the second, *associative encoding* task, participants viewed the type of stamp that went on each type of postcard. Participants were instructed that in the subsequent task, they would have to stamp *all* of their postcards, earning one point for each postcard stamped correctly. Critically, in the associative encoding task, regardless of the number of each type of postcard that they had (i.e. 1 or 5), participants saw each type of postcard with its corresponding stamp only *once*. Thus, participants were informed that the prior frequency of each postcard indicated the value of encoding its associated stamp, but they had equal exposure to the to-be-encoded associations across frequency conditions. In the *retrieval* task, participants had to indicate the stamp that went with each unique postcard from one of four options, earning one point for each postcard stamped correctly. After stamping each unique postcard once, participants were asked to report its original frequency on a

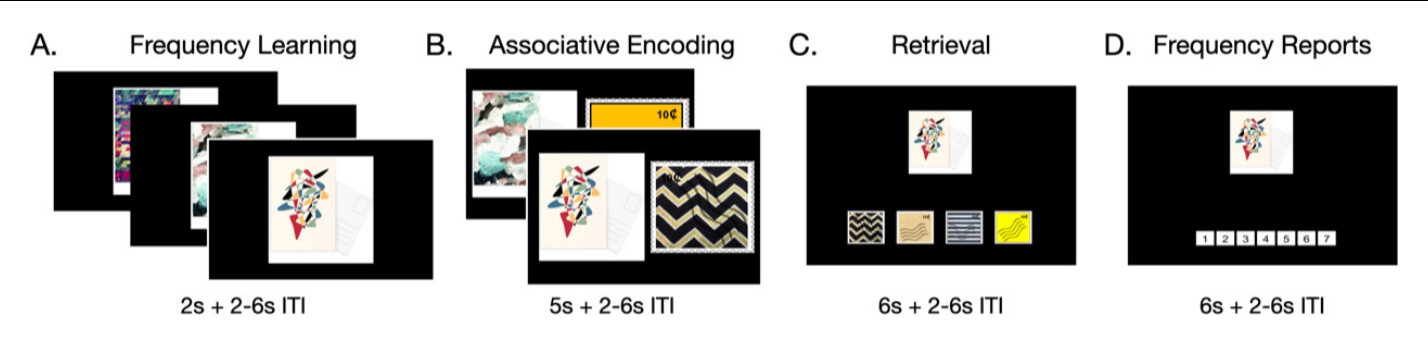

**Figure 1.** Task structure. Participants first learned the frequencies of each item (**A**) by viewing them in a continuous stream. They then were shown the information associated with each item (**B**). During retrieval, participants had to report the information associated with each item (**C**) as well as the item's original frequency (**D**).

scale from 1 to 7. Finally, participants stamped all remaining postcards, such that they completed 48 additional memory test trials (i.e. they stamped each of the postcards in the 5-frequency condition four more times.) These trials were not included in any analyses, but their inclusion ensured that correctly encoding the stamps that belonged on the high-frequency postcards would be more valuable for participants despite each retrieval trial being worth one point. After completing the set of tasks, participants were told that they were going to play a second set of similar games. The second set of tasks was identical to the first, except that the stimuli were changed from postcards and stamps to landscape pictures and picture frames. The order of the stimulus sets was counterbalanced across participants, and data were combined across blocks for analyses.

Across our behavioral analyses, we treated age as a continuous variable. To test for nonlinear effects of age, we first compared the fit of models with a linear age term and with both a linear and quadratic age term (*Braams et al., 2015*; *Somerville et al., 2013*). We dropped the quadratic age term when it did not significantly improve model fit. Because this study was cross-sectional, one concern was that the children, adolescents, and adults that we recruited may have come from different populations. Indeed, we observed a significant relation between age and age-normed Wechsler Abbreviated Scale of Intelligence (WASI; *Wechsler, 2011*) scores in our sample ($\beta = -0.60$, SE = 0.26, p = 0.0238), suggesting the children had slightly higher estimated IQs for their age relative to adults. To account for these age-related differences in reasoning ability, we included age-normed WASI scores as an interacting fixed effect in all analyses. Our aim in including WASI scores as a control variable was to partially account for confounding, population-level differences across our age groups, enabling us to more clearly examine the relation between age itself and our neurocognitive processes of interest.

## Experiential learning of environmental statistics improved with age

During frequency learning, participants across our age range responded to new and repeated items with a high degree of accuracy (new items: *mean = 0.90, SD = 0.30*; repeated items: *mean = 0.92, SD = 0.27*; *Appendix 2—figure 1A*). Older participants demonstrated higher accuracy in correctly identifying both new and repeated items (generalized mixed-effects model results: *new items: $\chi^2(1)$ = 25.52, p < 0.001, repeated items: $\chi^2(1)$ = 33.43, p < 0.001 Appendix 3—table 1* and *2*). Participants were also more accurate in identifying items as 'repeated' as the number of times they saw each item increased, $\chi^2 = 138.03$, p < 0.001, indicating learning throughout the task. This effect varied as a function of age — younger participants demonstrated a larger effect of the number of item repetitions on response accuracy, as indicated by a significant interaction, $\chi^2(1) = 17.41$, p < 0.001.

Response times to both new and old items decreased with age (*Appendix 2—figure 1B*, *Appendix 3—table 3* and *4*; linear mixed-effects model results: *new items: F(1, 85.99) = 32.51, p < 0.001; old items: F(1, 87.55) = 21.82, p < 0.001*), such that reaction times decreased steeply throughout childhood before leveling off into late adolescence and early adulthood. Finally, response times for old items also decreased as the number of item repetitions increased, $F(1, 69.94) = 282.21$, p < 0.001.

Participants' ability to distinguish old from new items was associated with a wide network of neural regions, some of which demonstrated greater activation in response to the last vs. first appearance of each item, and others of which demonstrated suppressed activation across repetitions. Specifically, whole-brain contrasts revealed greater recruitment of regions of the lateral occipital cortex, the frontal pole, precuneus, angular gyrus, and caudate (among other regions, see *Figure 2A* and *Appendix 4—table 1*), on the last vs. first appearance of each item. We observed widespread repetition suppression effects, reflected in *decreases* in neural responsivity in the lateral occipital cortex and temporal occipital cortex on the last vs. first appearance of each item. In line with our hypothesis, we also observed a robust decrease in activation in the parahippocampal cortex (*Figure 2B*, *Appendix 4—table 2*).

We next examined whether repetition suppression in the parahippocampal cortex changed with age. We defined a parahippocampal region of interest (ROI) by drawing a 5 mm sphere around the peak voxel from the group-level first > last appearance contrast ($x = 30$, $y = -39$, $z = -15$), and mirrored it to encompass both right and left parahippocampal cortex (*Figure 2C*). For each participant, we modeled the neural response to each appearance of each high-frequency item. We then examined how neural activation changed as a function of repetition number and age. To account for non-linear effects of repetition number, we included linear and quadratic repetition number terms. In line with our whole-brain analysis, we observed a main effect of repetition number, $F(1, 5015.9) = 30.64$, p<0.001, indicating that neural activation within the parahippocampal ROI decreased across repetitions (*Appendix 3—table 5*). Further, we observed a main effect of quadratic repetition number, $F(1, 9881.0) = 7.47$, p = 0.006, indicating that the reduction in neural activity was greatest across earlier repetitions (*Figure 3A*). Importantly, the influence of repetition number on neural activation varied with both linear age, $F(1, 7267.5) = 7.2$, p = 0.007, and quadratic age, $F(1, 7260.8) = 6.9$, p = 0.009. Finally, we also observed interactions between quadratic repetition number and both linear and quadratic age (ps < 0.026). These age-related differences suggest that repetition suppression was greatest in adulthood, with the steepest increases occurring from late adolescence to early adulthood (*Figure 3*).

For each participant for each item, we also computed a 'repetition suppression index' by taking the difference in mean beta values within our ROI on each item's first and last appearance (*Ward et al., 2013*). These indices demonstrated a similar pattern of age-related variance — we found that the reduction of neural activity from the first to last appearance of the items varied positively with linear age, $F(1, 78.32) = 3.97$, p = 0.05, and negatively with quadratic age, $F(1, 77.55) = 4.8$, p = 0.031 (*Figure 3B*, *Appendix 3—table 6*). Taken together, our behavioral and neural results suggest that sensitivity to the repetition of items in the environment was prevalent from childhood to adulthood but increased with age.

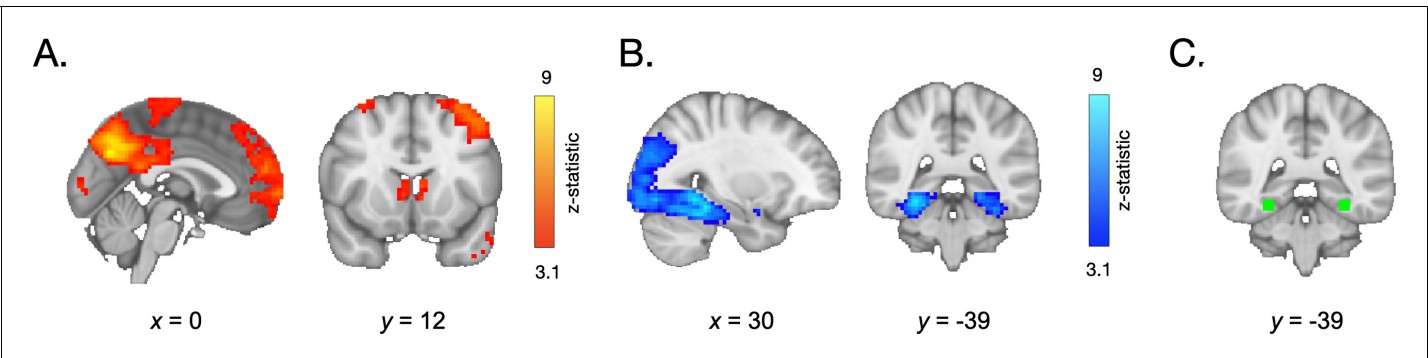

**Figure 2.** Neural activation during frequency learning. (A) During frequency learning, participants demonstrated increased recruitment of regions in the frontal cortex, angular gyrus, and striatum on the last vs. first appearance of high-frequency items. (B) They demonstrated *decreased* activation in the lateral occipital cortex, temporal occipital cortex, and parahippocampal cortex. (C) Within a parahippocampal ROI (shown in green), the decrease in responses to each stimulus on its last vs. first appearance was greater in older participants.

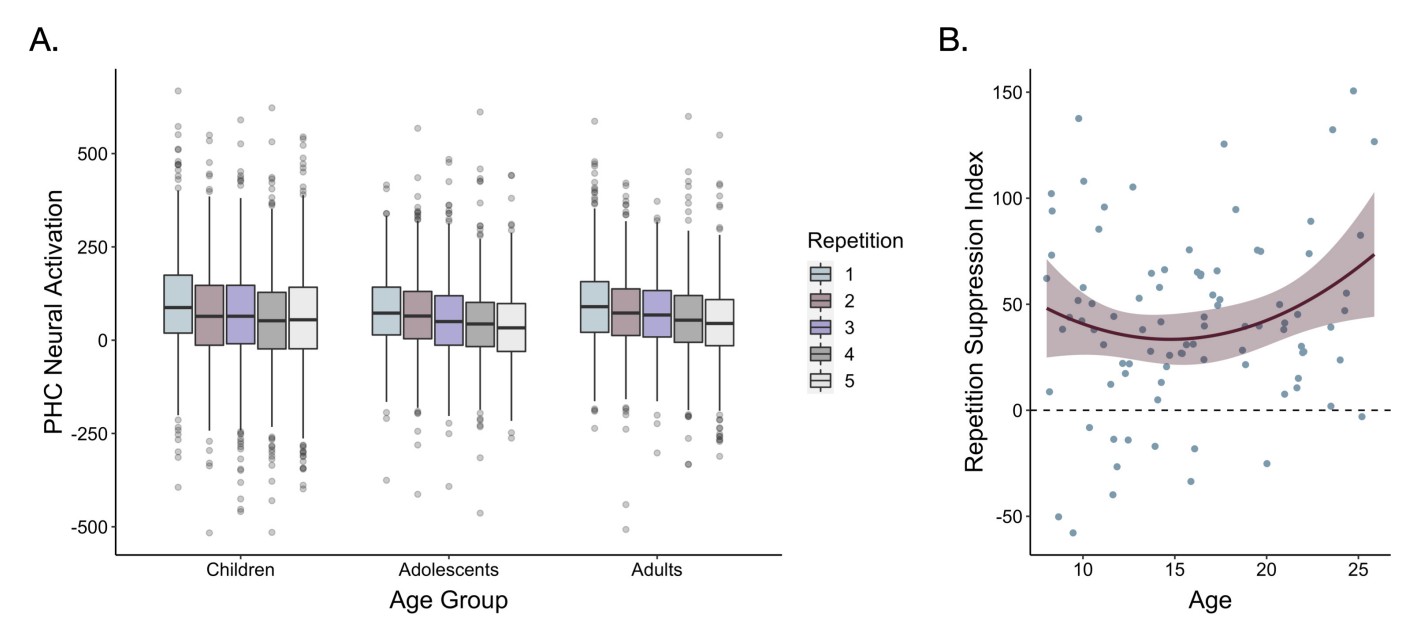

**Figure 3.** Repetition suppression during frequency learning. (**A**) Neural activation within a bilateral parahippocampal cortex ROI decreased across stimulus repetitions both linearly, $F(1, 5015.9) = 30.64$, p < 0.001, and quadratically, $F(1, 9881.0) = 7.47$, p = 0.006. Repetition suppression increased with linear age, $F(1, 7267.5) = 7.2$, p = 0.007, and quadratic age $F(1, 7260.8) = 6.9$, p = 0.009. The horizontal black lines indicate median neural activation values. The lower and upper edges of the boxes indicate the first and third quartiles of the grouped data, and the vertical lines extend to the smallest value no further than 1.5 times the interquartile range. Grey dots indicate data points outside those values. (**B**) The decrease in neural activation in the bilateral PHC ROI from the first to fifth repetition of each item also increased with both linear age, $F(1, 78.32) = 3.97$, p = 0.05, and quadratic age, $F(1, 77.55) = 4.8$, p = 0.031. The line on the scatter plot represents the best-fitting regression line from the model including both linear and quadratic age terms. The shaded region represents 95% confidence intervals.

## Age-related differences in explicit knowledge of environmental structure

Could participants transform their sensitivity to environmental statistics into explicit reports of item frequency? To address this question, we computed participants' frequency report error magnitudes by taking the absolute value of the difference between the item's true frequency (i.e. 1 or 5) and each participant's explicit report of its frequency (i.e. 1–7). We then examined how these report error magnitudes varied as a function of age and frequency condition (*Appendix 3—table 7*). We observed a main effect of age ($F(1, 94.30) = 17.57$, p < 0.001) such that error magnitudes decreased with increasing age (Children: *Mean* = 1.48, *SD* = 1.34; Adolescents: *Mean* = 1.10, *SD* = 1.12; Adults: *Mean* = 1.13, *SD* = 1.05). Error magnitudes were not related to frequency condition (p = 0.993), indicating that participants were not systematically better at representing the 'true' frequencies of items that appeared once or items that appeared five times.

To examine relations between online frequency learning and explicit knowledge, we tested whether repetition suppression indices for each item related to frequency reports (*Appendix 3—table 8*). We hypothesized that participants would report the items that elicited the greatest repetition suppression as most frequent. However, in line with other studies suggesting dissociations between repetition suppression and explicit memory (*Ward et al., 2013*), we did not observe any relation between repetition suppression indices and frequency reports, $F(1, 1360.74) = 0.01$, p = 0.903. Thus, while we observed parallel developmental improvements in online frequency learning and subsequent explicit reports, they may be driven by separable processes.

## Age-related differences in value-guided memory

Participants' frequency-learning performance and their explicit frequency reports indicate that older participants were better both at tracking repetitions of items within their environments and at explicitly representing item frequencies. Were participants able to use these representations of the

structure of their environment to prioritize memory for high-value information? To address this question, we examined how frequency condition and age influenced memory accuracy (*Appendix 3—table 9*). Memory accuracy varied as a function of both linear ($\chi 2(1) = 8.68$, $p = 0.003$) and quadratic age ($\chi 2(1) = 4.24$, $p = 0.039$), such that older participants demonstrated higher memory accuracy, with the steepest improvements in memory accuracy occurring from childhood into early adolescence (*Figure 4*). In line with our hypothesis, we observed a main effect of frequency condition on memory, $\chi 2(1) = 19.73$, $p < 0.001$, indicating that individuals used naturalistic value signals to prioritize memory for high-value information. Critically, this effect interacted with both linear age ($\chi 2(1) = 10.74$, $p = 0.001$) and quadratic age ($\chi 2(1) = 9.27$, $p = 0.002$), such that the influence of frequency condition on memory increased to the greatest extent throughout childhood and early adolescence.

To determine whether the interaction between quadratic age and frequency condition on memory accuracy reflected an adolescent peak in value-guided memory prioritization, we re-ran our memory accuracy model without including any age terms and extracted each participant's random slope across frequency conditions. We then submitted these random slopes to the 'two-lines' test (*Simonsohn, 2018*), which fits two regression lines with oppositely signed slopes to the data, algorithmically determining where the sign flip should occur. The results of this analysis revealed that the influence of frequency condition on memory significantly increased from age 8 to age 15.86 ($b = 0.03$, $z = 2.71$, $p = 0.0068$; *Appendix 2—figure 2*), but only marginally decreased from age 15.86 to

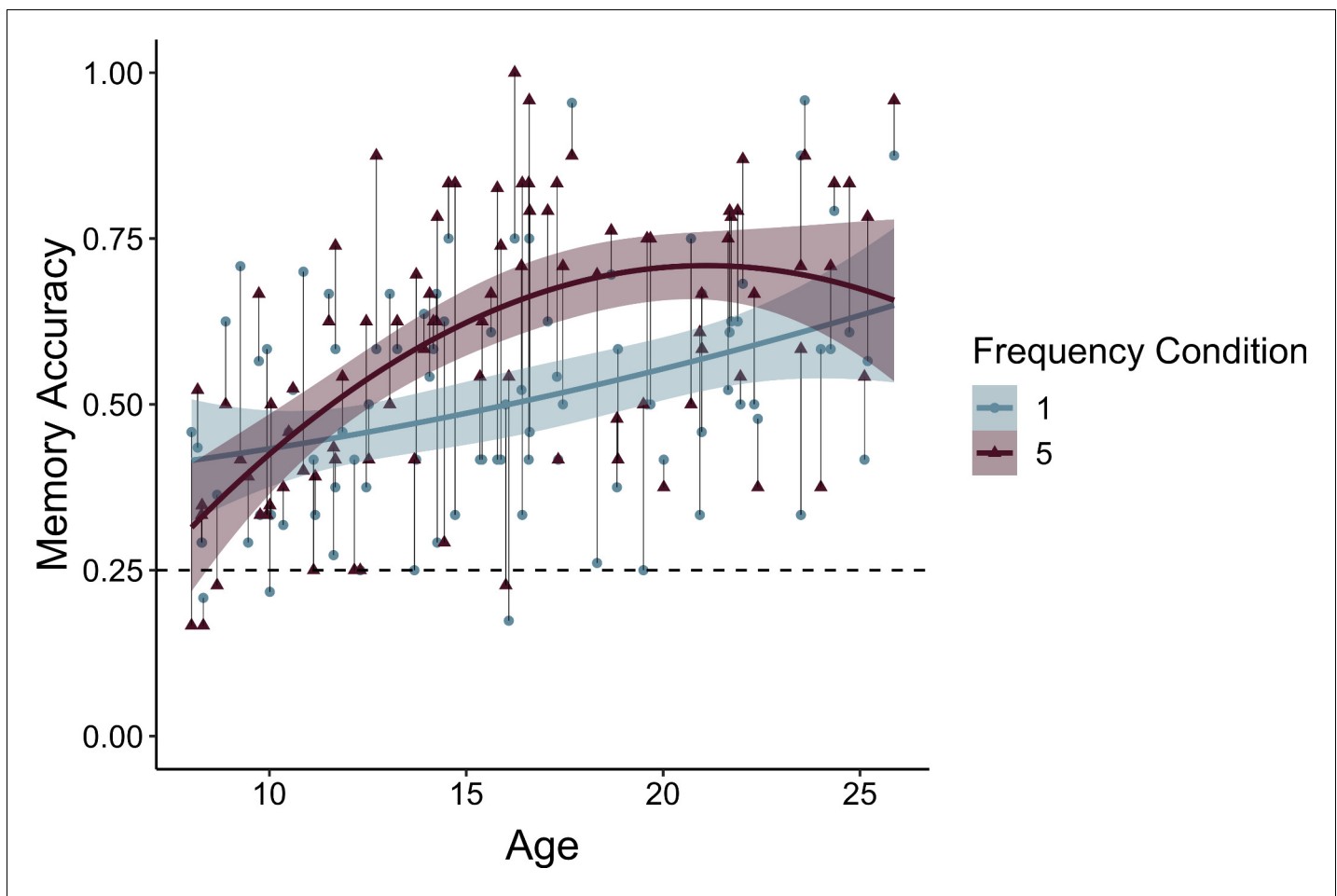

**Figure 4.** Memory accuracy by age and frequency condition. Participants demonstrated prioritization of memory for high-value information, as indicated by higher memory accuracy for associations involving items in the five- relative to the one-frequency condition ($\chi 2(1) = 19.73$, $p < 0.001$). The effects of item frequency on associative memory increased throughout childhood and into adolescence (linear age x frequency condition: $\chi 2(1) = 10.74$, $p = 0.001$; quadratic age x frequency condition: $\chi 2(1) = 9.27$, $p = 0.002$). The thin grey lines connect each dots representing each participant's memory accuracy for items in the one- and five-frequency condition. The thicker colored lines represent the best-fitting regression lines from models including linear and quadratic age terms. The shaded regions represent 95% confidence intervals.

age 25 (b = −0.02, *z* = 1.91, p = 0.0576). Thus, the interaction between frequency condition and quadratic age on memory performance suggests that the biggest age differences in value-guided memory occurred through childhood and early adolescence, with older adolescents and adults performing similarly.

Because we observed age-related differences in participants' online learning of item frequencies and in their explicit frequency reports, we further examined whether these age differences in initial learning could account for the age differences we observed in associative memory. To do so, we ran an additional model in which we included each participant's mean frequency learning accuracy, mean frequency learning accuracy on the last repetition of each item, and explicit frequency report error magnitude as covariates (*Appendix 3—table 11*). Here, explicit frequency report error magnitude predicted overall memory performance, χ2(1) = 13.05, p < 0.001, and we did not observe main effects of age or quadratic age on memory performance (ps > 0.20). However, we continued to observe a main effect of frequency condition, χ2(1) = 19.65 p < 0.001, as well as significant interactions between frequency condition and both linear age χ2(1) = 10.59, p = 0.001, and quadratic age χ2(1) = 9.15, p = 0.002. Thus, while age differences in initial learning related to overall memory performance, they did not account for age differences in the use of environmental regularities to strategically prioritize memory for valuable information.

## Neural mechanisms of value-guided encoding

We next examined how neural activation during encoding supported the use of learned value to guide memory. Specifically, we examined whether participants demonstrated different patterns of neural activation during encoding of information associated with high- vs. low-frequency items. In line with our hypothesis, a whole-brain contrast revealed increased engagement of the left lateral PFC and bilateral caudate (1765 voxels at *x* = −51, *y* = 42, *z* = 9; 232 voxels at *x* = 18, *y* = 12, *z* = 6; and 54 voxels at *x* = 18, *y* = 18, and *z* = 12; *Figure 5A*, *Appendix 4—table 4*) during encoding of the pairs involving high-frequency items relative to pairs involving low-frequency items. To examine how this pattern of activation related to behavior, we computed a 'memory difference score' for each participant by subtracting their memory accuracy for associations involving low-frequency items from their accuracy for associations involving high-frequency items. We then included these memory difference scores as a covariate in our group-level GLM examining neural activation during encoding of pairs involving high- vs. low-frequency items. Participants who demonstrated the greatest difference in memory accuracy for pairs involving high-frequency vs. low-frequency items also demonstrated greater value-based modulation of left lateral PFC activation (232 voxels at *x* = −48, *y* = 21, *z* = 27; *Figure 5B*, *Appendix 4—table 5*).

Because participants demonstrated effects of value on memory, neural signatures of encoding high- vs. low-value information may reflect successful vs. unsuccessful encoding. To de-confound the effects of value vs. subsequent memory accuracy on neural activation at encoding, we re-ran our high- vs. low-value contrast but restricted our analysis to associations that were subsequently retrieved correctly. We observed similar neural effects — increased recruitment of the left lateral prefrontal cortex and left caudate during encoding of high- vs. low- value pairs (1042 voxels at *x* =

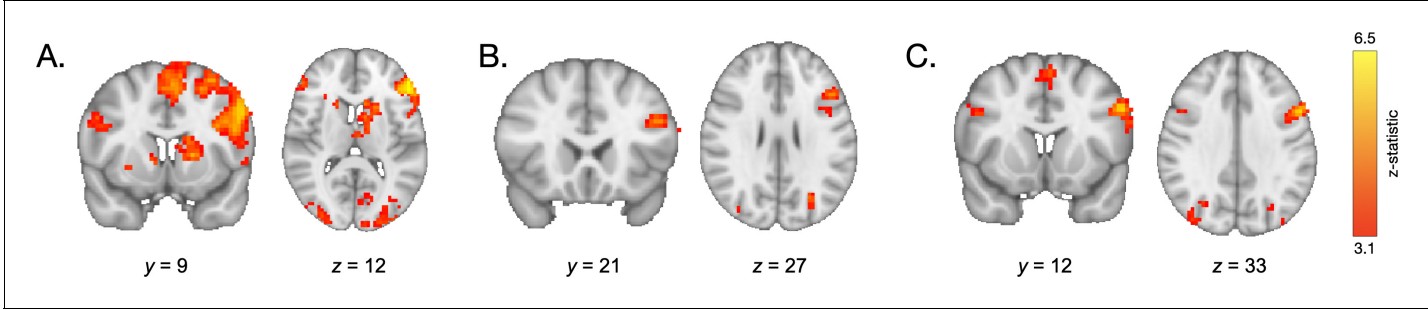

**Figure 5.** Neural activation during encoding. (A) During encoding of associations involving high- vs. low-frequency items, participants demonstrated greater engagement of the lateral PFC and caudate. (B) Participants who demonstrated the greatest value-based modulation of memory also demonstrated the greatest modulation of left prefrontal cortical activation during encoding of high- vs. low-value associations. (C) During encoding of both high- and low-value pairs, older participants demonstrated greater recruitment of the PFC relative to younger participants.

−42, $y$ = 15, $z$ = 30; 124 voxels at $x$ = −12, $y$ = −3, $z$ = 9). Further, neural signatures of successful vs. unsuccessful encoding differed from those of high- vs. low-value encoding. Though we observed similar activation in left lateral PFC (1273 voxels at $x$ = −48, $y$ = 9, $z$ = 27; *Appendix 4—table 6*), here, we did not observe differential recruitment of the caudate. In addition, consistent with previous observations of subsequent memory effects (*Davachi, 2006*), successful encoding was associated with increased activation in the right hippocampus (21 voxels at $x$ = 24, $y$ = −6, $z$ = −21).

## Age-related differences in neural activation during encoding

Next, we examined how neural activation during encoding vs. baseline (fixation) varied with age. Across trial types, during encoding, we observed widespread age-related increases in neural activation (*Appendix 4—table 3*), in regions including the left lateral PFC (120 voxels at $x$ = −54, $y$ = 12, $z$ = 33) and the right lateral PFC (24 voxels at $x$ = 48, $y$ = 12, $z$ = 30; *Figure 5C*).

To address our main question of interest — how age-related differences in differential neural activation during the encoding of high- vs. low-value information may support the development of adaptive memory — we conducted two ROI analyses. Given our *a priori* hypotheses about the role of the prefrontal cortex and striatum in value-guided encoding, and their exhibiting differential activation in the high- vs. low-value encoding group-level contrast, we examined neural activation within a prefrontal cortex and striatal ROI. Despite the absence of significant differential activation in the hippocampus and parahippocampal cortex, we also used the same ROI approach to test for age differences in activation in these *a priori* regions of interest but we did not observe any relations between age and hippocampal activation (see Appendix 2: Supplementary Results for details). The specific prefrontal and striatal ROIs were determined by taking the peak prefrontal voxel ($x$ = −51, $y$ = 42, $z$ = 9) and the peak striatal voxel ($x$ = −18, $y$ = 12, $z$ = 6) from the group-level high- vs. low-value associative encoding contrast and drawing 5 mm spheres around them. We then examined how the mean parameter estimate across voxels within each ROI for the high- vs. low-value encoding contrast related to both linear and quadratic age. Caudate activation did not vary significantly as a function of age ($\beta$ = 0.16, SE = 0.11, $p$ = 0.126) (*Appendix 3—table 12*), indicating that participants across our age range demonstrated similarly increased recruitment of the caudate while encoding high- vs. low-value associations. PFC activation, however, demonstrated a different pattern, varying as a function of both linear ($\beta$ = 1.97, SE = 0.74, $p$ = 0.01) and quadratic age ($\beta$ = −1.73, SE = 0.73, $p$ = 0.021), such that the difference in PFC engagement during encoding of high- vs. low-value associations increased to the greatest extent throughout childhood and early adolescence (*Appendix 2—figure 3*, *Appendix 3—table 13*).

The pattern of age-related differences that we observed in the PFC recruitment mirrored the age-related differences we observed in value-based memory. Given these parallel age effects across brain and behavior, we next asked whether age differences in PFC recruitment could account for our observed age differences in adaptive memory prioritization. First, we confirmed that in line with our whole-brain analysis, PFC modulation predicted memory difference scores, even when controlling for age ($\beta$ = 0.34, SE = 0.10, $p$ = 0.001). Next, we confirmed that these difference scores did in fact vary with age ($\beta$ = 0.22, SE = 0.10, $p$ = 0.041), with older participants demonstrating a larger difference in memory accuracy for high- vs. low- value associations. Critically, however, when controlling for PFC activation, age no longer related to memory difference scores ($\beta$ = 0.15, SE = 0.10, $p$ = 0.14). A formal mediation analysis revealed that PFC activation fully mediated the relation between linear age and memory difference scores (standardized indirect effect: .07, 95% confidence interval: [.01, .15], $p$ = 0.017; standardized direct effect: .15, 95% confidence interval: [−0.03, .33], $p$ = 0.108; *Figure 6*). This relation was directionally specific; age did *not* mediate the relation between PFC activation and memory difference scores (standardized indirect effect: .03, 95% confidence interval: [−0.007, .09], $p$ = 0.13; standardized direct effect: .34, 95% confidence interval: [.14, .54], $p$ < 0.001.) Further, when we included quadratic age, WASI scores, online frequency learning accuracy, online frequency learning accuracy on the final repetition of each item, and mean explicit frequency report error magnitudes as control variables in the mediation analysis, PFC activation continued to mediate the relation between linear age and memory difference scores (standardized indirect effect: .56, 95% confidence interval: [0.06, 1.35], $p$ = 0.023; standardized direct effect: 1.75, 95% confidence interval: [0.12, 3.38], $p$ = 0.034).

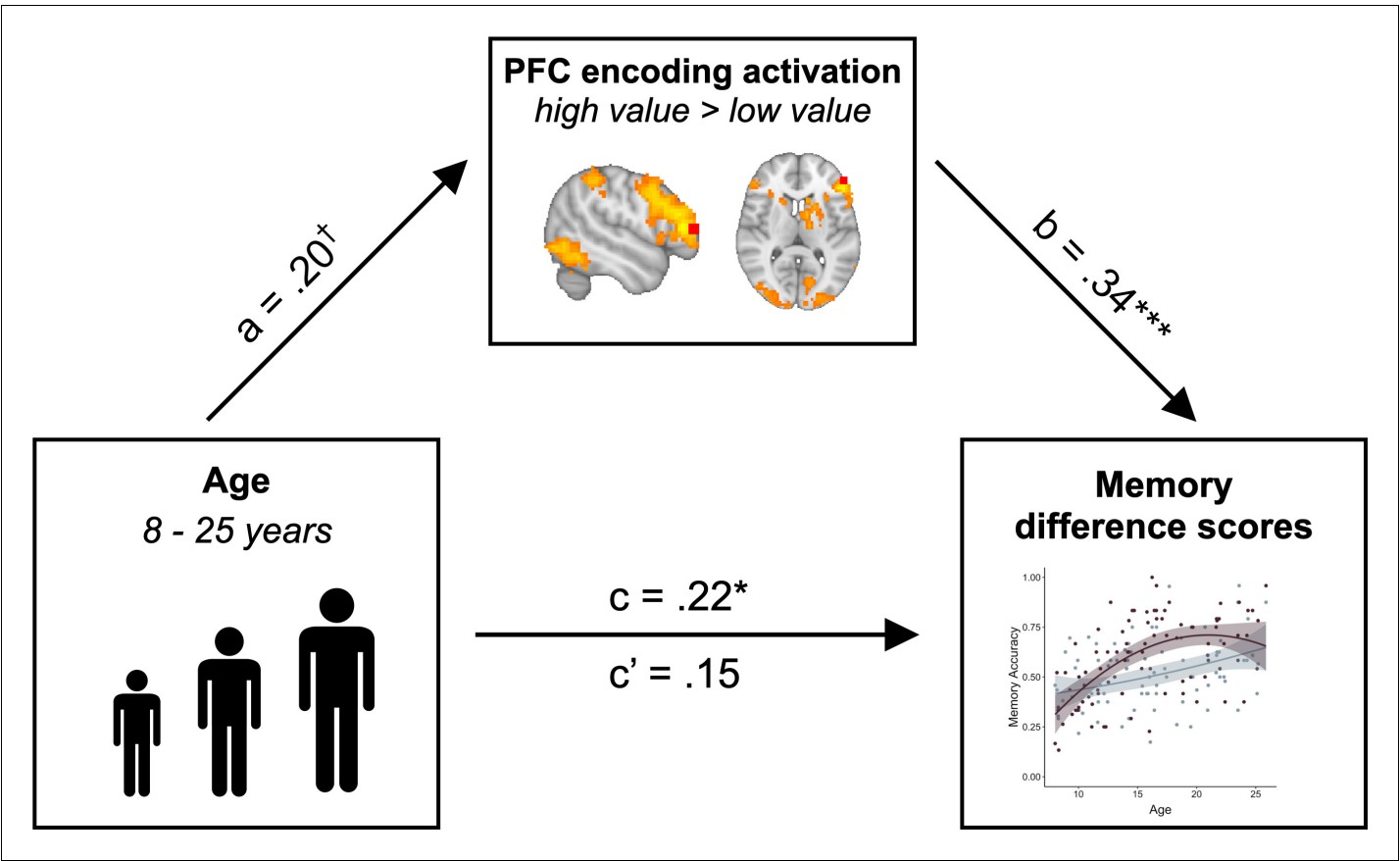

**Figure 6.** PFC activation mediates the relation between age and value-guided memory. The increased engagement of left lateral PFC (ROI depicted in red) during encoding of high- vs. low-value information mediated the relation between age and memory difference scores (standardized indirect effect: .07, 95% confidence interval: [0.01, 0.15], p = 0.017; standardized direct effect: .15, 95% confidence interval: [−0.03, 0.33], p = 0.108). Path a shows the regression coefficient of the relation between age and PFC modulation. Path b shows the regression coefficient of the relation between PFC activation and memory difference scores, while controlling for age. Paths c and c' show the regression coefficient of the relation between age and memory difference scores without and while controlling for PFC activation, respectively. † denotes p<0.06, * denotes p<0.05, ** denotes p<0.01.

### Age- and value-based modulation of neural activation during retrieval

We next examined how the neural mechanisms of memory retrieval related to both age and learned value signals. As during encoding, a whole-brain contrast comparing retrieval trials to baseline revealed age-related differences in bilateral PFC recruitment (*Figure 7A*; 42 voxels at $x = 51$, $y = 3$, $z = 21$; 36 voxels at $x = −60$, $y = 6$, $z = 21$) as well as regions of occipital cortex (see *Appendix 4— table 7*) across trials during retrieval. We further tested whether participants demonstrated value-based modulation of neural activation at retrieval (*Appendix 4—table 8*). During retrieval of associations involving high- vs. low-frequency items, we continued to observe increased engagement of the left lateral PFC (*Figure 7B*; 116 voxels at $x = −48$, $y = 21$, $z = 24$) and the bilateral caudate (128 voxels at $x = −12$, $y = −6$, $z = 15$ and 77 voxels at $x = 15$, $y = −3$, $z = 21$). This activation was not related to age or memory difference scores.

### Two distinct value representations influence memory

The overlap between the engagement of the neural systems we observed during encoding of high- vs. low-value information and those observed in prior studies of motivated memory that have used explicit value cues (*Cohen et al., 2019b*; *Cohen et al., 2014*) suggests that participants did indeed use learned regularities as value signals to guide memory. To what extent was memory supported by explicit representations of item frequency versus neural sensitivity to item repetitions during frequency learning? To examine the influence of these two types of value representations across age, we ran additional mixed-effects models. First, we examined how participants' explicit

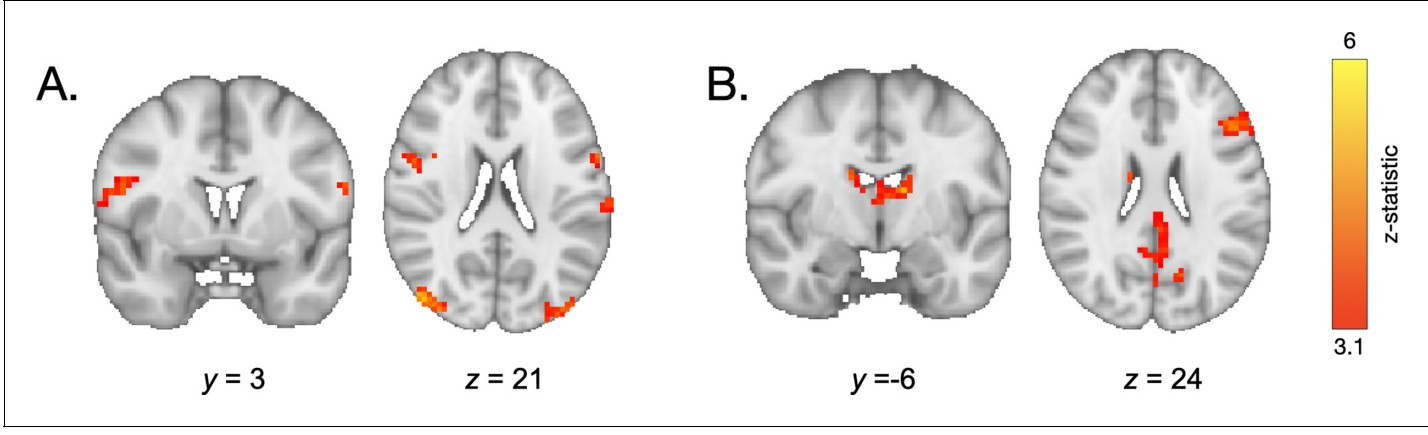

**Figure 7.** Neural activation during retrieval. (A) During retrieval, older participants demonstrated greater recruitment of the inferior frontal cortex relative to younger participants. (B) During retrieval of associations involving high- vs. low-frequency items, participants demonstrated greater engagement of the left lateral PFC and bilateral caudate.

representations of item frequency related to memory (*Appendix 3—table 14*). Participants demonstrated better associative memory for pairs involving items they reported were more frequent, $\chi2(1)$ = 31.20, p < 0.001 (*Figure 8A*). This effect was modulated by age ($\chi2(1)$ = 10.37, p = 0.001) and quadratic age ($\chi2(1)$ = 9.50, p = 0.002), indicating that participants' beliefs about item frequency influenced memory to the greatest degree in adolescence and early adulthood. Further, replicating our previous behavioral findings (*Nussenbaum et al., 2020*), we found that the linear model including explicit frequency reports (BIC = 5438.37) fit the data *better* than the linear model including the true frequency condition (BIC = 5449.19, $\chi2$ = 10.83, p < 0.001), indicating that participants' *representations* of item frequency influenced memory to a greater extent than the true item frequencies.

We further examined whether our neural measure of online frequency learning related to associative memory. Specifically, we asked whether greater sensitivity to item repetitions — as indexed by greater repetition suppression within the parahippocampal cortex — promoted better encoding of associative information (*Appendix 3—table 15*). Because we only had repetition suppression indices for items that appeared five times, our analysis was restricted to associations involving high-frequency items. We found that repetition suppression during frequency learning did indeed predict subsequent associative memory, $\chi2(1)$ = 11.21, p < 0.001 (*Figure 8B*). This effect did not interact with age, $\chi2(1)$ = 0.79, p = 0.374. Further, when we included both repetition suppression indices and explicit frequency reports in our model, *both* predictors continued to explain significant variance in memory accuracy (Frequency reports: $\chi2(1)$ = 21.16, p < 0.001, Repetition suppression: $\chi2(1)$ = 10.25, p = 0.001), suggesting that learned value signals that guide memory may be derived from multiple, distinct representations of prior experience.

Given the relations, we observed between memory and both repetition suppression and frequency reports, we examined whether they related to neural activation in both our caudate and PFC ROI during encoding. To do so, we computed each participant's average repetition suppression index, and their 'frequency distance' — or the average difference in their explicit reports for items in the high- and low-frequency conditions. We expected that participants with greater average repetition suppression indices and greater frequency distances represented the high- and low-frequency items as more distinct from one another and therefore would show greater differences in neural activation at encoding across frequency conditions. In line with our prior analyses, both metrics varied with age (though repetition suppression only marginally (linear age: p = 0.067; quadratic age: p = 0.042); *Appendix 3—tables 17* and *20*), suggesting that older participants demonstrated better learning of the structure of the environment. We ran linear regressions examining the relations between each metric, age, and their interaction on neural activation in both the caudate and PFC. We observed no significant effects or interactions of average repetition suppression indices on neural activation (ps > 0.15; *Appendix 3—tables 18* and *19*). We did, however, observe a significant effect of frequency distance on PFC activation (β = 0.42, SE = 0.12, p = 0.0012), such that

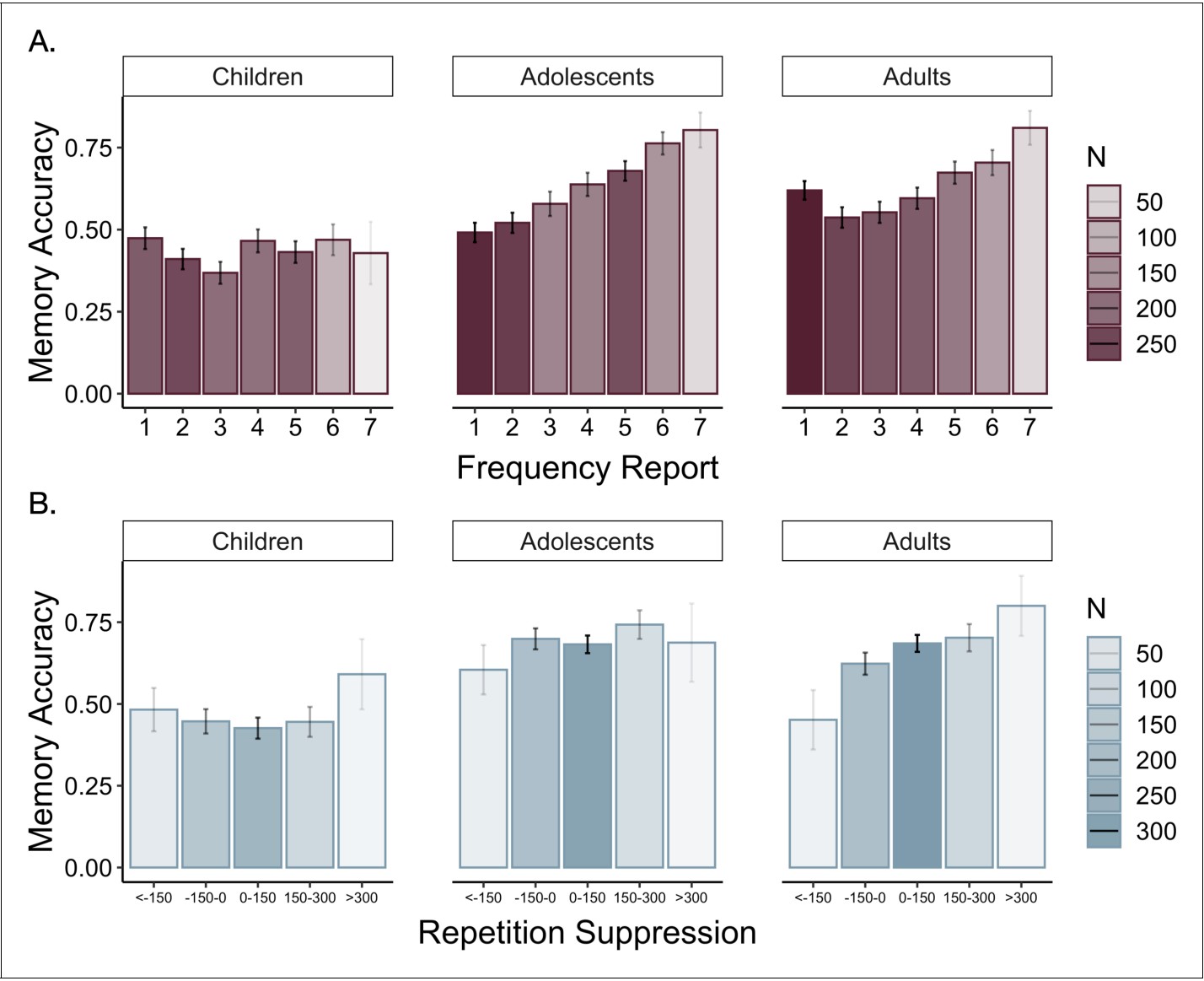

**Figure 8.** Memory accuracy by reported frequency. (**A**) Participants demonstrated increased associative memory accuracy for items that they reported as being more frequent ($\chi2(1) = 31.20$, $p < 0.001$). This effect strengthened with increasing age (frequency report x linear age: $\chi2(1) = 10.37$, $p = 0.001$; frequency report x quadratic age: $\chi2(1) = 9.50$, $p = 0.002$). (**B**) Participants also demonstrated better memory for associations involving high-frequency items to which they demonstrated the greatest repetition suppression during frequency learning ($\chi2(1) = 11.21$, $p < 0.001$). In both panels, the shading of the bars represents the number of trials included in each bin.

participants who believed that average frequencies of the high- and low-frequency items were further apart also demonstrated greater PFC activation during encoding of pairs with high- vs. low-frequency items (*Appendix 3—tables 21* and *22*). Here, we did not observe a significant effect of age on PFC activation ($\beta = -0.03$, SE = 0.13, $p = 0.82$), suggesting that age-related variance in PFC activation may be related to age differences in explicit frequency beliefs. Importantly, however, even when we accounted for both PFC activation and frequency distances, we continued to observe an effect of age on memory difference scores ($\beta = 0.56$, SE = 0.20, $p = 0.006$) (*Appendix 3—table 23*), which, together with our prior analyses (*Appendix 3—table 16*), suggest that developmental differences in value-guided memory are not driven solely by age differences in beliefs about the structure of the environment but also depend on the use of those beliefs to guide encoding.

## Discussion

Prioritizing memory for useful information is essential throughout individuals' lifetimes, but no prior work has investigated the development of the neural mechanisms that support value-guided memory prioritization from childhood to adulthood. The goal of the present study was to characterize how developmental differences in the neurocognitive processes that support both the *learning* and *use* of information value support improvements in adaptive memory from childhood to adulthood. In line with studies of motivated memory in adults (*Cohen et al., 2014*), we found that during encoding and retrieval, high- relative to low-value stimuli elicited increased activation in regions sensitive to value and motivational salience, including the caudate (*Delgado et al., 2004*), and those associated with strategic control processes, including the lateral PFC (*Badre and Wagner, 2007*; *Cole and Schneider, 2007*; *Power and Petersen, 2013*). Further, replicating previous work, we found that value-guided memory selectivity improved across childhood and adolescence (*Castel et al., 2011*; *Hanten et al., 2007*; *Nussenbaum et al., 2020*). Critically, here we demonstrate that increased engagement of the lateral PFC during encoding mediated the relation between age and memory selectivity. This relation was specific to the lateral PFC — although the caudate similarly demonstrated increased activation during encoding of high- vs. low-value information, value-based modulation of caudate activation did not vary as a function of age and did not relate to memory selectivity.

Two different signatures of value-learning predicted subsequent associative memory: Individuals demonstrated better memory for associations involving items that elicited stronger repetition suppression as well as for items that they reported as being more frequent. Moreover, the relation of these learning signals to memory performance varied with age. While all participants demonstrated a similar relation between repetition suppression and subsequent associative memory, the association between explicit frequency reports and memory was greater in older participants. These divergent developmental trajectories suggest that the influence of learned value on memory arises through distinct cognitive processes. One possibility is that while explicit beliefs about information value triggered the engagement of strategic control, stimulus familiarity (as indexed by repetition suppression [*Gonsalves et al., 2005*]) may have facilitated encoding of novel associations, even in the absence of controlled strategy use. Indeed, prior work suggests that stronger memory traces for constituent components enhance associative memory (*Chalmers and Humphreys, 2003*; *Popov and Reder, 2020*; *Reder et al., 2016*). However, in our previous behavioral work (*Nussenbaum et al., 2020*), we found that removing the relation between item frequency and reward value eliminated the memory benefit for associations involving high-frequency items, suggesting that stimulus familiarity itself did not account for the influence of item frequency on memory in our task. Still, when frequency does predict value, stimulus familiarity may serve as a proxy for information utility. This familiarity signal may exert age-invariant effects on subsequent memory, whereas explicit beliefs about item frequency may more strongly facilitate subsequent memory with increasing age.

Importantly, although we observed age-related differences in participants' learning of the structure of their environments, the strengthening of the relation between frequency reports and associative memory with increasing age suggests that age differences in learning cannot fully account for age differences in value-guided memory. Even when accounting for individual differences in participants' explicit knowledge of the structure of the environment, older participants demonstrated a stronger relation between their beliefs about item frequency and associative memory, suggesting that they used their beliefs to guide memory to a greater degree than younger participants. In addition, we continued to observe a robust interaction between age and frequency condition on associative memory, even when controlling for age-related differences in the accuracy of both online frequency learning and explicit frequency reports. Thus, although we observed age differences in the learning of environmental regularities *and* in their influence on subsequent associative memory encoding, our developmental memory effects cannot be fully explained by differences in initial learning.

Our neural results further suggest that developmental differences in memory were driven by both knowledge of the structure of the environment and *use* of that knowledge to guide encoding. Specifically, we observed age-related increases in both overall PFC engagement as well as its value-based modulation, which may reflect developmental differences in the engagement of strategic control. Our finding that lateral prefrontal cortex activation during encoding of high- vs. low-value information may underpin memory selectivity is also in line with prior studies of motivated memory in

*older* adults (*Cohen et al., 2016*). Older adults have been shown to demonstrate decreased neural activation in response to value cues (*Cohen et al., 2016*; *Geddes et al., 2018*) but preserved memory selectivity (*Castel, 2007*; *Castel et al., 2002*; *Cohen et al., 2016*) supported by the strategic recruitment of the left lateral PFC during encoding of high- vs. low-value information (*Cohen et al., 2016*). The PFC may support enhanced attention (*Uncapher et al., 2011*; *Uncapher and Wagner, 2009*) and semantic elaboration (*Kirchhoff and Buckner, 2006*) during encoding, and more focused (*Wais et al., 2012*) and organized search and selection (*Badre and Wagner, 2007*; *Yu et al., 2018*) during retrieval. From childhood to young adulthood, individuals demonstrate improvements in the implementation of strategic memory processes (*Bjorkland et al., 2009*; *Yu et al., 2018*), which are paralleled by increases in PFC recruitment during both encoding and retrieval (*Ghetti et al., 2010*; *Ghetti and Fandakova, 2020*; *Ofen et al., 2007*; *Shing et al., 2016*; *Tang et al., 2018*). In line with this prior work, we similarly observed age-related improvements in overall memory performance and in prefrontal recruitment during encoding and retrieval of novel associations.

The development of adaptive memory requires not only the implementation of encoding and retrieval strategies, but also the flexibility to up- or down-regulate the engagement of control in response to momentary fluctuations in information value (*Castel et al., 2007*; *Castel et al., 2013*; *Hennessee et al., 2017*). Importantly, value-based modulation of lateral PFC engagement during encoding mediated the relation between age and memory selectivity, suggesting that developmental change in both the representation of learned value and value-guided cognitive control may underpin the emergence of adaptive memory prioritization. Prior work examining other neurocognitive processes, including response inhibition (*Insel et al., 2017*) and selective attention (*Störmer et al., 2014*), has similarly found that increases in the flexible upregulation of control in response to value cues enhance goal-directed behavior across development (*Davidow et al., 2018*), and may depend on the engagement of both striatal and prefrontal circuitry (*Hallquist et al., 2018*; *Insel et al., 2017*). Here, we extend these past findings to the domain of memory, demonstrating that value signals derived from the structure of the environment increasingly elicit prefrontal cortex engagement and strengthen goal-directed encoding across childhood and into adolescence.

Further, we also demonstrate that in the absence of *explicit* value cues, the engagement of prefrontal control processes may reflect *beliefs* about information value that are learned through experience. Here, we found that differential PFC activation during encoding of high- vs. low-value information reflected individual and age-related differences in beliefs about the structure of the environment; participants who represented the average frequencies of the low- and high-frequency items as further apart *also* demonstrated greater value-based modulation of lateral PFC activation. It is important to note, however, that we collected explicit frequency reports *after* associative encoding and retrieval. Thus, the relation between PFC activation and explicit frequency reports may be bidirectional — while participants may have increased the recruitment of cognitive control processes to better encode information they believed was more valuable, the engagement of more elaborative or deeper encoding strategies that led to stronger memory traces may have also increased participants' subjective sense of an item's frequency (*Jonides and Naveh-Benjamin, 1987*).

During retrieval, we continued to observe increased activation of the caudate and dlPFC for high- vs. low-value pairs. However, this activation did not significantly vary as a function of memory difference scores or age, suggesting that the developmental differences in value-guided memory that we observed were likely driven by age-related change in encoding processes.

We found that memory prioritization varied with *quadratic* age, and our follow-up tests probing the quadratic age effect did not reveal evidence for significant age-related change in memory prioritization between late adolescence and early adulthood. However, in our prior behavioral work using a very similar paradigm (*Nussenbaum et al., 2020*), we found that memory prioritization varied with *linear* age only. In line with theoretical proposals (*Davidow et al., 2018*), subtle differences in the control demands between the two tasks (e.g. reducing the number of 'foils' presented on each trial of the memory test here relative to our prior study), may have shifted the age range across which we observed differences in behavior, with the more demanding variant of our task showing more linear age-related improvements into early adulthood. In addition, the specific control demands of our task may have also influenced the age at which value-guided memory *emerged*. Future studies should test whether younger children can modulate encoding based on the value of information if the mnemonic demands of the task are simpler.

One important caveat is that our study was cross-sectional — it will be important to replicate our findings in a longitudinal sample to measure more directly how developmental *changes* in cognitive control within an individual contribute to changes in their ability to selectively encode useful information. Our mediation results, in particular, must be interpreted with caution, as simulations have demonstrated that in cross-sectional samples, variables can emerge as significant mediators of age-related change due largely to statistical artifact (*Hofer et al., 2006*; *Lindenberger et al., 2011*). Indeed, our finding that PFC activation mediates the relation between age and value-guided memory does not necessarily imply that within an individual, PFC development leads to improvements in memory selectivity. Longitudinal work in which individuals' neural activity and memory performance is sampled densely within developmental windows of interest is needed to elucidate the complex relations between age, brain development, and behavior (*Hofer et al., 2006*; *Lindenberger et al., 2011*).

We did not find evidence to support two of our predictions. First, although we initially hypothesized that both the ventral and dorsal striatum may be involved in encoding of high-value information, the activation we observed was largely within the dorsal striatum, a region that may reflect the value of goal-directed actions (*Liljeholm and O'Doherty, 2012*). In our task, rather than each stimulus acquiring intrinsic value during frequency learning, participants may have represented the value of the 'action' of remembering each pair during encoding. Second, we did not observe differences in hippocampal engagement during encoding or retrieval of high- vs. low-value associates. This is somewhat surprising, as prior work has suggested that value cues bolster memory through their influence on medial temporal lobe activity (*Adcock et al., 2006*). However, because our task required the use of *learned* value signals, memory retrieval processes may have been required on every encoding trial to recall the frequency of each item. Thus, all items presented at encoding may have triggered the engagement of hippocampal-dependent retrieval processes (*Squire, 1992*) that underpin many forms of memory-guided behavior including attention (*Stokes et al., 2012*; *Summerfield et al., 2006*) and decision-making (*Murty et al., 2016*; *Shadlen and Shohamy, 2016*; *Wang et al., 2020*). Further, in line with prior work (*Davachi, 2006*), we did observe increased activation in the hippocampus during encoding of associations that were subsequently remembered vs. those that were subsequently forgotten. Medial temporal activation, including hippocampal activation, may thus more strongly reflect successful memory formation — whether or not it was facilitated by value — whereas the caudate and lateral prefrontal cortex may be more sensitive to fluctuations in the engagement of value-guided cognitive control.

There are multiple routes through which value signals influence memory (*Cohen et al., 2019b*), and in many contexts, reward-motivated memory may not require strategic control. Value anticipation and reward delivery lead to dopaminergic release in the VTA, which projects not only to corticostriatal circuits that implement goal-directed strategy selection (*Liljeholm and O'Doherty, 2012*), but also directly to the hippocampus and medial temporal lobes (*Adcock et al., 2006*; *Lisman and Grace, 2005*; *Murty et al., 2017*; *Shohamy and Adcock, 2010*; *Stanek et al., 2019*). Given the earlier development of subcortically restricted circuitry relative to the more protracted development of cortical-subcortical pathways (*Somerville and Casey, 2010*), it may be the case that the *direct* influence of reward on memory develops earlier than the controlled pathway we studied here. Rather than eliciting strategic control through incentivizing successful memory, this pathway can be engaged through direct delivery of rewards or reinforcement signals at the time of encoding (*Ergo et al., 2020*; *Jang et al., 2019*; *Rosenbaum et al., 2020*; *Rouhani et al., 2018*). In one such study, adolescents demonstrated greater reward-based modulation of hippocampal-striatal connectivity than adults, and the strength of this connectivity predicted reward-related memory (*Davidow et al., 2016*). However, other studies have not found evidence for developmental change in the influence of valenced outcomes on memory (*Cohen et al., 2019a*; *Katzman and Hartley, 2020*). The influence of different motivational and reward signals on memory across development may not be straightforward — individual and developmental differences in neurocognitive processes including sensitivity to valenced feedback (*Ngo et al., 2019*; *Rosenbaum et al., 2020*), curiosity (*Fandakova and Gruber, 2021*), and emotional processing (*Adelman and Estes, 2013*; *Eich and Castel, 2016*) may interact, leading to complex relations between age, motivation, and memory performance. Further work is needed to characterize both the influence of different types of reward signals on memory across development, as well as the development of the neural pathways that underlie age-related change in behavior.

The present study contributes to our understanding of the neurocognitive mechanisms that support memory across development. Specifically, we addressed the question of how motivated memory may operate in the absence of explicit value cues by examining the development of the neurocognitive mechanisms that support the *learning* and *use* of information value to guide encoding and retrieval. The present findings suggest that while development is marked by improvements in the ability to learn about the statistical structure of the environment, the emergence of adaptive memory also depends centrally on age-related differences in prefrontal control. Our findings demonstrate that prefrontal cortex development has implications not just for general memory processes but for the selective prioritization of useful information — a key component of adaptive memory throughout the lifespan.

## Materials and methods

### Participants

Ninety participants between the ages of 8.0 and 25.9 years took part in this experiment. Thirty participants were children between the ages of 8.0 and 12.7 years (n = 16 females), 30 participants were adolescents between the ages of 13.0 and 17.7 years (n = 16 females), and 30 participants were adults between the ages of 18.3 and 25.9 years (n = 15 females). Ten additional participants were tested but excluded from all analyses due to excessive motion during the fMRI scan (n = 8; see exclusion criteria below) or technical errors during data acquisition (n = 2). We based our sample size on other functional neuroimaging studies of the development of goal-directed behavior and memory across childhood and adolescence (*Insel et al., 2017*; *Tang et al., 2018*) as well as on our prior behavioral study that showed age-related change in the use of learned value to guide memory (*Nussenbaum et al., 2020*). According to self- or parental-report, participants were right-handed, had normal or corrected-to-normal vision, and no history of diagnosed psychiatric or learning disorders. Participants were recruited via flyers around New York University, and from science fairs and events throughout New York City. Based on self- or parent-report, 35.6% of participants were White, 26.7% were two or more races, 24.4% were Asian, 11.1% were Black and 2.2% were Native American. Additionally, 17.8% of the sample identified as Hispanic.

Research procedures were approved by New York University's Institutional Review Board. Adult participants provided written consent prior to participating in the study. Children and adolescents provided written assent, and their parents or guardians provided written consent on their behalf, prior to their participation. All participants were compensated $60 for the experimental session, which involved a 1 hr MRI scan. Participants were told that they would receive an additional bonus payment based on their performance in the experiment; in reality, all participants received an additional $5 bonus payment.

Prior to participating in the scanning session, child and adolescent participants who had never participated in a MRI study in our lab completed a mock scanning session to acclimate to the scanning environment. Mock scan sessions took place during a separate lab visit, at least one day in advance of scheduled scans. In the mock scanner, participants practiced staying as still as possible. We attached a Wii-mote to their heads, and set it to 'rumble' whenever it sensed that the participant had moved. Participants completed a series of three challenges of increasing duration (10, 30, and 90 s) and decreasing angular tolerance (10, 5, and 2 degrees) in which they tried to prevent the Wii-mote from rumbling by lying very still (*Casey et al., 2018*).

### Experimental tasks

Participants completed two blocks of three tasks (*Figure 1*), a variant of which we used in a previous behavioral study (*Nussenbaum et al., 2020*). Across tasks, participants made responses with two MRI-compatible button boxes, one for each hand. In between tasks, an experimenter reminded participants of the instructions for the next part, and participants viewed a diagram indicating which fingers and buttons they should use to make their responses. The tasks were presented using Psychtoolbox Version 3 (*Brainard, 1997*; *Kleiner et al., 2007*; *Pelli, 1997*) for Matlab 2017a (*Mathworks Inc, 2017*) and displayed on a screen behind the scanner, visible to participants via a mirror attached to the MRI head coil. FMRI BOLD activity was measured over eight functional runs, which ranged in duration from approximately 4 to 7.5 min.

The structure of each block of tasks was identical, but their narratives and stimuli differed. In one set of tasks, participants were told that they had a collection of postcards they needed to mail. Each type of postcard in their collection required a different type of stamp.

In the *frequency-learning* task, participants were told they had to sort through their postcards to learn how many of each type they had. They were told that they had more of some types of postcards relative to other types (e.g. they might have five postcards with the same, specific blue pattern but only one postcard with a specific red pattern [*Figure 1*]). Participants were instructed to try to keep track of how many of each kind of postcard they had, because it would be useful to them later on. Throughout the task, participants viewed 24 images of postcards. Twelve of these images were presented once and 12 of the images were presented five times, such that participants completed 72 trials total. On each trial, a postcard appeared in the center of the screen for 2 s. Across all tasks, stimulus presentation was followed by an inter-trial interval (ITI) of 2–6 s, which consisted of a black screen with a small, white fixation cross. Participants were instructed to press the button under their right index finger when they saw a new postcard they had not seen before and to press the button under their right middle finger when they saw a repeated postcard that they had already seen within the task. Participants were instructed to respond as quickly and as accurately as possible. The specific postcard assigned to each frequency condition (1 or 5) was counterbalanced across participants. The order of image presentation was randomized for each participant.

In the second task, the *associative encoding* task, participants were told that they would learn the correct stamp to put on each type of postcard. Participants were instructed that in the subsequent task, they would have to stamp all of their postcards, earning one point for each postcard stamped correctly. Critically, in the associative encoding task, regardless of the number of each type of postcard that they had (i.e. 1 or 5), participants saw each type of postcard with its corresponding stamp only *once*. Participants were instructed that they would earn more points if they focused on remembering the stamps that went on the types of postcards that they had the most of. Thus, participants had equal exposure to the to-be-encoded associations across frequency conditions. On each trial, participants viewed one of the types of postcards from the frequency task next to an image of a unique stamp (5 s). The stamp-postcard pairs, order of the trials, and side of the screen on which the stamp and postcard appeared were randomized for each participant.

Next, participants completed *retrieval*. In the first part of the retrieval task, participants viewed all 24 unique postcards, one at time. When each postcard appeared, participants also saw four stamps: the correct stamp, a foil stamp that had been presented with a high-frequency postcard in the previous paired-associates task, a foil stamp that had been presented with a low-frequency postcard, and a novel stamp. Participants used the four fingers on their right hands to select one of these four stamps. Participants had six seconds to make their selection. Regardless of when they made their selection, the card and all four stamps remained on the screen for 6 s. After participants selected a stamp, a faint, gray outline appeared around it. No feedback was given until the end of the set of tasks. The order of the postcard and the location of each stamp was randomized for each participant.

After stamping all 24 unique postcards once, participants' memory for the postcards' original frequencies was then probed. Participants again saw all 24 unique postcards, one at a time this time with the numbers 1–7 underneath them, and they were asked to provide *frequency reports*. Participants used three fingers on their left hand and all four fingers on their right hand to select the number that they believed matched the number of times they saw the card in the first task. As in the previous task, participants had six seconds to make their selection. Regardless of when they made their selection, the card and all seven numbers remained on the screen for 6 s. After participants selected a number, a faint, gray outline appeared around it. The order of the postcards was randomized for each participant.

Finally, participants stamped all remaining postcards, such that they completed 48 additional memory test trials (i.e. they stamped each of the postcards in the 5-frequency condition four more times.) These trials were not included in any analyses, but their inclusion ensured that correctly encoding the stamps that belonged on the high-frequency postcards would be more valuable for participants despite each retrieval trial being worth one point. Here, participants had 4 s to make each response. We did not measure neural activation during this run, so each trial was followed by a 500 ms black screen with a white fixation cross. At the end of the memory test, participants saw a screen that displayed how many postcards they stamped correctly.

After completing the three tasks, participants were told that they were going to play a second set of similar games. The second set of tasks was identical to the first, except that the stimuli were changed from postcards and stamps to landscape pictures and picture frames. The order of the stimulus sets was counterbalanced across participants.

Prior to entering the scanner, all participants completed a short task tutorial on a laptop to learn the overall task structure and the instructions for each part. The task tutorial comprised a full set of identical tasks but with only two stimuli within each frequency condition. Participants who first did the tasks with postcards completed a tutorial with four novel postcards and novel stamps; participants who first did the tasks with pictures completed a tutorial with four novel pictures and novel frames.

Child and adolescent participants were administered the Vocabulary and Matrix Reasoning subtests of the Wechsler Abbreviated Scale of Intelligence (WASI) (*Wechsler, 2011*) during the mock scanning session. Adults were administered the same two subtests immediately following their scan. We followed the standard procedure to compute age-normed IQ scores for each participant based on their performance on these two sub-tests.

## Analysis of behavioral data

All behavioral data processing and statistical analyses were conducted in R version 3.5.1 (*R Development Core Team, 2018*). Data were combined across blocks (but we include analyss of block effects on memory performance in Appendix 2 (*Appendix 2—tables 1–4*), Trials in which participants failed to make a response were excluded from analyses. Mixed effects models were run using the 'afex' package version 0.21–2 (*Singmann et al., 2020*). Numeric variables were z-scored across the entire data set prior to their inclusion in each model. To determine the random effects structures of our mixed effects models, we began with the maximal model to minimize Type I errors (*Barr et al., 2013*). We included random participant intercepts and slopes across all fixed effects (except age and WASI scores) and their interactions. We also included random stimulus intercepts and slopes across all fixed effects and their interactions. Because stimuli were randomly paired during associative encoding and only repeated, on average, around four times across participants, our stimulus random effects accounted for individual items (e.g. postcard 1) rather than pairs of items (e.g. postcard 1 and stamp 5). We set the number of model iterations to one million and used the 'bobyqa' optimizer. When the maximal model gave convergence errors or failed to converge within a reasonable timeframe (~24 hr), we removed correlations between random slopes and random intercepts, followed by random slopes for interaction effects, followed by random slopes across stimuli. For full details about the fixed- and random-effects structure of all models, see 'Appendix 3: Full Model Specification and Results.' To test the significance of the fixed effects in our models, we used likelihood ratio tests for logistic models and F tests with Satterthwaite approximations for degrees of freedom for linear models. Mediation analyses were conducted with the 'mediation' R package (*Tingley et al., 2014*) and significance of the mediation effects was assessed via 10,000 bootstrapped samples.

For our memory analyses, trials were scored as 'correct' if the participant selected the correct association from the set of four possible options presented during the memory test, 'incorrect' if the participant selected an incorrect association, and 'missed' if the participant failed to respond within the 6 s response window. Missed trials were excluded from all analyses. Because participants had to select the correct association from four possible options, chance-level performance was 25%. Two child participants performed at or below chance-level on the memory test. They were included in all analyses reported in the manuscript; however, we report full details of the results of our memory analyses when we exclude these two participants in *Appendix 3—table 10*. Importantly, our main findings remain unchanged.

## Image acquisition, preprocessing, and quality assessment

Participants were scanned at New York University's Center for Brain Imaging using a Siemens Prisma 3T MRI scanner with a 64-channel head coil. Anatomical data were acquired with high-resolution, T1- weighted anatomical scans using a magnetization-prepared rapidly acquired gradient echo (MPRAGE) sequence (TR = 2.3 s, TE = 2.3 ms, TI = 0.9s; 8° flip angle;. 9 mm isotropic voxels, field of view = 192 x 256 x 256 voxels; acceleration: GRAPPA 2 in the phase-encoding direction, with 24

reference lines) and T2- weighted anatomical scans using a 3D turbo spin echo (TSE) sequence (T2: TR = 3.2 s, TE = 564 ms, Echo Train Length = 314; 120° flip angle, 9 mm isotropic voxels, field of view = 240 x 256 x 256 voxels; acceleration: GRAPPA 2x2 with 32 reference lines in both the phase- and slice-encoding directions). Functional data were acquired with a T2*-weighted, multi-echo EPI sequence with the following parameters: TR=2s, TEs=12.2, 29.48, 46.76, 64.04 ms; MB factor = 2; acceleration: GRAPPA 2, with 24 reference lines; effective echo spacing:. 245 ms; 44 axial slices; 75° flip angle, 3 mm isotropic voxels, from the University of Minnesota's Center for Magnetic Resonance Research (*Feinberg et al., 2010*; *Moeller et al., 2010*; *Xu et al., 2013*).

All anatomical and functional MRI data were preprocessed using fMRIPrep v.1.5.1rc2 (*Esteban et al., 2019*), a robust preprocessing pipeline that adjusts to create the optimal workflow for the input dataset, and then visually inspected. FMRIPrep uses tedana (for implementation details, see *Kundu et al., 2013*; *Kundu et al., 2012*) to combine each four-echo time series based on the signal decay rate of each voxel, taking a weighted average of the four echoes that optimally balances signal strength and BOLD sensitivity. This approach enables the acquisition of BOLD data with a higher signal-to-noise ratio, giving us greater sensitivity to detect neural effects of interest (*Kundu et al., 2013*). This combined time series was then used in subsequent preprocessing steps (e.g. susceptibility distortion correction, confound estimation, registration). Runs in which more than 15% of TRs were censored for motion (relative motion > 0.9 mm framewise displacement) were excluded from neuroimaging analyses (see *Appendix 1—table 1* for the number of participants included in each analysis). Participants who did not have at least one usable run of each task (frequency learning, associative encoding, retrieval), were excluded from all behavioral and neuroimaging analyses (n = 8), leaving N = 90 participants in our analyzed sample.

## Analysis of fMRI data

Statistical analyses were completed in FSL v. 6.0.2. (*Jenkinson et al., 2012*; *Smith et al., 2004*). Preprocessed BOLD data, registered to fMRIPrep's MNI152 template space and smoothed with a 5 mm Gaussian kernel, were combined across runs via fixed-effects analyses and then submitted to mixed-effects GLM analyses, implemented in FEAT 6.0.0 (*Woolrich et al., 2001*; *Woolrich et al., 2004*), to estimate relevant task effects. For all GLM analyses, nuisance regressors included six motion parameters and their derivatives, framewise displacement values, censored frames, the first six anatomical noise components (aCompCor) from fMRIPrep, and cosine regressors from fMRIprep to perform high-pass filtering of the data. All task-based temporal onset regressors were convolved with a double gamma hemodynamic response function and included temporal derivatives. Analyses were thresholded using a whole-brain correction of $z > 3.1$ and a cluster-defining threshold of $p < 0.05$ using FLAME 1.

## Frequency-learning GLM

Our frequency-learning model included six task-based temporal onset regressors. Trials were divided based on appearance count and frequency condition to create the following regressors: (1) low-frequency items the first (and only) time they appeared, (2) high-frequency items the first time they appeared, (3) high-frequency items the second time they appeared, (4) High-frequency items the third time they appeared, (5) high-frequency items the fourth time they appeared, (6) high-frequency items the fifth time they appeared.

## Repetition suppression analyses

For each stimulus in the high-frequency condition, we examined repetition suppression by measuring activation within a parahippocampal ROI during the presentation of each item during frequency learning. We defined our ROI by taking the peak voxel ($x = 30$, $y = -39$, $z = -15$) from the group-level first > last item appearance contrast for high-frequency items during frequency learning and drawing a 5 mm sphere around it. This voxel was located in the right parahippocampal cortex, though we observed widespread and largely symmetric activation in bilateral parahippocampal cortex. To encompass both left and right parahippocampal cortex within our ROI, we mirrored the peak voxel sphere. For each participant, we modeled the neural response to each appearance of each item using the Least Squares Single approach (*Mumford et al., 2014*). Each first-level model included a regressor for the trial of interest, as well as separate regressors for the

onsets of all other items, grouped by repetition number (e.g. a regressor for item onsets on their first appearance, a regressor for item onsets on their second appearance, etc.). Values that fell outside five standard deviations from the mean level of neural activation across all subjects and repetitions were excluded from subsequent analyses (18 out of 10,320 values; .01% of observations). In addition to examining neural activation as a function of stimulus repetition, we also computed an index of repetition suppression for each high-frequency item by computing the difference in mean beta values within our ROI on its first and last appearance.

### Associative encoding and retrieval GLMs

Our associative encoding and retrieval models included six task-based temporal onset regressors. Trials were divided based on frequency condition (high- vs. low-) and subsequent memory (remembered, forgotten, missed). Missed trials were included as nuisance regressors and not included in any contrasts.

### Associative encoding regions of interest (ROIs)

Given our a priori hypotheses about the role of the prefrontal cortex and striatum in value-guided encoding, we examined neural activation within a prefrontal cortex and striatal ROI. The specific ROIs were determined by taking the peak prefrontal voxel ($x = -51$, $y = 42$, $z = 9$) and the peak striatal voxel ($x = -18$, $y = 12$, $z = 6$) from the group-level high > low value associative encoding contrast and drawing 5 mm spheres around them.

## Acknowledgements

We thank Daphne Valencia, Jamie Greer, Nora Keathley, and Michael Liu for assistance with data collection, Ali Cohen and Gail Rosenbaum for valuable discussions and help with analyses, and the staff of NYU's Center for Brain Imaging, especially Pablo Velasco, for technical support and guidance. This project was supported by a National Science Foundation CAREER Grant (1654393) to CAH, a Jacobs Foundation Early Career Fellowship to CAH and a National Defense Science and Engineering Graduate Fellowship to KN.

## Additional information

### Competing interests

Catherine A Hartley: Reviewing editor, *eLife*. The other author declares that no competing interests exist.

### Funding

| Funder | Grant reference number | Author |
| --- | --- | --- |
| National Science Foundation | 1654393 | Catherine Hartley |
| Jacobs Foundation | Early Career Research Fellowship | Catherine Hartley |
| U.S. Department of Defense | National Defense Science and Engineering Graduate Fellowship | Kate Nussenbaum |

The funders had no role in study design, data collection and interpretation, or the decision to submit the work for publication.

### Author contributions

Kate Nussenbaum, Conceptualization, Data curation, Software, Formal analysis, Funding acquisition, Validation, Investigation, Visualization, Methodology, Writing - original draft, Project administration, Writing - review and editing; Catherine A Hartley, Conceptualization, Resources, Supervision, Funding acquisition, Methodology, Writing - review and editing

## Author ORCIDs

Kate Nussenbaum  https://orcid.org/0000-0002-7185-6880
Catherine A Hartley  https://orcid.org/0000-0003-0177-7295

## Ethics

Human subjects: Research procedures were approved by New York University's Institutional Review Board (IRB-2016-1194). Adult participants provided written consent prior to participating in the study. Children and adolescents provided written assent, and their parents or guardians provided written consent on their behalf, prior to their participation. All participants were compensated $60 for the experimental session, which involved a 1-hour MRI scan.

## Decision letter and Author response

Decision letter https://doi.org/10.7554/eLife.69796.sa1
Author response https://doi.org/10.7554/eLife.69796.sa2

## Additional files

### Supplementary files

• Transparent reporting form

### Data availability

Task code, behavioral data, and analysis code are available on the Open Science Framework: https://osf.io/2fkbj/ Unthresholded z-statistic maps from the neuroimaging analyses are available on Neurovault: https://neurovault.org/collections/BUMNZQXA/ BIDS-formatted neuroimaging data are available on Open Neuro: https://openneuro.org/datasets/ds003499.

The following datasets were generated:

| Author(s) | Year | Dataset title | Dataset URL | Database and Identifier |
|-----------|------|---------------|-------------|-------------------------|
| Nussenbaum K, Hartley CA | 2021 | Developmental change in prefrontal cortex recruitment supports the emergence of value-guided memory | https://osf.io/2fkbj/ | Open Science Framework, 2fkb |
| Nussenbaum K | 2021 | Developmental change in prefrontal cortex recruitment supports the emergence of value-guided memory | https://doi.org/10.18112/openneuro.ds003499.v1.0.1 | OpenNeuro, 10.18112/openneuro.ds003499.v1.0.1 |

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

## Appendix 1

### Participant information

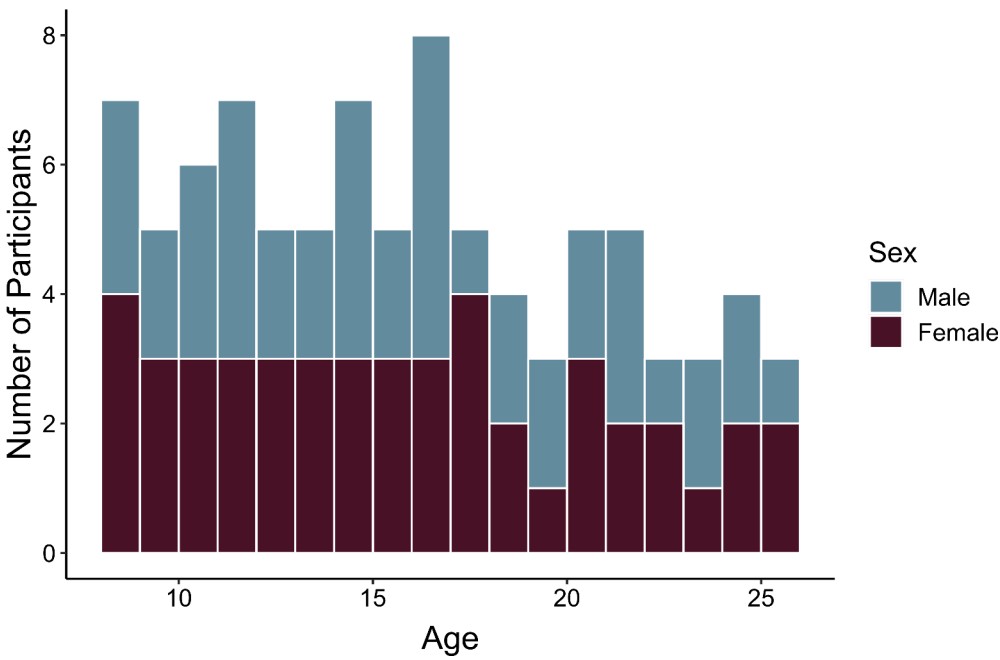

**Appendix 1—figure 1.** Participant age and sex distribution.

**Appendix 1—table 1.** Number of participants included in each analysis.

| Block | Data type | Frequency- learning | Associative encoding | Retrieval | Frequency reports |
|---|---|---|---|---|---|
| 1 | Behavioral | 89 | NA | 90 | 90 |
| 1 | Neural | 88 | 90 | 90 | NA |
| 2 | Behavioral | 86 | NA | 85 | 85 |
| 2 | Neural | 84 | 81 | 81 | NA |

## Appendix 2

### Supplementary results
Frequency learning: Accuracy and reaction times

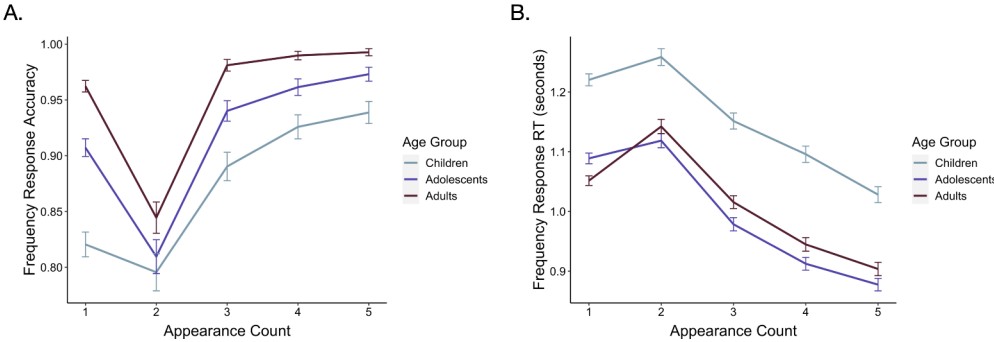

**Appendix 2—figure 1.** Frequency learning accuracy and reaction times. (**A**) During frequency learning, older participants were more accurate in identifying items as new ($\chi2(1)$ = 25.54, p < 0.001) and as repeated ($\chi2(1)$ = 33.81, p < 0.001). All participants became more accurate in identifying items as repeated as the number of repetitions increased ($\chi2$ = 138.20, p < 0.001), though younger participants demonstrated a greater increase in accuracy throughout learning ($\chi2(1)$ = 17.52, p < 0.001). (**B**) Older participants also responded to both new ($F(1, 85.99)$ = 32.51, p < 0.001) and repeated ($F(1, 87.55)$ = 21.82, p < 0.001) items more quickly than younger participants. Reaction times to old items became faster as the a function of item repetition number ($F(1, 69.94)$ = 282.21, p < 0.001).

Relation between age and associative memory: Two-lines test

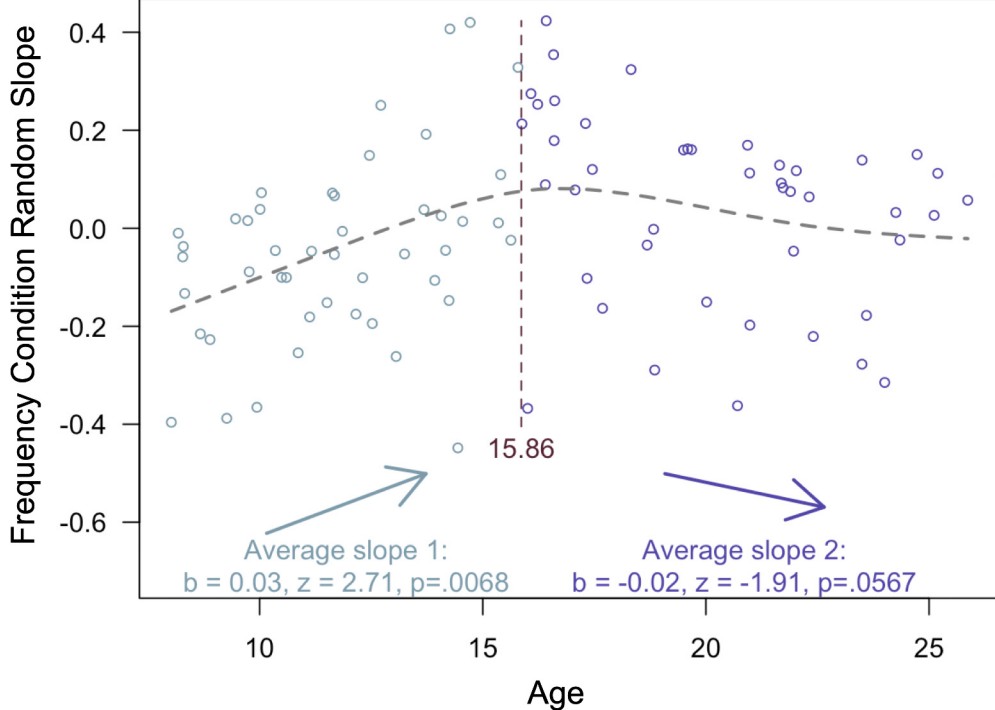

*Appendix 2—figure 2 continued on next page*

**Appendix 2—figure 2.** Relation between age and associative memory. Results from the two-lines test (*Simonsohn, 2018*) revealed that the influence of frequency condition on memory accuracy increased throughout childhood and early adolescence, and did not significantly decrease from adolescence into early adulthood.

## Prefrontal cortex activation during encoding

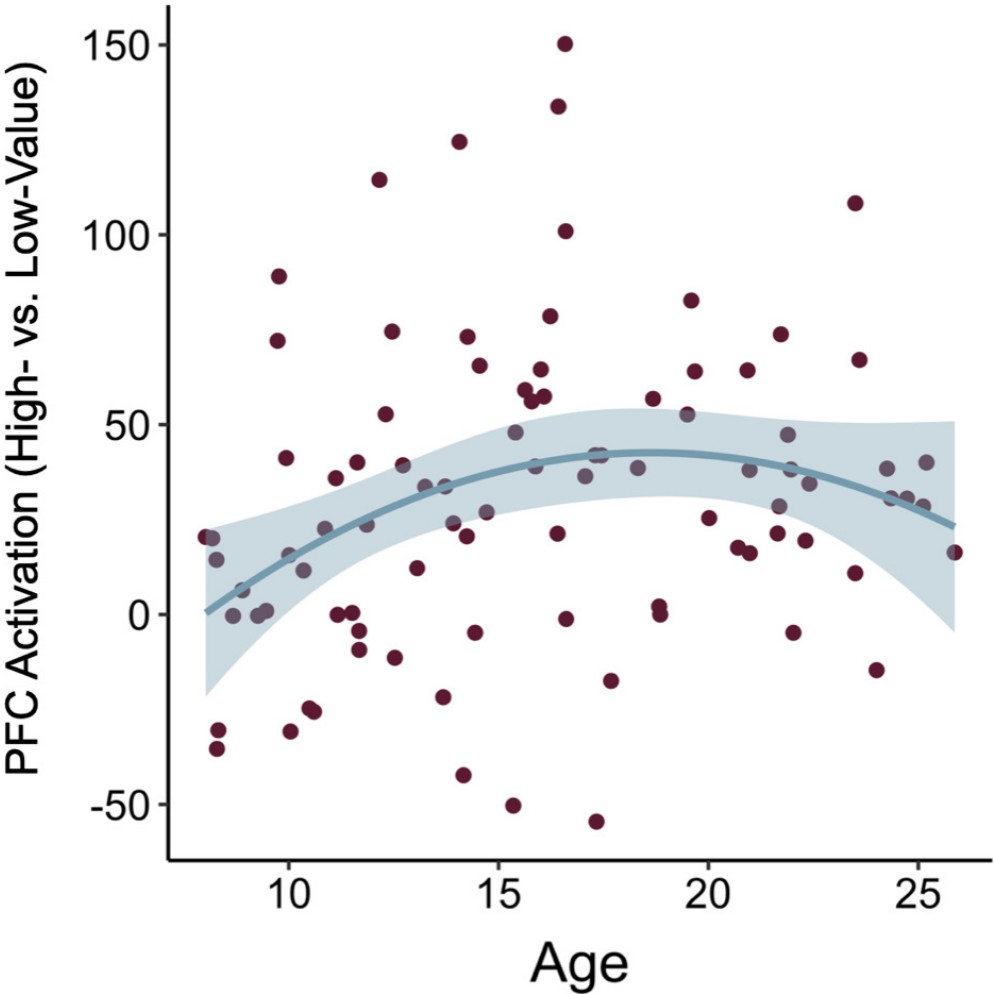

**Appendix 2—figure 3.** Prefrontal cortex activation during encoding. Mean beta weights averaged over voxels within a prefrontal cortex ROI (see 'methods' in main text) during encoding of associations involving high- vs. low-frequency items increased with age. The increase was greatest in childhood before leveling out into late adolescence and early adulthood. The line represents the best-fitting regression line from the model including both linear and quadratic age. The shaded region represents 95% confidence intervals.

## Hippocampal and parahippocampal cortex activation during encoding

A priori, we expected that regions in the medial temporal lobe that have been linked to successful memory formation, including the hippocampus and parahippocampal cortex (*Davachi, 2006*), may be differentially engaged during encoding of high- vs. low- value information. Further, we hypothesized that the differential engagement of these regions across age may contribute to age differences

in value-guided memory. Though we did not see any significant clusters of activation in the hippocampus or parahippocampal cortex in our group level high value vs. low value encoding contrast, we conducted additional ROI analyses to test these hypotheses. As with our other ROI analyses, we first identified the peak voxel (based on its $z$-statistic; hippocampus: $x = 24$, $y = 34$, $z = 23$; parahippocampal cortex: $x = 22$, $y = 41$, $z = 16$) in each region from our group-level contrast, and then drew 5 mm spheres around them. We then examined how average parameter estimates within these spheres related to both age and memory difference scores.

First, we ran a linear regression modeling the effects of age, WASI scores, and their interaction on hippocampal activation. We did not observe a main effect of age on hippocampal activation, ($\beta = 0.00$, SE = 0.10, $p > 0.99$). We did, however, observe a significant age x WASI score interaction effect ($\beta = 0.30$, SE = 0.10, $p = 0.003$). Next, we conducted another linear regression to examine the effects of hippocampal activation, age, WASI scores, and their interaction on memory difference scores. In contrast to our prefrontal cortex activation results, activation in the hippocampus did not relate to memory difference scores, ($\beta = -0.02$, SE = 0.03, $p = 0.50$).

We repeated these analyses with our parahippocampal cortex sphere. Here, we did not observe any significant effects of age on parahippocampal activation ($\beta = -0.07$, SE = 0.11, $p = 0.50$), nor did we observe any effects of parahippocampal activation on memory difference scores ($\beta = 0.01$, SE = 0.03, $p = 0.25$).

## Effects of block order and type on associative memory

### Block order

To examine whether participants' memory varied across blocks, we re-ran our associative memory accuracy model with block order (e.g. 1 or 2) as an additional interacting fixed effect. Our full model included frequency condition, WASI scores, linear and quadratic age, and block order as interacting fixed effects. We included random intercepts and random slopes across frequency condition and block order for each participant, and random intercepts and random slopes across frequency condition, IQ, linear and quadratic age, and block order for each stimulus. We did not observe a significant effect of block order on associative memory ($p = 0.676$; *Appendix 2—table 1*), nor did block order interact with any other predictors (ps > 0.18). Thus, we did not observe any evidence that participants performed the task differently across blocks.

### Block type

To examine whether participants' memory varied across blocks depending on their content, we re-ran our associative memory accuracy model with block type (e.g. pictures/frames or postcards/stamps) as an additional interacting fixed effect. Our full model included frequency condition, WASI scores, linear and quadratic age, and block type as interacting fixed effects. We included random intercepts and random slopes across frequency condition and block type for each participant, and random intercepts and random slopes across frequency condition, IQ, linear and quadratic age, and block type for each stimulus. We did not observe a main effect of block type on associative memory ($p = 0.061$; *Appendix 2—table 2*; *Appendix 2—figure 4*). We did, however, observe a significant block type x frequency condition interaction, such that participants better remembered low-value pairs in the block with the pictures. Importantly, when we included block type as a covariate, we continued to observe a robust influence of frequency condition on associative memory, as well as significant interactions between frequency condition and both age terms.

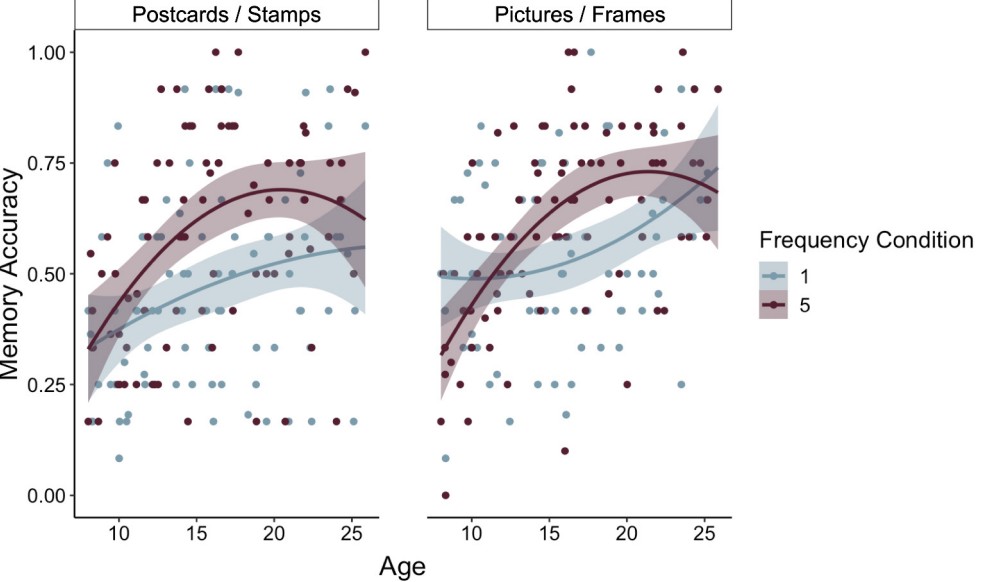

**Appendix 2—figure 4.** Associative memory across blocks. Participants demonstrated a greater influence of frequency condition on associative memory in the task block involving postcards and stamps relative to the task block involving pictures and frames ($\chi2(1) = 0.4.40$, $p = 0.036$).

**Appendix 2—table 1.** Associative memory accuracy by frequency condition with block order.

|  | Estimate | 95% CI | $\chi^2$ | p |
|---|---|---|---|---|
| Intercept | 0.26 | 0.12–0.40 |  |  |
| Age | 1.42 | 0.52–2.33 | 8.97 | 0.003 |
| Age$^2$ | −0.97 | −1.87 - −0.08 | 4.39 | 0.036 |
| WASI | 0.28 | 0.14–0.41 | 15.09 | <0.001 |
| Frequency Condition | −0.22 | −0.31 - −0.13 | 19.47 | <0.001 |
| Block Order | −0.02 | −0.11–0.07 | 0.18 | 0.676 |
| Age x WASI | 0.15 | −0.80–1.10 | 0.10 | 0.756 |
| Age$^2$ x WASI | −0.10 | −1.00–0.80 | 0.04 | 0.833 |
| Age x Block Order | −0.23 | −0.85–0.39 | 0.53 | 0.468 |
| Age$^2$ x Block Order | 0.13 | −0.48–0.75 | 0.18 | 0.675 |
| Age x Frequency Condition | −1.05 | −1.68 - −0.43 | 10.31 | 0.001 |
| Age$^2$ x Frequency Condition | 0.97 | −0.35–1.59 | 8.94 | 0.003 |
| WASI x Frequency Condition | −0.04 | −0.13–0.05 | 0.65 | 0.419 |
| Block Order x Frequency Condition | −0.03 | −0.10–0.05 | 0.76 | 0.384 |
| Block Order x WASI | −0.05 | −0.15–0.04 | 1.13 | 0.288 |
| Age x WASI x Frequency Condition | −0.53 | −1.19–0.13 | 2.43 | 0.119 |
| Age$^2$ x WASI x Frequency Condition | 0.54 | −0.08–1.17 | 2.83 | 0.092 |
| Age x Block Order x Frequency Condition | −0.35 | −0.86–0.16 | 1.80 | 0.180 |
| Age$^2$ x Block Order x Frequency Condition | 0.27 | −0.23–0.77 | 1.10 | 0.294 |
| WASI x Block Order x Frequency Condition | −0.05 | −0.12–0.03 | 1.35 | 0.246 |
| WASI x Age x Block Order | 0.05 | −0.61–0.70 | 0.02 | 0.887 |
| WASI x Age$^2$ x Block Order | −0.01 | −0.63–0.62 | 0.00 | 0.985 |
| Age x WASI x Frequency Condition x Block Order | 0.35 | −0.19–0.89 | 1.61 | 0.205 |
| Age$^2$ x WASI x Frequency Condition x Block Order | −0.33 | −0.85–0.18 | 1.62 | 0.203 |

**Appendix 2—table 2.** Associative memory accuracy by frequency condition with block type.

|  | Estimate | 95% CI | $X^2$ | p |
|---|---|---|---|---|
| Intercept | 0.26 | 0.12–0.40 |  |  |
| Age | 1.40 | 0.49–2.31 | 8.61 | 0.003 |
| $Age^2$ | −0.95 | −1.85 - −0.05 | 4.16 | 0.041 |
| WASI | 0.27 | 0.14–0.40 | 14.46 | <0.001 |
| Frequency Condition | −0.22 | −0.31 - −0.13 | 20.07 | <0.001 |
| Block Type | −0.10 | −0.20–0.00 | 3.52 | 0.061 |
| Age x WASI | 0.17 | −0.79–1.12 | 0.12 | 0.734 |
| $Age^2$ x WASI | −0.10 | −1.01–0.80 | 0.05 | 0.822 |
| Age x Block Type | 0.23 | −0.39–0.85 | 0.53 | 0.465 |
| $Age^2$ x Block Type | −0.28 | −0.89–0.33 | 0.79 | 0.375 |
| Age x Frequency Condition | −1.05 | −1.68 - −0.42 | 10.06 | 0.002 |
| $Age^2$ x Frequency Condition | 0.96 | 0.34–1.59 | 8.64 | 0.003 |
| WASI x Frequency Condition | −0.04 | −0.13–0.05 | 0.65 | 0.419 |
| Block Type x Frequency Condition | −0.08 | −0.15 - −0.01 | 4.40 | 0.036 |
| Block Type x WASI | 0.01 | −0.08–0.10 | 0.03 | 0.866 |
| Age x WASI x Frequency Condition | −0.51 | −1.18–0.15 | 2.24 | 0.135 |
| $Age^2$ x WASI x Frequency Condition | 0.52 | −0.11–1.15 | 2.56 | 0.109 |
| Age x Block Type x Frequency Condition | 0.31 | −0.19–0.82 | 1.44 | 0.230 |
| $Age^2$ x Block Type x Frequency Condition | −0.29 | −0.79–0.21 | 1.28 | 0.258 |
| WASI x Block Type x Frequency Condition | −0.03 | −0.10–0.05 | 0.44 | 0.505 |
| WASI x Age x Block Type | 0.62 | −0.03–1.27 | 3.44 | 0.064 |
| WASI x $Age^2$ x Block Type | −0.60 | −1.22–0.02 | 3.57 | 0.059 |
| Age x WASI x Frequency Condition x Block Type | −0.31 | −0.84–0.23 | 1.28 | 0.258 |
| $Age^2$ x WASI x Frequency Condition x Block Type | 0.34 | −0.17–0.85 | 1.68 | 0.195 |

Given that we observed a significant block type x frequency condition interaction on memory, we next examined whether the relation between age and lateral PFC activation during encoding varied across block type. To do so, we used fslmeans to extract the mean parameter estimate within our PFC ROI for the 5 vs. 1 encoding contrast for each participant, for each block. We then examined how these parameter estimates varied with age. To do so, we ran a linear mixed-effects model with linear age, quadratic age, WASI scores, and block type as interacting fixed effects, and included random participant intercepts. We did not observe a significant effect of block type on PFC activation, nor did it interact with any other predictors (*ps* > 0.22). In line with the analyses reported in the main text, we observed a significant relation between linear age and PFC activation (p = 0.007; Appendix 2 - table 3).

**Appendix 2—table 3.** High- vs. low-value encoding PFC activation by age with block type.

|  | Estimate | 95% CI | df | F | p |
|---|---|---|---|---|---|
| Intercept | −0.01 | −0.17–0.15 |  |  |  |
| Age | 1.60 | 0.47–2.73 | 1, 79.46 | 7.76 | 0.007 |
| $Age^2$ | −1.39 | −2.50 - −0.28 | 1, 79.24 | 6.04 | 0.016 |
| WASI | 0.15 | −0.02–0.32 | 1, 86.82 | 3.06 | 0.084 |
| Block Type | −0.04 | −0.19–0.11 | 1, 80.30 | 0.30 | 0.585 |

*Continued on next page*

*Appendix 2—table 3 continued*

|  | Estimate | 95% CI | df | F | p |
|---|---|---|---|---|---|
| Age x WASI | 0.45 | −0.79–1.69 | 1, 87.35 | 0.50 | 0.479 |
| Age$^2$ x WASI | −0.53 | −1.69–0.63 | 1, 86.03 | 0.79 | 0.376 |
| Age x Block Type | 0.11 | −0.94–1.16 | 1, 78.81 | 0.04 | 0.843 |
| Age$^2$ x Block Type | −0.12 | −1.16–0.91 | 1, 78.59 | 0.05 | 0.818 |
| WASI x Block Type | −0.06 | −0.22–0.10 | 1, 86.37 | 0.54 | 0.462 |
| Age x WASI x Block Type | −0.73 | −1.89–0.44 | 1, 86.58 | 1.49 | 0.226 |
| Age$^2$ x WASI x Block Type | 0.68 | −0.41–1.77 | 1, 85.26 | 1.48 | 0.227 |

Finally, we examined how PFC activation across blocks influenced memory difference scores via a linear mixed-effects model with PFC activation, age, WASI scores, and block type as interacting fixed effects. Our model also included random participant intercepts. Here, including quadratic age did not improve model fit ($X^2(8) = 0.00$, p = 1). We found that PFC activation related to memory difference scores (p = 0.002; *Appendix 2—table 4*). No other main effects or interactions were significant (ps > 0.063).

**Appendix 2—table 4.** Memory difference scores by PFC activation and age with block type.

|  | Estimate | 95% CI | df | F | p |
|---|---|---|---|---|---|
| Intercept | 0.09 | 0.05–0.13 |  |  |  |
| PFC Activation | 0.06 | 0.02–0.10 | 1, 153.78 | 9.85 | 0.002 |
| Age | 0.02 | −0.01–0.06 | 1, 84.35 | 1.52 | 0.221 |
| WASI | −0.00 | −0.04–0.04 | 1, 86.86 | 0.04 | 0.836 |
| Block Type | 0.03 | −0.00–0.07 | 1, 82.48 | 2.92 | 0.091 |
| PFC Activation x Age | 0.01 | −0.03–0.06 | 1, 153.53 | 0.28 | 0.598 |
| PFC Activation x WASI | 0.03 | −0.01–0.08 | 1, 154.29 | 2.13 | 0.147 |
| Age x WASI | −0.01 | −0.05–0.03 | 1, 86.37 | 0.22 | 0.642 |
| PFC Activation x Block Type | −0.01 | −0.04–0.03 | 1, 151.41 | 0.09 | 0.770 |
| Age x Block Type | 0.01 | −0.03–0.04 | 1, 82.91 | 0.13 | 0.718 |
| WASI x Block Type | 0.02 | −0.02–0.05 | 1, 85.40 | 0.76 | 0.385 |
| PFC Activation x Age x WASI | −0.01 | −0.06–0.04 | 1, 152.59 | 0.14 | 0.711 |
| PFC Activation x Age x Block Type | 0.03 | −0.02–0.08 | 1, 152.38 | 1.56 | 0.214 |
| PFC Activation x WASI x Block Type | −0.04 | −0.09–0.00 | 1, 151.64 | 3.49 | 0.064 |
| Age x WASI x Block Type | −0.02 | −0.05–0.01 | 1, 85.20 | 1.23 | 0.270 |
| PFC Activation x Age x WASI x Block Type | 0.05 | −0.01–0.10 | 1, 154.25 | 3.00 | 0.085 |

# Appendix 3

## Full model specification and results

For each model described in the manuscript, we report here its full random-effects structure (when relevant) and effect estimates.

## Model 1: Frequency-learning accuracy: new items

We examined how participants' accuracy in identifying new items during frequency-learning varied as a function of age, WASI scores, and their interaction via a mixed-effects logistic regression (*Appendix 3—table 1*). We included random intercepts for each participant and each stimulus. Including quadratic age in the model did not improve model fit ($X^2(2) = 3.29$, p = 0.19).

**Appendix 3—table 1.** Frequency-learning accuracy: new items.

|  | *Estimate* | *95% CI* | *$X^2$* | *p* |
|---|---|---|---|---|
| Intercept | 3.08 | 2.73–3.44 |  |  |
| Age | 0.92 | 0.58–1.25 | 25.52 | <0.001 |
| WASI | 0.43 | 0.09–0.77 | 6.18 | 0.013 |
| Age x WASI | 0.25 | −0.08–0.57 | 2.25 | 0.134 |

## Model 2: Frequency-learning accuracy: repeated items

We examined how participants' accuracy in identifying repeated items during frequency learning varied as a function of the number of times the item had appeared, age, WASI scores, and their interactions via a mixed-effects logistic regression (*Appendix 3—table 2*). We included random intercepts and random slopes across item appearances for each participant and random intercepts for each stimulus. Including quadratic age did not improve model fit ($X^2(4) = 1.80$, p = 0.77).

**Appendix 3—table 2.** Frequency-learning accuracy: repeated item appearances.

|  | *Estimate* | *95% CI* | *$X^2$* | *p* |
|---|---|---|---|---|
| Intercept | 3.83 | 3.46–4.20 |  |  |
| Appearance | 1.53 | 1.28–1.78 | 138.03 | <0.001 |
| Age | 0.97 | 0.64–1.29 | 33.43 | <0.001 |
| WASI | 0.46 | 0.13–0.79 | 7.58 | 0.006 |
| Appearance x Age | 0.45 | 0.24–0.67 | 17.41 | <0.001 |
| Appearance x WASI | 0.05 | −0.17–0.26 | 0.18 | 0.672 |
| Age x WASI | 0.12 | −0.20–0.45 | 0.57 | 0.449 |
| Appearance x Age x WASI | 0.04 | −0.17–0.25 | 0.14 | 0.707 |

## Model 3: Frequency-learning reaction times: new items

We examined how participants' reaction times when they correctly identified new items during frequency learning varied as a function of age, WASI scores, and their interaction via a mixed-effects linear regression (*Appendix 3—table 3*). We included random intercepts for each participant and each stimulus, and random slopes across age and WASI scores for each stimulus. We also estimated the correlation between random stimulus intercepts and slopes. Including quadratic age did not improve model fit ($X^2(13) = 21.26$, p = 0.068).

**Appendix 3—table 3.** Frequency-learning reaction times: new items.

|  | *Estimate* | *95% CI* | *df* | *F* | *p* |
|---|---|---|---|---|---|
| Intercept | 1.12 | 1.09–1.15 |  |  |  |

*Continued on next page*

|  | Estimate | 95% CI | df | F | p |
|---|---|---|---|---|---|
| Age | −0.08 | −0.11 - −0.05 | 1, 85.99 | 32.51 | <.001 |
| WASI | −0.01 | −0.04–0.02 | 1, 82.34 | 0.56 | .457 |
| Age x WASI | −0.02 | −0.05 - −0.01 | 1, 83.14 | 2.12 | .149 |

## Model 4: Frequency-learning reaction times: repeated items

We examined how participants' reaction times when they correctly identified repeated items during frequency learning varied as a function of the number of times the item had appeared, age, WASI scores, and their interactions via a mixed-effects linear regression (*Appendix 3—table 4*). We included random intercepts and random slopes across item appearances for each participant, random intercepts and slopes across age, WASI scores, and item appearances for each stimulus, and estimated the correlation between random stimulus intercepts and slopes. Including quadratic age did not improve model fit ($X^2(9) = 3.18$, p = 0.96).

**Appendix 3—table 4.** Frequency-learning reaction times: repeated items.

|  | Estimate | 95% CI | df | F | p |
|---|---|---|---|---|---|
| Intercept | 1.03 | 1.00–1.06 |  |  |  |
| Age | −0.07 | −0.10 − −0.04 | 1, 87.55 | 21.82 | <0.001 |
| WASI | −0.03 | −0.06 − −0.01 | 1, 86.27 | 2.65 | 0.108 |
| Appearance | −0.08 | −0.09 − −0.07 | 1, 69.94 | 282.21 | <0.001 |
| Age x WASI | −0.01 | −0.03–0.02 | 1, 84.97 | 0.22 | 0.641 |
| Age x Appearance | −0.01 | −0.01–0.00 | 1, 77.06 | 1.26 | 0.265 |
| WASI x Appearance | 0.00 | −0.01–0.01 | 1, 75.79 | 0.00 | 0.992 |
| Age x WASI x Appearance | 0.00 | −0.01–0.01 | 1, 74.96 | 0.68 | 0.413 |

## Model 5: Parahippocampal cortex neural activation by stimulus repetition and age

For items in the high-frequency condition, we examined how neural activation in a parahippocampal cortex ROI varied as a function of age, quadratic age, stimulus repetition number, quadratic stimulus repetition number, WASI scores, and their interactions (*Appendix 3—table 5*). We included random intercepts for each participant and stimulus, random slopes across linear and quadratic repetition number for each participant, and random slopes across linear and quadratic repetition number, linear and quadratic age, and their interactions for each stimulus stimuli.

**Appendix 3—table 5.** Parahippocampal cortex neural activation by stimulus repetition and age.

|  | Estimate | 95% CI | df | F | p |
|---|---|---|---|---|---|
| Intercept | 65.70 | 52.23–79.17 |  |  |  |
| Age | −78.41 | −164.25–7.43 | 1, 82.78 | 3.21 | 0.077 |
| Age$^2$ | 82.54 | −2.27–167.35 | 1, 82.77 | 3.64 | 0.060 |
| Repetition | −30.20 | −40.89 − −19.50 | 1, 5015.94 | 30.64 | <0.001 |
| Repetition$^2$ | 14.52 | 4.10–24.93 | 1, 9881.00 | 7.47 | 0.006 |
| WASI | −1.11 | −13.49–11.27 | 1, 83.31 | 0.03 | 0.861 |
| Age x Repetition | 101.65 | 27.40–175.90 | 1, 7267.46 | 7.20 | 0.007 |
| Age x Repetition$^2$ | −88.18 | 161.49 - −14.87 | 1, 9857.85 | 5.56 | 0.018 |
| Age$^2$ x Repetition | 97.99 | −171.28 - −24.71 | 1, 7260.70 | 6.87 | 0.009 |

*Continued on next page*

*Appendix 3—table 5 continued*

| | Estimate | 95% CI | df | F | p |
|---|---|---|---|---|---|
| Age² x Repetition² | 82.76 | 10.40–155.11 | 1, 9854.92 | 5.03 | 0.025 |
| WASI x Age | 28.87 | −61.72–119.47 | 1, 83.51 | 0.39 | 0.534 |
| WASI x Age² | −20.73 | −106.34–64.88 | 1, 83.47 | 0.23 | 0.636 |
| WASI x Repetition | −7.56 | −18.46–3.35 | 1, 7402.99 | 1.84 | 0.175 |
| WASI x Repetition² | 7.40 | −3.38–18.18 | 1, 7857.10 | 1.81 | 0.178 |
| WASI x Age x Repetition | −52.32 | −130.26–25.61 | 1, 7243.45 | 1.73 | 0.188 |
| WASI x Age x Repetition² | 42.15 | −34.79–119.08 | 1, 9868.65 | 1.15 | 0.283 |
| WASI x Age² x Repetition | 45.97 | −27.58–119.53 | 1, 7235.30 | 1.50 | 0.221 |
| WASI x Age² x Repetition² | −38.01 | −110.62–34.59 | 1, 9867.59 | 1.05 | 0.305 |

## Model 6: Repetition suppression indices and age

For items in the high-frequency condition, we examined how repetition suppression varied as a function of age, quadratic age, WASI scores, and their interactions (*Appendix 3—table 6*). We included random intercepts for each participant and stimulus, and random slopes across age, quadratic age, and WASI scores for each stimulus.

**Appendix 3—table 6.** Repetition suppression indices.

| | Estimate | 95% CI | df | F | p |
|---|---|---|---|---|---|
| Intercept | 45.54 | 35.63–55.45 | | | |
| Age | −61.34 | −8.56–10.77 | 1, 78.32 | 3.97 | 0.050 |
| Age² | 66.52 | −121.70 - −0.98 | 1, 77.55 | 4.80 | 0.031 |
| WASI | 1.11 | 7.01–126.03 | 1, 58.06 | 0.05 | 0.823 |
| Age x WASI | 60.90 | −2.53–124.34 | 1, 77.38 | 3.54 | 0.064 |
| Age² x WASI | −51.17 | −111.03–8.70 | 1, 77.16 | 2.81 | 0.098 |

## Model 7: Frequency report error magnitudes

We examined how the magnitude of participants' errors in their frequency reports varied as a function of age, WASI scores, frequency condition, and their interactions via a mixed-effects linear regression (*Appendix 3—table 7*). We included random intercepts and random slopes across frequency conditions for each participant and random intercepts and random slopes across age, WASI scores, and frequency conditions for each stimulus. Including quadratic age did not improve model fit ($X^2(8) = 7.96$, p = 0.437).

**Appendix 3—table 7.** Frequency report error magnitudes.

| | Estimate | 95% CI | df | F | p |
|---|---|---|---|---|---|
| Intercept | 1.21 | 1.12–1.30 | | | |
| Age | −0.18 | −0.27 - - 0.10 | 1, 94.30 | 17.57 | <0.001 |
| WASI | −0.11 | −0.19 - −0.02 | 1, 83.80 | 6.47 | 0.014 |
| Frequency Condition | −0.00 | −0.11–0.11 | 1, 93.81 | 0.00 | 0.993 |
| Age x WASI | −0.07 | −0.15–0.01 | 1, 86.35 | 3.24 | 0.075 |
| Age x Frequency Condition | −0.05 | −0.17–0.06 | 1, 85.48 | 0.95 | 0.332 |
| WASI x Frequency Condition | −0.03 | −0.14–0.09 | 1, 86.55 | 0.20 | 0.652 |
| Age x WASI x Frequency Condition | −0.07 | −0.17–0.03 | 1, 85.94 | 1.77 | 0.187 |

## Model 8: Frequency reports by repetition suppression indices

For items in the high-frequency condition, we examined how frequency reports varied as a function of age, WASI scores, repetition suppression, and their interactions via a mixed-effects linear regression (*Appendix 3—table 8*). We included random intercepts and random slopes across repetition suppression for each participant, and random intercepts and random slopes across repetition suppression, WASI scores, age, and their interactions for each stimulus. Including a quadratic age term did not improve model fit ($X^2(8) = 6.45$, p = 0.60).

**Appendix 3—table 8.** Frequency reports by repetition suppression.

|  | Estimate | 95% CI | df | F | p |
|---|---|---|---|---|---|
| Intercept | 4.44 | 4.26–4.63 | | | |
| Age | 0.26 | 0.09–0.42 | 1, 82.87 | 8.93 | 0.004 |
| WASI | 0.19 | 0.02–0.36 | 1, 85.19 | 4.77 | 0.032 |
| Repetition Suppression | 0.00 | −0.06–0.07 | 1, 1360.74 | 0.01 | 0.903 |
| Age x WASI | 0.09 | −0.07–0.25 | 1, 84.13 | 1.34 | 0.251 |
| Age x Repetition Suppression | 0.06 | −0.00–0.12 | 1, 938.87 | 3.61 | 0.058 |
| WASI x Repetition Suppression | 0.04 | −0.02–0.10 | 1, 58.81 | 1.52 | 0.222 |
| Age x WASI x Repetition Suppression | −0.03 | −0.08–0.03 | 1, 313.72 | 1.10 | 0.296 |

## Model 9: Associative memory accuracy

We examined how memory accuracy varied as a function of age, quadratic age, WASI scores, frequency condition, and their interactions via a mixed-effects logistic regression (*Appendix 3—table 9*). We included random intercepts and random slopes across frequency conditions for each participant, and random intercepts and random slopes across frequency condition, WASI scores, age, and quadratic age for each stimulus.

**Appendix 3—table 9.** Associative memory accuracy by frequency condition.

|  | Estimate | 95% CI | $X^2$ | p |
|---|---|---|---|---|
| Intercept | 0.26 | 0.12–0.40 | | |
| Age | 1.38 | 0.49–2.28 | 8.68 | 0.003 |
| Age$^2$ | −0.95 | −1.83 − −0.06 | 4.24 | 0.039 |
| WASI | 0.26 | 0.13–0.39 | 14.18 | <0.001 |
| Frequency Condition | −0.21 | −0.30 − −0.13 | 19.73 | <0.001 |
| Age x WASI | 0.18 | −0.76–1.12 | 0.14 | 0.704 |
| Age$^2$ x WASI | −0.12 | −1.01–0.77 | 0.07 | 0.789 |
| Age x Frequency Condition | −1.06 | −1.68 − −0.45 | 10.74 | 0.001 |
| Age$^2$ x Frequency Condition | 0.98 | 0.37–1.59 | 9.27 | 0.002 |
| WASI x Frequency Condition | −0.04 | −0.13–0.05 | 0.86 | 0.355 |
| Age x WASI x Frequency Condition | −0.50 | −1.15–0.15 | 2.26 | 0.133 |
| Age$^2$ x WASI x Frequency Condition | 0.52 | −0.10–1.13 | 2.65 | 0.104 |

## Model 10: Associative memory accuracy excluding participants who performed below chance

Two participants (both children) responded correctly to 25% or fewer memory test trials. We re-ran our memory accuracy model, excluding these two participants (*Appendix 3—table 10*).

**Appendix 3—table 10.** Associative memory accuracy by frequency condition (below-chance subjects excluded).

|  | Estimate | 95% CI | $X^2$ | p |
|---|---|---|---|---|
| Intercept | 0.30 | 0.16–0.44 |  |  |
| Age | 1.19 | 0.29–2.09 | 6.48 | 0.011 |
| Age$^2$ | −0.79 | −1.68–0.10 | 2.98 | 0.084 |
| WASI | 0.24 | 0.11–0.37 | 12.44 | <0.001 |
| Frequency Condition | −0.22 | −0.31 – −0.13 | 20.04 | <0.001 |
| Age x WASI | 0.31 | −0.62–1.24 | 0.43 | 0.513 |
| Age$^2$ x WASI | −0.23 | −1.11–0.66 | 0.25 | 0.615 |
| Age x Frequency Condition | −1.07 | −1.70 - −0.43 | 10.25 | 0.001 |
| Age$^2$ x Frequency Condition | 0.99 | 0.36–1.61 | 8.97 | 0.003 |
| WASI x Frequency Condition | −0.04 | −0.14–0.05 | 0.81 | 0.368 |
| Age x WASI x Frequency Condition | −0.50 | −1.16–0.16 | 2.17 | 0.141 |
| Age$^2$ x WASI x Frequency Condition | 0.52 | −0.11–1.15 | 2.54 | 0.111 |

## Model 11: Associative memory accuracy controlling for individual differences in frequency learning

We examined how memory accuracy varied as a function of age, quadratic age, WASI scores, frequency condition, and their interactions via a mixed-effects logistic regression (*Appendix 3—table 11*). We also included mean frequency report error magnitudes, overall mean accuracy during frequency learning, and mean accuracy on the last appearance of each item during frequency learning as non-interacting fixed effects. We included random intercepts and random slopes across frequency conditions for each participant, and random intercepts and random slopes across frequency condition, WASI scores, age, and quadratic age for each stimulus.

**Appendix 3—table 11.** Associative memory accuracy by frequency condition (with frequency-learning covariates).

|  | Estimate | 95% CI | $X^2$ | p |
|---|---|---|---|---|
| Intercept | 0.25 | 0.12–0.38 |  |  |
| Age | 0.59 | −0.31–1.49 | 1.62 | 0.203 |
| Age$^2$ | −0.30 | −1.17–0.56 | 0.47 | 0.491 |
| WASI | 0.16 | 0.03–0.29 | 6.02 | 0.014 |
| Frequency Condition | −0.21 | −0.30 – −0.13 | 19.65 | <0.001 |
| Mean Frequency Report Error Magnitude | −0.26 | −0.39 – −0.12 | 13.05 | <0.001 |
| Frequency-learning Accuracy | 0.17 | −0.04–0.38 | 2.36 | 0.125 |
| Frequency-learning Accuracy (last item appearance) | −0.10 | −0.30–0.09 | 1.06 | 0.304 |
| Age x WASI | 0.11 | −0.75–0.96 | 0.06 | 0.807 |
| Age$^2$ x WASI | −0.08 | −0.89–0.72 | 0.04 | 0.843 |
| Age x Frequency Condition | −1.06 | −1.67 - −0.44 | 10.59 | 0.001 |
| Age$^2$ x Frequency Condition | 0.97 | 0.36–1.58 | 9.15 | 0.002 |
| WASI x Frequency Condition | −0.04 | −0.13–0.05 | 0.83 | 0.362 |

*Continued on next page*

|  | *Estimate* | *95% CI* | $X^2$ | *p* |
|---|---|---|---|---|
| Age x WASI x Frequency Condition | −0.50 | −1.15–0.15 | 2.22 | 0.136 |
| Age$^2$ x WASI x Frequency Condition | 0.51 | −0.10–1.13 | 2.61 | 0.106 |

## Model 12: High- vs. low-value encoding caudate activation and age

We ran a linear regression to examine the effects of age, WASI scores, and their interaction on differential caudate activation during encoding of high- vs. low-value information (*Appendix 3—table 12*). Including quadratic age did not improve model fit ($X^2(2) = 2.26$, p$p = 0.11$).

**Appendix 3—table 12.** High vs. low-value encoding caudate activation by age.

|  | *Estimate* | *SE* | *t* | *p* |
|---|---|---|---|---|
| Intercept | −0.07 | .104 |  |  |
| Age | 0.16 | .107 | 1.55 | 0.126 |
| WASI | 0.18 | .110 | 1.64 | 0.105 |
| Age x WASI | −0.29 | .101 | −2.86 | 0.005 |

## Model 13: High- vs. low-value encoding PFC activation and age

We ran a linear regression to examine the effects of linear and quadratic age, WASI scores, and their interactions on differential PFC activation during encoding of high- vs. low-value information (*Appendix 3—table 13*).

**Appendix 3—table 13.** High vs. low-value encoding PFC activation by age.

|  | *Estimate* | *SE* | *t* | *p* |
|---|---|---|---|---|
| Intercept | 0.00 | .105 |  |  |
| Age | 1.97 | .743 | 2.65 | 0.009 |
| Age$^2$ | −1.73 | .734 | −2.35 | 0.021 |
| WASI | 0.26 | .109 | 2.34 | 0.022 |
| Age x WASI | 0.93 | .789 | 1.18 | 0.240 |
| Age$^2$ x WASI | −1.02 | .745 | −1.37 | 0.174 |

## Model 14: Relation between frequency reports and associative memory accuracy

We examined how memory accuracy varied as a function of age, quadratic age, WASI scores, frequency reports, and their interactions via a mixed-effects logistic regression (*Appendix 3—table 14*). We included random intercepts and random slopes across frequency reports for each participant, and random intercepts and random slopes across frequency reports, age, quadratic age, and WASI scores for each stimulus.

**Appendix 3—table 14.** Associative memory accuracy by frequency report.

|  | *Estimate* | *95% CI* | $X^2$ | *p* |
|---|---|---|---|---|
| Intercept | 0.26 | 0.12–0.40 |  |  |
| Age | 1.46 | 0.55–2.38 | 9.25 | 0.002 |
| Age$^2$ | −1.01 | −1.92 – −0.11 | 4.64 | 0.031 |
| WASI | 0.27 | 0.13–0.40 | 13.94 | <0.001 |

*Continued on next page*

*Appendix 3—table 14 continued*

|  | Estimate | 95% CI | X² | p |
|---|---|---|---|---|
| Frequency Report | 0.28 | 0.19–0.37 | 31.20 | <0.001 |
| Age x WASI | 0.15 | −0.81–1.12 | 0.09 | 0.759 |
| Age² x WASI | −0.10 | −1.01–0.82 | 0.04 | 0.838 |
| Age x Frequency Report | 1.13 | 0.46–1.79 | 10.37 | 0.001 |
| Age² x Frequency Report | −1.07 | −1.73 − −0.41 | 9.50 | 0.002 |
| WASI x Frequency Report | 0.02 | −0.08–0.11 | 0.10 | 0.754 |
| Age x WASI x Frequency Report | 0.37 | −0.35–1.09 | 1.00 | 0.316 |
| Age² x WASI x Frequency Report | −0.33 | −1.01–0.35 | 0.89 | 0.345 |

## Model 15: Influence of repetition suppression on associative memory accuracy

For associations involving items in the high-frequency condition, we examined how memory accuracy varied as a function of age, quadratic age, WASI scores, repetition suppression, and their interactions via a mixed-effects logistic regression (*Appendix 3—table 15*). We included random intercepts and random slopes across repetition suppression for each participant, and random intercepts and random slopes across repetition suppression, WASI scores, age, quadratic age, and their interactions for each stimulus.

**Appendix 3—table 15.** Associative memory accuracy by repetition suppression.

|  | Estimate | 95% CI | X² | p |
|---|---|---|---|---|
| Intercept | 0.51 | 0.32–0.69 |  |  |
| Age | 2.63 | 1.43–3.83 | 16.87 | <0.001 |
| Age² | −2.13 | −3.31 − −0.94 | 11.55 | <0.001 |
| WASI | 0.31 | 0.14–0.49 | 11.47 | <0.001 |
| Repetition Suppression | 0.23 | 0.10–0.37 | 11.21 | <0.001 |
| Age x WASI | 0.78 | −0.49–2.05 | 1.44 | 0.230 |
| Age² x WASI | −0.75 | −1.95–0.45 | 1.47 | 0.225 |
| Age x Repetition Suppression | −0.42 | −1.32–0.49 | 0.79 | 0.374 |
| Age² x Repetition Suppression | 0.64 | −0.28–1.57 | 1.79 | 0.181 |
| WASI x Repetition Suppression | 0.00 | −0.13–0.14 | 0.00 | 0.954 |
| Age x WASI x Repetition Suppression | −0.17 | −1.02–0.68 | 0.15 | 0.700 |
| Age² x WASI x Repetition Suppression | 0.27 | −0.57–1.10 | 0.37 | 0.541 |

## Model 16: Effects of frequency reports and repetition suppression on associative memory accuracy

For associations involving items in the high-frequency condition, we examined how memory accuracy varied as a function of age, quadratic age, WASI scores, repetition suppression, frequency reports, and their interactions via a mixed-effects logistic regression (*Appendix 3—table 16*). We included random intercepts and random slopes across repetition suppression, frequency reports, and their

interaction for each participant. We also included random intercepts and random slopes across repetition suppression, frequency reports, age, quadratic age, and WASI scores for each stimulus.

**Appendix 3—table 16.** Associative memory accuracy by repetition suppression and frequency reports.

| | Estimate | 95% CI | $X^2$ | p |
|---|---|---|---|---|
| Intercept | 0.51 | 0.32–0.69 | | |
| Repetition Suppression | 0.23 | 0.09–0.36 | 10.25 | 0.001 |
| WASI | 0.26 | 0.08–0.44 | 7.59 | 0.006 |
| Frequency Report | 0.3 | 0.17–0.42 | 21.16 | <0.001 |
| Age | 2.47 | 1.25–3.69 | 14.4 | <0.001 |
| Age$^2$ | −2.02 | −3.23 − −0.81 | 9.98 | 0.002 |
| Repetition Suppression x WASI | 0.00 | −0.13–0.14 | 0.00 | 0.968 |
| Repetition Suppression x Frequency Report | −0.11 | −0.25–0.04 | 2.18 | 0.140 |
| WASI x Frequency Report | −0.05 | −0.18–0.08 | 0.48 | 0.488 |
| Repetition Suppression x Age | −0.12 | −1.05–0.81 | 0.06 | 0.804 |
| Repetition Suppression x Age$^2$ | 0.33 | −0.62–1.27 | 0.45 | 0.503 |
| WASI x Age | 0.92 | −0.36–2.21 | 1.96 | 0.161 |
| WASI x Age$^2$ | −0.89 | −2.12–0.33 | 2.04 | 0.153 |
| Frequency Report x Age | 0.12 | −0.74–0.99 | 0.08 | 0.783 |
| Frequency Report x Age$^2$ | −0.09 | −0.97–0.79 | 0.04 | 0.843 |
| RS x WASI x Frequency Report | −0.04 | −0.19–0.11 | 0.27 | 0.603 |
| RS x WASI x Age | −0.28 | −1.16–0.61 | 0.36 | 0.550 |
| RS x WASI x Age$^2$ | 0.39 | −0.49–1.26 | 0.71 | 0.400 |
| RS x Frequency Report x Age | 0.13 | −0.78–1.03 | 0.07 | 0.786 |
| RS x Frequency Report x Age$^2$ | −0.12 | −1.09–0.85 | 0.06 | 0.809 |
| WASI x Frequency Report x Age | −0.33 | −1.31–0.65 | 0.44 | 0.509 |
| WASI x Frequency Report x Age$^2$ | 0.44 | −0.52–1.40 | 0.79 | 0.374 |
| RS x WASI x Frequency Report x Age | −0.73 | −1.68–0.22 | 2.29 | 0.130 |
| RS x WASI x Frequency Report x Age$^2$ | 0.83 | −0.14–1.80 | 2.86 | 0.091 |

## Model 17: Relation between age and mean repetition suppression indices

We ran a linear regression to examine how average repetition suppression indices (across items, for each participant) varied as a function of age, quadratic age, WASI scores, and their interactions.

**Appendix 3—table 17.** Mean repetition suppression indices by age.

| | Estimate | SE | t | p |
|---|---|---|---|---|
| Intercept | 44.76 | 4.53 | | |
| Age | −59.00 | 31.82 | −1.85 | 0.067 |
| Age$^2$ | 64.93 | 31.37 | 2.07 | 0.042 |
| WASI | 1.86 | 4.72 | 0.39 | 0.695 |
| Age x WASI | 57.06 | 33.89 | 1.68 | 0.096 |
| Age$^2$ x WASI | −48.02 | 31.93 | −1.50 | 0.136 |

## Model 18: Relation between mean repetition suppression indices and neural activation in caudate

We ran a linear regression to examine how differential caudate activation in response to high- vs. low-value information during encoding related to average repetition suppression indices age, WASI scores, and their interactions (*Appendix 3—table 18*). Including quadratic age did not improve model fit ($X^2(4) = 1.37$, $p = 0.25$).

**Appendix 3—table 18.** Caudate activation by repetition suppression indices.

|  | Estimate | SE | t | p |
|---|---|---|---|---|
| Intercept | 8.23 | 1.54 |  |  |
| Repetition Suppression | 2.31 | 1.60 | 1.44 | 0.153 |
| Age | 1.48 | 1.64 | 0.91 | 0.367 |
| WASI | 1.93 | 1.64 | 1.17 | 0.244 |
| Repetition Suppression x Age | 1.29 | 1.42 | 0.91 | 0.366 |
| Repetition Suppression x WASI | 0.26 | 1.54 | 0.17 | 0.864 |
| Age x WASI | −4.40 | 1.50 | −2.94 | 0.004 |
| Repetition Suppression x Age x WASI | −0.09 | 1.22 | −0.07 | 0.945 |

## Model 19: Relation between mean repetition suppression indices and PFC neural activation

We ran a linear regression to examine how differential PFC activation in response to high- vs. low-value information during encoding related to average repetition suppression indices, age, quadratic age, WASI scores, and their interactions (*Appendix 3—table 19*).

**Appendix 3—table 19.** PFC activation by repetition suppression indices.

|  | Estimate | SE | t | p |
|---|---|---|---|---|
| Intercept | 28.66 | 4.51 |  |  |
| Repetition Suppression | −6.78 | 4.83 | −1.40 | 0.165 |
| Age | 103.36 | 35.33 | 2.93 | 0.004 |
| $Age^2$ | −94.25 | 35.13 | −2.68 | 0.009 |
| WASI | 13.59 | 4.74 | 2.87 | 0.005 |
| Repetition Suppression x Age | −54.15 | 34.13 | −1.59 | 0.112 |
| Repetition Suppression x $Age^2$ | 53.85 | 32.79 | 1.64 | 0.105 |
| Repetition Suppression x WASI | −3.28 | 4.49 | −0.73 | 0.467 |
| Age x WASI | 30.43 | 34.44 | 0.88 | 0.380 |
| $Age^2$ x WASI | −38.59 | 32.41 | −1.91 | 0.237 |
| Repetition Suppression x Age x WASI | −41.33 | 29.80 | −1.39 | 0.169 |
| Repetition Suppression x $Age^2$ x WASI | 41.46 | 27.58 | 1.50 | 0.137 |

## Model 20: Relation between age and frequency distance

We ran a linear regression to examine how mean frequency distances related to age, quadratic age, WASI scores, and their interactions (*Appendix 3—table 20*).

**Appendix 3—table 20.** Frequency distance by age.

|  | Estimate | SE | T | p |
|---|---|---|---|---|
| Intercept | 2.23 | .09 |  |  |

*Continued on next page*

*Appendix 3—table 20 continued*

|  | Estimate | SE | T | p |
|---|---|---|---|---|
| Age | 2.75 | .67 | 4.10 | <0.001 |
| Age$^2$ | −2.28 | .66 | −3.44 | <0.001 |
| WASI | 0.38 | .10 | 3.89 | <0.001 |
| Age x WASI | 0.33 | .71 | 0.46 | 0.646 |
| Age$^2$ x WASI | −0.12 | .67 | −0.18 | 0.857 |

## Model 21: Relation between frequency distance and neural activation in caudate

We ran a linear regression to examine how differential caudate activation in response to high- vs. low-value information during encoding related to average repetition suppression indices age, WASI scores, and their interactions (*Appendix 3—table 21*). Including quadratic age did not improve model fit ($X^2(4) = 1.36$, p = 0.25).

**Appendix 3—table 21.** Caudate activation by frequency distance.

|  | Estimate | SE | t | p |
|---|---|---|---|---|
| Intercept | 7.00 | 1.78 |  |  |
| Frequency Distance | 2.55 | 1.86 | 1.37 | 0.175 |
| Age | 1.39 | 1.89 | 0.74 | 0.463 |
| WASI | 1.13 | 1.77 | 0.64 | 0.526 |
| Frequency Distance x Age | 2.27 | 1.77 | 1.29 | 0.202 |
| Frequency Distance x WASI | −1.50 | 1.77 | −0.85 | 0.399 |
| Age x WASI | −5.49 | 1.64 | −3.43 | 0.001 |
| Frequency Distance x Age x WASI | 1.71 | 1.40 | 1.22 | 0.227 |

## Model 22: Relation between frequency distance and PFC neural activation

We ran a linear regression to examine how differential PFC activation in response to high- vs. low-value information during encoding related to frequency distance, age, WASI scores, and their interactions (*Appendix 3—table 22*). Including quadratic age did not improve model fit ($X^2(4) = 1.49$, p = 0.21).

**Appendix 3—table 22.** PFC activation by frequency distance.

|  | Estimate | SE | t | p |
|---|---|---|---|---|
| Intercept | −0.02 | 0.12 |  |  |
| Frequency Distance | 0.42 | 0.12 | 3.36 | 0.001 |
| Age | −0.03 | 0.13 | −0.22 | 0.824 |
| WASI | 0.08 | 0.12 | 0.64 | 0.522 |
| Frequency Distance x Age | −0.18 | 0.12 | −1.51 | 0.136 |
| Frequency Distance x WASI | 0.15 | 0.12 | 1.23 | 0.223 |
| Age x WASI | −0.17 | 0.11 | −1.52 | 0.132 |
| Frequency Distance x Age x WASI | 0.05 | 0.09 | 0.52 | 0.607 |

## Model 23: Effects of frequency distance and PFC neural activation on memory difference scores

We ran a linear regression to examine how memory difference scores were related to differential PFC activation in response to high- vs. low-value information during encoding, frequency distance, age, quadratic age, WASI scores, and their interactions (*Appendix 3—table 23*).

**Appendix 3—table 23.** Memory difference scores by PFC activation and frequency distance.

|  | Estimate | SE | t | p |
|---|---|---|---|---|
| **Intercept** | **0.07** | | | |
| Frequency Distance | −0.02 | 0.03 | −0.55 | 0.582 |
| Age | 0.56 | 0.2 | 2.83 | 0.006 |
| Age$^2$ | −0.51 | 0.2 | −2.6 | 0.012 |
| WASI | 0.02 | 0.03 | 0.85 | 0.398 |
| PFC Activation | 0.08 | 0.04 | 1.95 | 0.055 |
| Frequency Distance x Age | 0.29 | 0.2 | 1.5 | 0.139 |
| Frequency Distance x Age$^2$ | −0.27 | 0.2 | −1.36 | 0.178 |
| Frequency Distance x WASI | 0.01 | 0.03 | 0.38 | 0.709 |
| Age x WASI | 0.03 | 0.19 | 0.17 | 0.866 |
| Age$^2$ x WASI | −0.05 | 0.18 | −0.25 | 0.800 |
| Frequency Distance x PFC Activation | 0.01 | 0.04 | 0.25 | 0.800 |
| Age x PFC Activation | −0.15 | 0.34 | −0.43 | 0.672 |
| Age$^2$ x PFC Activation | 0.21 | 0.34 | 0.62 | 0.538 |
| WASI x PFC Activation | −0.04 | 0.04 | −0.84 | 0.406 |
| Frequency Distance x Age x WASI | −0.36 | 0.22 | −1.66 | 0.102 |
| Frequency Distance x Age$^2$ x WASI | 0.33 | 0.2 | 1.62 | 0.111 |
| Frequency Distance x Age x PFC Activation | −0.07 | 0.27 | −0.24 | 0.809 |
| Frequency Distance x Age$^2$ x PFC Activation | 0.08 | 0.28 | 0.28 | 0.778 |
| Frequency Distance x WASI x PFC Activation | −0.05 | 0.03 | −1.49 | 0.142 |
| Age$^2$ x WASI x PFC Activation | 0.46 | 0.36 | 1.29 | 0.202 |
| Age$^2$ x WASI x PFC Activation | −0.46 | 0.36 | −1.28 | 0.204 |
| Frequency Distance x Age$^2$ x WASI x PFC Activation | 0.27 | 0.27 | 0.99 | 0.324 |
| Frequency Distance x Age$^2$ x WASI x PFC Activation | −0.28 | 0.27 | −1.07 | 0.288 |

# Appendix 4

## Supplemental neural results

**Appendix 4—table 1.** Frequency-learning: Last vs. first item appearance cluster table.

| Region | x | y | z | Cluster size | z-max |
|---|---|---|---|---|---|
| Frontal pole | −42 | 51 | 0 | 2419 | 6.33 |
| Precuneus | 0 | −66 | 33 | 1322 | 8.99 |
| Left lateral occipital cortex / angular gyrus | −57 | −66 | 30 | 1319 | 7.2 |
| Right lateral occipital cortex / angular gyrus | 51 | −63 | 33 | 637 | 6.03 |
| Right middle temporal gyrus | 66 | −33 | −12 | 304 | 5.48 |
| Right cerebellum | 15 | −87 | −27 | 164 | 5.33 |
| Precentral gyrus | 3 | −18 | 75 | 124 | 4.53 |
| Left cerebellum | −42 | −75 | −42 | 92 | 4.72 |
| Left middle temporal gyrus | −57 | -3 | −27 | 73 | 4.44 |
| Left caudate | -6 | 12 | 9 | 62 | 4.91 |
| Right caudate | 9 | 24 | 6 | 60 | 5.04 |
| Occipital pole | -3 | −96 | 12 | 46 | 4.81 |

**Appendix 4—table 2.** Frequency-learning: First vs. last item appearance cluster table.

| Region | x | y | z | Cluster size | z-max |
|---|---|---|---|---|---|
| Right temporal fusiform cortex / lateral occipital cortex / parahippocampal gyrus | 30 | −39 | −15 | 2137 | 8.62 |
| Left temporal fusiform cortex / lateral occipital cortex / parahippocampal gyrus | −33 | −63 | −15 | 1858 | 7.53 |
| Cingulate gyrus | 9 | 9 | 42 | 100 | 6 |
| Right precuneus | 18 | −51 | 9 | 80 | 6.05 |
| Left postcentral gyrus | −51 | −15 | 57 | 74 | 4.73 |
| Right precentral gyrus | 42 | 3 | 33 | 73 | 5.54 |
| Juxtapositional lobule cortex | 6 | 6 | 57 | 24 | 4.11 |
| Left amygdala | −24 | -6 | −15 | 22 | 4.23 |
| Cingulate gyrus | 3 | -3 | 33 | 22 | 4.68 |
| Central opercular cortex | 39 | -3 | 15 | 21 | 4.8 |

**Appendix 4—table 3.** Encoding: Encoding vs. baseline by linear age cluster table.

| Region | x | y | z | Cluster size | z-max |
|---|---|---|---|---|---|
| Right lateral occipital cortex | 45 | −81 | 21 | 132 | 5 |
| Left precentral gyrus / middle frontal gyrus | −54 | 12 | 33 | 120 | 5.9 |
| Left lateral occipital cortex | −45 | −84 | 21 | 54 | 4.69 |
| Left lateral occipital cortex | −27 | −72 | 42 | 50 | 4.32 |
| Right lateral occipital cortex | 30 | −69 | 45 | 44 | 4.17 |
| Left cerebellum | −18 | −45 | −48 | 39 | 4.55 |
| Superior frontal gyrus | -3 | 12 | 60 | 39 | 4.35 |
| Left supramarginal gyrus | −36 | −45 | 36 | 37 | 4.57 |
| Left middle frontal gyrus | −33 | 3 | 63 | 33 | 4.6 |

*Continued on next page*

*Appendix 4—table 3 continued*

| Region | x | y | z | Cluster size | z-max |
|---|---|---|---|---|---|
| Right superior parietal lobule | 33 | −45 | 42 | 31 | 3.76 |
| Right inferior frontal gyrus | 48 | 12 | 30 | 24 | 3.97 |

**Appendix 4—table 4.** Encoding: High- vs. low-value cluster table.

| Region | x | y | z | Cluster size | z-max |
|---|---|---|---|---|---|
| Superior parietal lobule / lateral occipital cortex / temporal occipital fusiform cortex / cerebellum | −33 | −54 | 51 | 4262 | 6.23 |
| Left frontal pole / inferior frontal gyrus / middle frontal gyrus | −51 | 42 | 9 | 1765 | 6.92 |
| Left caudate / thalamus | −18 | 12 | 6 | 232 | 5.67 |
| Right caudate | 18 | 18 | 12 | 54 | 4.17 |
| Right precentral gyrus | 51 | 9 | 33 | 50 | 4.26 |
| Right precentral gyrus | 24 | -9 | 54 | 47 | 4.79 |
| Left cerebellum | -3 | −51 | 0 | 38 | 4.73 |
| Right frontal pole | 51 | 39 | 12 | 35 | 4.53 |
| Right postcentral gyrus | 42 | −36 | 51 | 28 | 4.32 |
| Right putamen | 27 | 15 | -3 | 26 | 4.52 |
| Left thalamus | -3 | −24 | -3 | 23 | 4.65 |
| Left putamen | −30 | −15 | -6 | 20 | 5.7 |

**Appendix 4—table 5.** Encoding: High- vs. low-value by memory difference scores cluster table.

| Region | x | y | z | Cluster size | z-max |
|---|---|---|---|---|---|
| Left lateral occipital cortex | −45 | −69 | -6 | 377 | 4.78 |
| Left middle frontal gyrus / inferior frontal gyrus | −48 | 21 | 27 | 232 | 4.95 |
| Right lateral occipital cortex (inferior) | 39 | −90 | 3 | 189 | 4.89 |
| Right temporal occipital fusiform cortex | 42 | −57 | -9 | 87 | 4.45 |
| Right lateral occipital cortex (superior) | 27 | −66 | 33 | 61 | 4.83 |

**Appendix 4—table 6.** Encoding: Remembered vs. not remembered cluster table.

| Region | x | y | z | Cluster size | z-max |
|---|---|---|---|---|---|
| Right lateral occipital cortex / temporal occipital fusiform gyrus | 48 | −75 | -9 | 1296 | 5.75 |
| Left lateral occipital cortex / temporal occipital fusiform gyrus / inferior temporal gyrus | −48 | −72 | -6 | 1273 | 5.97 |
| Left inferior frontal gyrus | −48 | 9 | 27 | 103 | 4.22 |
| Left inferior frontal gyrus | −57 | 21 | -3 | 40 | 3.81 |
| Right hippocampus / amygdala | 24 | -6 | −21 | 21 | 4.06 |

**Appendix 4—table 7.** Retrieval: Retrieval vs. baseline by linear age cluster table.

| Region | x | y | z | Cluster size | z-max |
|---|---|---|---|---|---|
| Right lateral occipital cortex | 45 | −81 | 21 | 147 | 5.28 |

*Continued on next page*

*Appendix 4—table 7 continued*

| Region | x | y | z | Cluster size | z-max |
|---|---|---|---|---|---|
| Left lateral occipital cortex | −39 | −87 | 21 | 59 | 4.25 |
| Right precentral gyrus / inferior frontal gyrus | 51 | 3 | 21 | 42 | 4.21 |
| Right precentral gyrus | 39 | −12 | 39 | 41 | 4.4 |
| Left postcentral gyrus / supramarginal gyrus | −63 | −24 | 48 | 38 | 4.45 |
| Left precentral gyrus | −60 | 6 | 21 | 36 | 4.39 |
| Right lateral occipital cortex | 27 | −60 | 57 | 34 | 3.76 |
| Cingulate gyrus / left thalamus | 3 | −33 | 0 | 30 | 4.74 |
| Left supramarginal gyrus | −69 | −24 | 24 | 28 | 4.33 |

**Appendix 4—table 8.** Retrieval: High- vs. low-value cluster table.

| Region | x | y | z | Cluster size | z-max |
|---|---|---|---|---|---|
| Precuneus cortex | 0 | −72 | 39 | 299 | 4.81 |
| Left lateral occipital cortex | −33 | −69 | 51 | 236 | 5.28 |
| Left caudate / thalamus | −12 | -6 | 15 | 128 | 5.15 |
| Right cerebellum | 3 | −81 | −30 | 125 | 6.12 |
| Left inferior frontal gyrus / middle frontal gyrus | −48 | 21 | 24 | 116 | 4.57 |
| Left frontal orbital cortex | −30 | 30 | -3 | 88 | 4.82 |
| Right cerebellum | 39 | −69 | −36 | 85 | 4.65 |
| Right caudate | 15 | -3 | 21 | 77 | 4.96 |
| Left middle frontal gyrus | −45 | 12 | 48 | 74 | 4.31 |
| Cerebellum | -3 | −60 | −36 | 60 | 4.57 |
| Left inferior temporal gyrus | −57 | −60 | −15 | 60 | 4.33 |
| Cingulate gyrus | 0 | −33 | 3 | 25 | 4.09 |
| Right frontal orbital cortex | 33 | 33 | 3 | 21 | 4.69 |
| Left frontal pole | −36 | 57 | 3 | 20 | 3.79 |
| Right lateral occipital cortex | 27 | −66 | 42 | 20 | 4.24 |

