## [Decision Letter]

**Acceptance summary:**

This study will be relevant to those interested in the neurodevelopment of reward learning, episodic memory, and memory-guided decision-making. The combination of a clever task and thorough data analysis make this an impactful paper.

**Decision letter after peer review:**

Thank you for submitting your article "Developmental change in prefrontal cortex recruitment supports the emergence of value-guided memory" for consideration by *eLife*. Your article has been reviewed by 3 peer reviewers, one of whom is a member of our Board of Reviewing Editors, and the evaluation has been overseen by Christian Büchel as the Senior Editor. The following individual involved in review of your submission has agreed to reveal their identity: Vishnu Murty (Reviewer #2).

Essential revisions:

All reviewers agreed that your manuscript presents important and interesting results suitable for publication in *eLife*. They generally liked the study and felt that the experimental design and data analysis approaches were clever and thorough. However, reviewers also raised several issues that we would like you to address in a revised version of the manuscript:

1. Do younger subjects not prioritize memory or do they not learn what information is valuable (e.g., R1, comment 3; R2, comment 2; R3, comment 2)?

2. Please address the modest relationship between the repetition suppression index in the parahippocampal cortex and age (e.g., R1, comment 1and2).

3. It would be important to be more specific regarding anatomical subregions in the introduction and discussion (R2, comment 1).

4. It would be interesting to connect the neural measures during initial frequency learning and memory-guided adaptive encoding (e.g., R2, comment 5).

5. Please discuss problems with cross-sectional designs, especially as they relate to mediation analyses (e.g., R3, comment 1).

6. Please discuss the implications of the quadratic effects (e.g., R2, comment 3).

We also encourage you to consider the other suggestions made by individual reviewers.

*Reviewer #1 (Recommendations for the authors):*

1. Given the modest relationship between the repetition suppression index in the parahippocampal cortex and age, it would be important to make sure this relationship is robust. One way would be to use more data to improve the estimation of the repetition suppression index. For instance, data from all presentations could be used to estimate linear and quadratic effects of repetition number. This would not only improve the reliability of the estimates, but also offer the opportunity to examine whether the dynamics of frequency learning across multiple presentations changes with age.

2. At the very least, it would be important to include a scatter plot of repetition index versus age in the main text, and to show the parameter estimates across all repetitions in the parahippocampal cortex (e.g., per age group).

3. Given the effects of age on both initial frequency learning (new vs. old response accuracy and frequency estimation error) and the effects of item frequency on memory, it would be important to control for frequency learning when testing the effects of age on memory prioritization, and whether lateral PFC mediates this relationship.

4. Please include more information on how the four echo time series were combined. What is considered optimal in this regard?

*Reviewer #2 (Recommendations for the authors):*

Before I start critiquing individual components of the study, I want to say I am quite excited about these results, and believe this a very clever design and interesting neuroimaging findings.

1. I found the introduction quite nice, and it really provided a strong foundation for a very timely paper. However, I think more specificity is needed for the predictions based on anatomical targets. For example, the ventral and dorsal lateral PFC are thought to sub-serve quite discrete processes, both of which could be relevant for this study. I think discussing prior work, and the authors own work, with a greater deal of anatomical specificity would help readers better interpret that findings. A similar weakness was in the lack of discussion of where in the prior and work value-related differences emerged. For example there is a lot of discussion of prior work by Davidow and Shohamy, but those all occurred in ventral striatum which does not overlap with the current findings.

2. For the frequency learning task, it would be helpful to report on accuracy and RT for the 5th trial only. It would be extremely important to know if children and adults were entering the second phase of the experiment with the same acquired knowledge, or different amounts of acquired knowledge. If this 5th trial accuracy is different across age groups, the authors need to include this measure as a co-variate in all neural analyses, as age differences in biases towards high frequency information may result from not having learned the information rather than not being able to effectively use the information to guide adaptive encoding.

3. I think more discussion is warranted on why some neurobehavioral targets show quadratic effects while others show linear. Quadratic effects are often predicted in theoretical models, but from my read on the literature rarely show up in developmental analyses. These data could be leveraged to better understand those models by explaining why some processes are quadratic and others are linear.

4. The authors show directly compare the model fits of the mediation models that manipulate directionality.

5. I found the predictors from the frequency analysis predicting behavior during memory encoding/retrieval to perhaps be the most interesting finding in the paper, especially given that both implicit and explicit measures were predicting memory independently. However, then I was left wanting to know (somewhat desperately!), how much these signals related to the lateral PFC and caudate signals seen during memory encoding. I think this type of analysis would really help make the paper a complete package.

6. Regarding the discussion, I think it would be helpful for the authors to discuss a few features of their data and how they relate to development. The first would be the WASI findings which were quite prominent in most analyses, and in a few showed interactions with age. If this measure is a proxy of executive function, discussing the role of executive function for adaptive memory could help provide a more concrete mechanisms of adaptive memory formation across development. Additionally, it would be helpful for the authors to discuss the negative findings during retrieval. The fact that these age-related differences in adaptive memory processes are most likely seeming from encoding versus retrieval is not only highly interesting, but are a major prediction that stems from an animal literature on dopaminergic influences on hippocampal-dependent memory.

7. It would be helpful to report all behavioral results in the youngest sample only, as the interpretation of the data is quite different if children can or cannot perform these processes.

8. Could the authors provide data on differences in behavioral performance for both task content (which could drive motivational differences) or task order (which might lead to practice effects). If notable differences emerge across these factors, I strongly believe they needed to be included as co-variates in ally analyses.

*Reviewer #3 (Recommendations for the authors):*

1) Empirical findings directly comparing cross-sectional and longitudinal effects have demonstrated that cross-sectional analyses of age differences do not readily generalize to longitudinal research (e.g., Raz et al., 2005; Raz and Lindenberger, 2012). Formal analyses have demonstrated that proportion of explained age-related variance in cross-sectional mediation models may stem from various factors, including similar mean age trends, within-time correlations between a mediator and an outcome, or both (Lindenberger et al., 2011; see also Hofer, Flaherty, and Hoffman, 2006; Maxwell and Cole, 2007). Thus, the results of the mediation analysis showing that PFC activation explains age-related variance in memory difference scores, cannot be taken to imply that changes in PFC activation are correlated with changes in value-guided memory. While the general limitations of a cross-sectional study are noted in the Discussion of the manuscript, it would be important to discuss the critical limitations of the mediation analysis. While the main conclusions of the paper do not critically depend on this analysis, it would be important to alert the reader to the limited information value in performing cross-sectional mediation analyses of age variance.

2) It would be helpful to provide more information on how chance memory performance was handled during data analysis, especially as it is more likely to occur in younger participants. Related to this, please connect the points that belong to the same individual in Figure 3 to facilitate evaluation of individual differences in the memory difference scores.

3) I would like to see some consideration of how the different signatures of value learning, repetition suppression and reported item frequency, are related to the observed PFC and caudate effects during memory encoding. Such a discussion would help the reader connect the findings on learning and using information value across development.

4) A point worthy of discussion are the implications of the finding that younger participants demonstrated greater deviations in their frequency reports for the development of value learning, given that frequency reports were found to predict associative memory accuracy.

5) It would be helpful to include (supplementary) figures accompanying the behavioral results from the frequency learning phase.

6) Supplementary Figure S2 – providing an (additional) plot of the estimated age effects would help match the displayed results better to the description in the main text.

---

## [Author Response]

Essential revisions:All reviewers agreed that your manuscript presents important and interesting results suitable for publication in eLife. They generally liked the study and felt that the experimental design and data analysis approaches were clever and thorough. However, reviewers also raised several issues that we would like you to address in a revised version of the manuscript:1. Do younger subjects not prioritize memory or do they not learn what information is valuable (e.g., R1, comment 3; R2, comment 2; R3, comment 2)?

Thank you for raising this important point. Indeed, one of our main findings is that older participants are better both at *learning* the structure of their environments and also at *using* structured knowledge to strategically prioritize memory. In our original manuscript, we described results of a model that included participants’ explicit frequency reports as a predictor of memory. Model comparison revealed that participants’ frequency reports — which we interpret as reflecting their beliefs about the structure of the environment — predicted memory more strongly than the item’s true frequency. In other words, participants’ beliefs about the structure of the environment (even if incorrect) more strongly influenced their memory encoding than the true structure of the environment. Critically, however, frequency reports interacted with age to predict memory (Figure 8). Even when we accounted for age-related differences in knowledge of the structure of the environment, older participants demonstrated a stronger influence of frequency on memory, suggesting they were better able to use their beliefs to control subsequent associative encoding. We have now clarified our interpretation of this model in our discussion on p. 23:

“Importantly, though we observed age-related differences in participants’ learning of the structure of their environment, the strengthening of the relation between frequency reports and associative memory with increasing age suggests that age differences in learning cannot fully account for age differences in value-guided memory. Even when accounting for individual differences in participants’ explicit knowledge of the structure of the environment, older participants demonstrated a stronger relation between their beliefs about item frequency and associative memory, suggesting that they used their beliefs to guide memory to a greater degree than younger participants.”

As noted by the reviewer, however, our initial memory analysis did not account for age-related differences in participants’ initial, online learning of item frequency, and our neural analyses further did not account for age differences in explicit frequency reports. We have now run additional control analyses to account for the potential influence of individual differences in frequency learning on associative memory. Specifically, for each participant, we computed three metrics: (1) their overall accuracy during frequency learning, (2) their overall accuracy for the last presentation of each item during frequency learning (as suggested by Reviewer 2), and (3) the mean magnitude of the error in their frequency reports. We then included these metrics as covariates in our memory analyses.

When we include these control variables in our model, we continue to observe a robust effect of frequency condition (*p* < 0.001) as well as robust interactions between frequency condition and linear and quadratic age (*p*s < 0.003) on associative memory accuracy. We also observed a main effect of frequency error magnitude on memory accuracy (*p* < 0.001). Here, however, we no longer observe main effects of age or quadratic age on overall memory accuracy. Given the relation we observed between frequency error magnitudes and age, the results from this model suggests that there may be age-related improvements in overall memory that influence *both* memory for associations as well as learning of and memory for item frequencies. The fact that age no longer relates to overall memory when controlling for frequency error magnitudes suggest that age-related variance in memory for item frequencies and memory for associations are strongly related within individuals. Importantly, however, age-related variance in memory for item frequencies *did not* explain age-related variance in the influence of frequency condition on associative memory, suggesting that there are developmental differences in the *use* of knowledge of environmental structure to prioritize valuable information in memory that persist even when controlling for age-related differences in initial *learning* of environmental regularities. Given the importance of this analysis in elucidating the relation between the learning of environmental structure and value-guided memory, we have now updated the results in the main text of our manuscript to include them. Specifically, on p. 13, we now write:

“Because we observed age-related differences in participants’ online learning of item frequencies and in their explicit frequency reports, we further examined whether these age differences in initial learning could account for the age differences we observed in associative memory. To do so, we ran an additional model in which we included each participant’s mean frequency learning accuracy, mean frequency learning accuracy on the last repetition of each item, and explicit report error magnitude as covariates. Here, explicit report error magnitude predicted overall memory performance, χ2(1) = 13.05, *p* < 0.001, and we did not observe main effects of age or quadratic age on memory performance (ps > 0.20). However, we continued to observe a main effect of frequency condition, χ2(1) = 19.65 *p* < 0.001, as well as significant interactions between frequency condition and both linear age χ2(1) = 10.59, *p* = 0.001, and quadratic age χ2(1) = 9.15, *p* = 0.002. Thus, while age differences in initial learning related to overall memory performance, they did not account for age differences in the use of environmental regularities to strategically prioritize memory for valuable information.”

In addition, as suggested by the reviewer, we also included the three covariates as control variables in our mediation analysis. When controlling for online frequency learning and explicit frequency report errors, PFC activity continued to mediate the relation between age and memory difference scores. We have now included these results on p. 16 – 17 of the main text:

“Further, when we included quadratic age, WASI scores, online frequency learning accuracy, online frequency learning accuracy on the final repetition of each item, and mean explicit frequency report error magnitudes as control variables in the mediation analysis, PFC activation continued to mediate the relation between linear age and memory difference scores (standardized indirect effect:.56, 95% confidence interval: [.06, 1.35], *p* = 0.023; standardized direct effect; 1.75, 95% confidence interval: [.12,.3.38], *p* = 0.034).”

We also refer to these analyses when we interpret our findings in our discussion. On p. 23, we write:

“In addition, we continued to observe a robust interaction between age and frequency condition on associative memory, even when controlling for age-related change in the accuracy of both online frequency learning and explicit frequency reports. Thus, though we observed age differences in the learning of environmental regularities and in their influence on subsequent associative memory encoding, our developmental memory effects cannot be fully explained by differences in initial learning.”

We thank the reviewer for this constructive suggestion, as we believe these control analyses strengthen our interpretation of age differences in both the learning and use of environmental regularities to prioritize memory.

2. Please address the modest relationship between the repetition suppression index in the parahippocampal cortex and age (e.g., R1, comment 1 and 2).

As recommended, we have now computed neural activation within our parahippocampal region of interest not just for the first and last appearance of each item during frequency learning, but for *all* appearances. Specifically we extended our repetition suppression analysis described in the manuscript to include *all* image repetitions (p. 36 – 37). Our new methods description reads:

“For each stimulus in the high-frequency condition, we examined repetition suppression by measuring activation within a parahippocampal ROI during the presentation of each item during frequency learning. We defined our ROI by taking the peak voxel (*x* = 30, *y* = -39, *z* = -15) from the group-level first > last item appearance contrast for high-frequency items during frequency learning and drawing a 5 mm sphere around it. This voxel was located in the right parahippocampal cortex, though we observed widespread and largely symmetric activation in bilateral parahippocampal cortex. To encompass both left and right parahippocampal cortex within our ROI, we mirrored the peak voxel sphere. For each participant, we modeled the neural response to each appearance of each item using the Least Squares-Separate approach (Mumford et al., 2014). Each first-level model included a regressor for the trial of interest, as well as separate regressors for the onsets of all other items, grouped by repetition number (e.g., a regressor for item onsets on their first appearance, a regressor for item onsets on their second appearance, etc.). Values that fell outside five standard deviations from the mean level of neural activation across all subjects and repetitions were excluded from subsequent analyses (18 out of 10,320 values; .01% of observations). In addition to examining neural activation as a function of stimulus repetition, we also computed an index of repetition suppression for each high-frequency item by computing the difference in mean β values within our ROI on its first and last appearance.”

As suggested, we ran a mixed effects model examining the influence of linear and quadratic age and linear and quadratic repetition number on neural activation. In line with our whole-brain analysis, we observed a robust effect of linear and quadratic repetition number, suggesting that neural activation decreased non-linearly across stimulus repetitions. In addition, we observed significant interactions between our age and repetition number terms, suggesting that repetition suppression increased into early adulthood. Thus, although the relation we observed between age and repetition suppression is modest, the results from our new analyses suggest it is robust. Because these results largely aligned with the pattern of age-related change we observed in our analysis of repetition suppression indices, we continued to use that compressed metric in subsequent analyses looking at relations with behavior. However, we have updated our Results section to include the full analysis taking into account all item repetitions, as suggested. Our updated manuscript now reads (p. 9):

“We next examined whether repetition suppression in the parahippocampal cortex changed with age. We defined a parahippocampal region of interest (ROI) by drawing a 5mm sphere around the peak voxel from the group-level first > last appearance contrast (*x* = 30, *y* = -39, *z* = -15), and mirrored it to encompass both right and left parahippocampal cortex (Figure 2C). For each participant, we modeled the neural response to each appearance of each high-frequency item. We then examined how neural activation changed as a function of repetition number and age. To account for non-linear effects of repetition number, we included linear and quadratic repetition number terms. In line with our whole-brain analysis, we observed a main effect of repetition number, *F*(1, 5016.0) = 30.64, *p* < 0.001, indicating that neural activation within the parahippocampal ROI decreased across repetitions. Further, we observed a main effect of quadratic repetition number, *F*(1, 9881.0) = 7.47, *p* = 0.006, indicating that the reduction in neural activity was greatest across earlier repetitions (Figure 3A). Importantly, the influence of repetition number on neural activation varied with both linear age, *F*(1, 7267.5) = 7.2, *p* = 0.007 and quadratic age , *F*(1, 7260.8) = 6.9, *p* = 0.009. Finally, we also observed interactions between quadratic repetition number and both linear and quadratic age (*p*s < 0.026). These age-related differences suggest that repetition suppression was greatest in adulthood, with the steepest increases occurring from late adolescence to early adulthood (Figure 3).”

For each participant for each item, we also computed a “repetition suppression index” by taking the difference in mean β values within our ROI on each item’s first and last appearance (Ward et al., 2013). These indices demonstrated a similar pattern of age-related variance — we found that the reduction of neural activity from the first to last appearance of the items varied positively with linear age, *F*(1, 78.32) = 3.97, *p* = 0.05, and negatively with quadratic age, *F*(1, 77.55) = 4.8, *p* = 0.031 (Figure 3B). Taken together, our behavioral and neural results suggest that sensitivity to the repetition of items in the environment was prevalent from childhood to adulthood but increased with age.”

In addition, in the main text on p. 10, we have now included the suggested scatter plot (see new Figure 3B) as well as a modified version of our previous figure S2 to show neural activation across all repetitions in the parahippocampal cortex (see new Figure 3A). We thank the reviewer for this helpful suggestion, as we believe these new figures much more clearly illustrate the repetition suppression effects we observed during frequency learning.

3. It would be important to be more specific regarding anatomical subregions in the introduction and discussion (R2, comment 1).

We agree with the reviewer that our introduction and discussion would benefit from more anatomical granularity, and we did indeed have a priori predictions about more specific neural regions that might be involved in our task.

First, we expected that both the ventral and dorsal striatum might be responsive to stimulus value across our age range. Prior work has suggested that activity in the ventral striatum often correlates with the intrinsic value of a stimulus, whereas activity in the dorsal striatum may reflect goal-directed action values (Liljeholm and O’Doherty, 2012). In our task, we expected that high-frequency items may acquire intrinsic value during frequency learning that is then reflected in the striatal response to these items during encoding. However, because participants were not rewarded when they encountered these images, but rather incentivized to encode associations involving them, we hypothesized that the dorsal striatum may represent the value of the ‘action’ of remembering each pair. In line with this prediction, the dorsal striatum, and the caudate in particular, have also been shown to be engaged during value-guided cognitive control (Hikosaka et al., 2014; Insel et al., 2017).

We have now revised our introduction to include greater specificity in our anatomical predictions on p. 3:

“When individuals need to remember information associated with previously encountered stimuli (e.g., the grocery store aisle where an ingredient is located), frequency knowledge may be instantiated as value signals, engaging regions along the mesolimbic dopamine pathway that have been implicated in reward anticipation and the encoding of stimulus and action values. These areas include the ventral tegmental area (VTA) and the ventral and dorsal striatum (Adcock et al., 2006; Liljeholm and O’Doherty, 2012; Shigemune et al., 2014).”

Though we initially predicted that encoding of high-value information would be associated with increased activation in both the ventral and dorsal striatum, the activation we observed was largely within the dorsal striatum, and specifically, the caudate. We have now revised our discussion accordingly on p. 26:

“Though we initially hypothesized that both the ventral and dorsal striatum may be involved in encoding of high-value information, the activation we observed was largely within the dorsal striatum, a region that may reflect the value of goal-directed actions (Liljeholm and O’Doherty, 2012). In our task, rather than each stimulus acquiring intrinsic value during frequency learning, participants may have represented the value of the ‘action’ of remembering each pair during encoding.”

Second, while the ventromedial PFC often reflects value, given the control demands of our task, we expected to see greater activity in the dorsolateral PFC, which is often engaged in tasks that require the implementation of cognitive control (Botvinick and Braver, 2015). Thus, we hypothesized that individuals would show increased activation in the dlPFC during encoding of high- vs. low-value information, and that this activation would vary as a function of age. We have now clarified this hypothesis on p. 3:

“Value responses in the striatum may signal the need for increased engagement of the dorsolateral prefrontal cortex (dlPFC) (Botvinick and Braver, 2015), which supports the implementation of strategic control.”

In our discussion, we review disparate findings in the developmental literature and discuss factors that may contribute to these differences across studies. For example, in our discussion of Davidow et al., (2016), we highlight differences between their task design and the present study, focusing on how their task involved immediate receipt of reward at the time of encoding, while our task incentivized memory accuracy. We further note that studies that involve reward delivery at the time of encoding may engage different neural pathways than those that promote goal-directed encoding. Beyond Davidow et al., (2016), there are no other neuroimaging studies that examine the influence of reward on memory across development. Thus, we cannot relate our present neural findings to prior work on the development of value-guided memory. As we note in our discussion (p. 28), “Further work is needed to characterize both the influence of different types of reward signals on memory across development, as well as the development of the neural pathways that underlie age-related change in behavior.”

4. It would be interesting to connect the neural measures during initial frequency learning and memory-guided adaptive encoding (e.g., R2, comment 5).

Thank you for this valuable suggestion. We agree that it would be interesting to link frequency-learning behavior to neural activity at encoding. As such, we have now conducted additional analyses to explore these relations.

In the original version of our manuscript, we examined behavior at the *item* level through mixed-effects models, and neural activation during encoding at the *participant* level. Thus, to examine the relation between frequency-learning metrics and neural activation at encoding, we created two additional participant-level metrics. For each participant we computed their average *repetition suppression index*, and a measure of *frequency distance*. The average repetition suppression index reflects the overall extent to which the participant demonstrated repetition suppression in response to the fifth presentation of the high-frequency items, and is computed by averaging each participant’s repetition suppression indices across items. We hypothesized that participants who demonstrated the greatest degree of repetition suppression might be the most sensitive to the difference between the 1- and 5-frequency items, and therefore, show the greatest differences in striatal and PFC activation during encoding of high- vs. low-value information. The *frequency distance* metric reflects the average distance between participants’ explicit frequency reports for items that appeared once and items that appeared five times, and is computed by averaging their explicit frequency reports for items in each frequency condition, and then subtracting the average reports in the low-frequency condition from those in the high-frequency condition. We hypothesized that participants with the largest frequency distances might similarly be the most sensitive to the difference between the 1- and 5-frequency items, and therefore, show the greatest differences in striatal and PFC activation during encoding of high- vs. low-value information.

We first wanted to confirm that the relations we observed between repetition suppression, frequency reports, and age, could also be observed at the participant level. In line with our prior, behavioral analyses, we found that age related to both mean repetition suppression indices (marginally; linear age: *p* = 0.067; quadratic age: *p* = 0.042); and frequency distances (linear and quadratic age: *p*s < 0.001).

In addition, we further tested whether these two metrics related to memory performance. In contrast to our item-level findings, we did not observe a significant relation between repetition suppression indices and memory (*p* = 0.83). We did observe an effect of frequency distance on memory performance. Specifically, we observed significant interactions between frequency distance and age (*p* = 0.014) and frequency distance and quadratic age (*p* = 0.021) on memory difference scores, such that the influence of frequency distance on memory difference scores increased with increasing age from childhood to adolescence.

We next examined how mean repetition suppression indices and frequency distances related to differential neural activation during encoding of high- and low-value pairs. In line with our memory findings, we did not observe any significant relations between mean repetition suppression indices and neural activation in the caudate or prefrontal cortex during encoding (*p*s > 0.15).

Frequency distance did not relate to caudate activation during encoding nor did we observe a frequency distance x age interaction effect (*p*s > 0.16). Frequency distance did, however, relate to differential PFC activation during encoding of high- vs. low-value pairs. Specifically, we observed a main effect of frequency distance on PFC activation (*p* = 0.0012), such that participants whose explicit reports of item frequency, were on average, more distinct across frequency conditions, demonstrated increased PFC activation during encoding of pairs involving high- vs. low-frequency items. Interestingly, when we included frequency distance in our model, we no longer observed a significant effect of age on differential PFC activation, nor did we observe a significant frequency distance x age interaction (*p*s > 0.13). These findings suggest that PFC activation during encoding may have, in part, reflected participants’ beliefs about the structure of the environment, with participants demonstrating stronger differential engagement of control processes across conditions when their representations of the conditions themselves were more distinct.

Finally, we examined how age, frequency distance, and PFC activation related to memory difference scores. Here, even when controlling for both frequency distance and PFC activation, we continued to observe main effects of age and quadratic age on memory difference scores (linear age: *p* = 0.006; quadratic age: *p* = 0.001). In line with our analysis of the relation between frequency reports and memory, these results suggest that age-related variance in value-guided memory may depend on both knowledge of the structure of the environment and use of that knowledge to effectively control encoding.

We have now added these results to our manuscript on p. 13 – 14. We write:

“Given the relations we observed between memory and both repetition suppression and frequency reports, we examined whether they related to neural activation in both our caudate and PFC ROI during encoding. […] Importantly, however, even when we accounted for both PFC activation and frequency distances, we continued to observe an effect of age on memory difference scores (β = 0.56, SE = 0.20, p = 0.006), which, together with our prior analyses, suggest that developmental differences in value-guided memory are not driven solely by age differences in beliefs about the structure of the environment but also depend on the use of those beliefs to guide encoding.”

We have added the full model results to Appendix 3.

Given these results, we have now revised our interpretation of our neural data. Our memory analyses demonstrate that across our age range, we observed age-related differences in *both* the acquisition of knowledge of the structure of the environment and in its *use*. Originally, we interpreted the PFC activation as reflecting the use of learned value to guide memory. However, the strong relation we found between frequency distance and PFC activation suggests that the age differences in PFC activation that we observed may also be related to age differences in knowledge of the structure of the environment that governs when control processes should be engaged most strongly. However, these results must be interpreted cautiously. Participants provided explicit frequency reports *after* they completed the encoding and retrieval tasks, and so explicit frequency reports may have been influenced not only by participants’ memories of online frequency learning, but also by the strength with which they encoded the item and its paired associate, and the experience of successfully retrieving it.

We have now revised our discussion to consider these results. On p. 23, we now write,

“Our neural results further suggest that developmental differences in memory were driven by both knowledge of the structure of the environment and *use* of that knowledge to guide encoding.”

On p. 24, we write,

“The development of adaptive memory requires not only the implementation of encoding and retrieval strategies, but also the flexibility to up- or down-regulate the engagement of control in response to momentary fluctuations in information value (Castel et al., 2007, 2013; Hennessee et al., 2017). Importantly, value-based modulation of lateral PFC engagement during encoding mediated the relation between age and memory selectivity, suggesting that developmental change in both the representation of learned value and value-guided cognitive control may underpin the emergence of adaptive memory prioritization. Prior work examining other neurocognitive processes, including response inhibition (Insel et al., 2017) and selective attention (Störmer et al., 2014), has similarly found that increases in the flexible upregulation of control in response to value cues enhance goal-directed behavior across development (Davidow et al., 2018), and may depend on the engagement of both striatal and prefrontal circuitry (Hallquist et al., 2018; Insel et al., 2017). Here, we extend these past findings to the domain of memory, demonstrating that value signals derived from the structure of the environment increasingly elicit prefrontal cortex engagement and strengthen goal-directed encoding across childhood and into adolescence.”

And on p. 25, we have added an additional paragraph:

“Further, we also demonstrate that in the absence of explicit value cues, the engagement of prefrontal control processes may reflect beliefs about information value that are learned through experience. Here, we found that differential PFC activation during encoding of high- vs. low-value information reflected individual and age-related differences in beliefs about the structure of the environment; participants who represented the average frequencies of the low- and high-frequency items as further apart also demonstrated greater value-based modulation of lateral PFC activation. It is important to note, however, that we collected explicit frequency reports after associative encoding and retrieval. Thus the relation between PFC activation and explicit frequency reports may be bidirectional — while participants may have increased the recruitment of cognitive control processes to better encode information they believed was more valuable, the engagement of more elaborative or deeper encoding strategies that led to stronger memory traces may have also increased participants’ subjective sense of an item’s frequency (Jonides and Naveh-Benjamin, 1987).”

5. Please discuss problems with cross-sectional designs, especially as they relate to mediation analyses (e.g., R3, comment 1).

Thank you for raising this critical point. We have expanded our discussion to specifically note the limitations of our mediation analysis and to more strongly emphasize the need for future longitudinal studies to reveal how changes in neural circuitry may support the emergence of motivated memory across development. Specifically, on p. 26, we now write:

“One important caveat is that our study was cross-sectional — it will be important to replicate our findings in a longitudinal sample to more directly measure how developmental changes in cognitive control within an individual contribute to changes in their ability to selectively encode useful information. Our mediation results, in particular, must be interpreted with caution as simulations have demonstrated that in cross-sectional samples, variables can emerge as significant mediators of age-related change due largely to statistical artifact (Hofer, Flaherty, and Hoffman, 2006; Lindenberger et al., 2011). Indeed, our finding that PFC activation mediates the relation between age and value-guided memory does not necessarily imply that within an individual, PFC development leads to improvements in memory selectivity. Longitudinal work in which individuals’ neural activity and memory performance is sampled densely within developmental windows of interest is needed to elucidate the complex relations between age, brain development, and behavior (Hofer, Flaherty, and Hoffman, 2006; Lindenberger et al., 2011).”

6. Please discuss the implications of the quadratic effects (e.g., R2, comment 3).

We agree with the reviewer that more discussion is warranted here. While many cognitive processes tend to improve with increasing age, the significant interaction between quadratic age and frequency condition on memory accuracy could reflect a number of different patterns of developmental variance. Because quadratic curves are U-shaped, the significant interaction between quadratic age and frequency condition could reflect a peak in value-guided memory in adolescence. However, the combination of linear and quadratic effects can also capture “plateauing” effects, where the influence of age on a particular cognitive process decreases at a particular developmental timepoint. To determine how to interpret the quadratic effect of age on value-guided memory — and specifically, to test for the presence of an adolescent peak — we ran an additional analysis.

To test for an adolescent peak in value-guided memory, we first fit our memory accuracy model *without* any age terms, and then extracted the random slope across frequency conditions for each subject. We then conducted a ‘two lines test’ (Simonsohn, 2018) to examine the relation between age and these random slopes. In brief, the two-lines test fits the data with two linear models — one with a positive slope and one with a negative slope, algorithmically determining the breakpoint in the estimates where the signs of the slopes change. When we analyzed our memory data in this way, we found a robust, positive relation between age and value-guided memory (see newly added Appendix 2 – Figure 3) from childhood to mid-adolescence, that peaked around age 16 (age 15.86). From age ~16 to early adulthood, however, we observed only a marginal negative relation between age and value-guided memory (*p* = 0.0567). Thus, our findings do not offer strong evidence in support of an adolescent peak in value-guided memory — instead, they suggest that improvements in value-guided memory are strongest from childhood to adolescence.

To more clearly demonstrate the relation between age and value-guided memory, we have now included the results of the two-lines test in the Results section of our main text. On p. 12 – 13, we write:

“In line with our hypothesis, we observed a main effect of frequency condition on memory, χ2(1) = 21.51, *p* < 0.001, indicating that individuals used naturalistic value signals to prioritize memory for high-value information. Critically, this effect interacted with both linear age (χ2(1) = 11.03, *p* < 0.001) and quadratic age (χ2(1) = 9.51, *p* = 0.002), such that the influence of frequency condition on memory increased to the greatest extent throughout childhood and early adolescence.

To determine whether the interaction between quadratic age and frequency condition on memory accuracy reflected an adolescent peak in value-guided memory prioritization, we re-ran our memory accuracy model without including any age terms, and extracted each participant’s random slope across frequency conditions. We then submitted these random slopes to the “two-lines” test (Simonsohn, 2018), which fits two regression lines with oppositely signed slopes to the data, algorithmically determining where the sign flip should occur. The results of this analysis revealed that the influence of frequency condition on memory significantly increased from age 8 to age 15.86 (b = 0.03, *z* = 2.71, *p* = 0.0068; Appendix 2 – Figure 3), but only marginally decreased from age 15.86 to age 25 (b = -.02, *z* = 1.91, *p* = 0.0576). Thus, the interaction between frequency condition and quadratic age on memory performance suggests that the biggest age differences in value-guided memory occurred through childhood and early adolescence, with older adolescents and adults performing similarly.”

That said, this developmental trajectory is likely specific to the particular demands of our task. In our previous behavioral study that used a very similar paradigm (Nussenbaum, Prentis, and Hartley, 2020), we observed only a linear relation between age and value-guided memory. Although the task used in our behavioral study was largely similar to the task we employed here, there were subtle differences in the design that may have extended the age range through which we observed improvements in memory prioritization. In particular, in our previous behavioral study, the memory test required participants to select the correct associate from a grid of 20 options (i.e., 1 correct and 19 incorrect options), whereas here, participants had to select the correct associate from a grid of 4 options (1 correct and 3 incorrect options). In our prior work, the need to differentiate the ‘correct’ option from many more foils may have increased the demands on either (or both) memory encoding or memory retrieval, requiring participants to encode and retrieve more specific representations that would be less confusable with other memory representations. By decreasing the task demands in the present study, we may have shifted the developmental curve we observed toward earlier developmental timepoints.

We originally did not emphasize our quadratic findings in the discussion of our manuscript because, given the marginal decrease in memory selectivity we observed from age 16 to age 25 and the different age-related findings across our two studies, we did not want to make strong claims about the specific shape of developmental change. However, we agree with the reviewer that these points are worthy of discussion within the manuscript. We have now amended our discussion on p. 25 accordingly:

“We found that memory prioritization varied with quadratic age, and our follow-up tests probing the quadratic age effect did not reveal evidence for significant age-related change in memory prioritization between late adolescence and early adulthood. However, in our prior behavioral work using a very similar paradigm (Nussenbaum et al., 2020), we found that memory prioritization varied with linear age only. In line with theoretical proposals (Davidow et al., 2018), subtle differences in the control demands between the two tasks (e.g., reducing the number of ‘foils’ presented on each trial of the memory test here relative to our prior study), may have shifted the age range across which we observed differences in behavior, with the more demanding variant of our task showing more linear age-related improvements into early adulthood. In addition, the specific control demands of our task may have also influenced the age at which value-guided memory emerged. Future studies should test whether younger children can modulate encoding based on the value of information if the mnemonic demands of the task are simpler.”

We thank the reviewer for this helpful suggestion, and believe our additions that expand on the quadratic age effects help clarify our developmental findings.

We also encourage you to consider the other suggestions made by individual reviewers.

We appreciate the thoughtful comments and suggestions from the reviewers and have addressed all of them, in turn, below. Given the overlap across many of the reviewer comments, some of our responses are repeated where relevant.

Reviewer #1 (Recommendations for the authors):1. Given the modest relationship between the repetition suppression index in the parahippocampal cortex and age, it would be important to make sure this relationship is robust. One way would be to use more data to improve the estimation of the repetition suppression index. For instance, data from all presentations could be used to estimate linear and quadratic effects of repetition number. This would not only improve the reliability of the estimates, but also offer the opportunity to examine whether the dynamics of frequency learning across multiple presentations changes with age.

Thank you for this helpful suggestion. As recommended, we have now computed neural activation within our parahippocampal region of interest not just for the first and last appearance of each item during frequency learning, but for *all* appearances. Specifically we extended our repetition suppression analysis described in the manuscript to include *all* image repetitions (p. 36 – 37). Our new methods description reads:

“For each stimulus in the high-frequency condition, we examined repetition suppression by measuring activation within a parahippocampal ROI during the presentation of each item during frequency learning. […] In addition to examining neural activation as a function of stimulus repetition, we also computed an index of repetition suppression for each high-frequency item by computing the difference in mean β values within our ROI on its first and last appearance.”

As suggested, we ran a mixed effects model examining the influence of linear and quadratic age and linear and quadratic repetition number on neural activation. In line with our whole-brain analysis, we observed a robust effect of linear and quadratic repetition number, suggesting that neural activation decreased non-linearly across stimulus repetitions. In addition, we observed significant interactions between our age and repetition number terms, suggesting that repetition suppression increased into early adulthood. Thus, although the relation we observed between age and repetition suppression is modest, the results from our new analyses suggest it is robust. Because these results largely aligned with the pattern of age-related change we observed in our analysis of repetition suppression indices, we continued to use that compressed metric in subsequent analyses looking at relations with behavior. However, we have updated our Results section to include the full analysis taking into account all item repetitions, as suggested. Our updated manuscript now reads (p. 9):

“We next examined whether repetition suppression in the parahippocampal cortex changed with age. […] These age-related differences suggest that repetition suppression was greatest in adulthood, with the steepest increases occurring from late adolescence to early adulthood (Figure 3).”

For each participant for each item, we also computed a “repetition suppression index” by taking the difference in mean β values within our ROI on each item’s first and last appearance (Ward et al., 2013). These indices demonstrated a similar pattern of age-related variance — we found that the reduction of neural activity from the first to last appearance of the items varied positively with linear age, *F*(1, 78.32) = 3.97, *p* = 0.05, and negatively with quadratic age, *F*(1, 77.55) = 4.8, *p* = 0.031 (Figure 3B). Taken together, our behavioral and neural results suggest that sensitivity to the repetition of items in the environment was prevalent from childhood to adulthood but increased with age.”

2. At the very least, it would be important to include a scatter plot of repetition index versus age in the main text, and to show the parameter estimates across all repetitions in the parahippocampal cortex (e.g., per age group).

In the main text on p. 10, we have now included the suggested scatter plot (see new Figure 3B) as well as a modified version of our previous figure S2 to show neural activation across all repetitions in the parahippocampal cortex (see new Figure 3A). We thank the reviewer for this helpful suggestion, as we believe these new figures much more clearly illustrate the repetition suppression effects we observed during frequency learning.

3. Given the effects of age on both initial frequency learning (new vs. old response accuracy and frequency estimation error) and the effects of item frequency on memory, it would be important to control for frequency learning when testing the effects of age on memory prioritization, and whether lateral PFC mediates this relationship.

Thank you for raising this important point. Indeed, one of our main findings is that older participants are better both at *learning* the structure of their environments and also at *using* structured knowledge to strategically prioritize memory. In our original manuscript, we described results of a model that included participants’ explicit frequency reports as a predictor of memory. Model comparison revealed that participants’ frequency reports — which we interpret as reflecting their beliefs about the structure of the environment — predicted memory more strongly than the item’s true frequency. In other words, participants’ beliefs about the structure of the environment (even if incorrect) more strongly influenced their memory encoding than the true structure of the environment. Critically, however, frequency reports interacted with age to predict memory (Figure 8). Even when we accounted for age-related differences in knowledge of the structure of the environment, older participants demonstrated a stronger influence of frequency on memory, suggesting they were better able to use their beliefs to control subsequent associative encoding. We have now clarified our interpretation of this model in our discussion on p. 23:

“Importantly, though we observed age-related differences in participants’ learning of the structure of their environment, the strengthening of the relation between frequency reports and associative memory with increasing age suggests that age differences in learning cannot fully account for age differences in value-guided memory. Even when accounting for individual differences in participants’ explicit knowledge of the structure of the environment, older participants demonstrated a stronger relation between their beliefs about item frequency and associative memory, suggesting that they used their beliefs to guide memory to a greater degree than younger participants.”

As noted by the reviewer, however, our initial memory analysis did not account for age-related differences in participants’ initial, online learning of item frequency, and our neural analyses further did not account for age differences in explicit frequency reports. We have now run additional control analyses to account for the potential influence of individual differences in frequency learning on associative memory. Specifically, for each participant, we computed three metrics: (1) their overall accuracy during frequency learning, (2) their overall accuracy for the last presentation of each item during frequency learning (as suggested by Reviewer 2), and (3) the mean magnitude of the error in their frequency reports. We then included these metrics as covariates in our memory analyses.

When we include these control variables in our model, we continue to observe a robust effect of frequency condition (*p* < 0.001) as well as robust interactions between frequency condition and linear and quadratic age (*p*s < 0.003) on associative memory accuracy. We also observed a main effect of frequency error magnitude on memory accuracy (*p* < 0.001). Here, however, we no longer observe main effects of age or quadratic age on overall memory accuracy. Given the relation we observed between frequency error magnitudes and age, the results from this model suggests that there may be age-related improvements in overall memory that influence *both* memory for associations as well as learning of and memory for item frequencies. The fact that age no longer relates to overall memory when controlling for frequency error magnitudes suggest that age-related variance in memory for item frequencies and memory for associations are strongly related within individuals. Importantly, however, age-related variance in memory for item frequencies *did not* explain age-related variance in the influence of frequency condition on associative memory, suggesting that there are developmental differences in the *use* of knowledge of environmental structure to prioritize valuable information in memory that persist even when controlling for age-related differences in initial *learning* of environmental regularities. Given the importance of this analysis in elucidating the relation between the learning of environmental structure and value-guided memory, we have now updated the results in the main text of our manuscript to include them. Specifically, on p. 13, we now write:

“Because we observed age-related differences in participants’ online learning of item frequencies and in their explicit frequency reports, we further examined whether these age differences in initial learning could account for the age differences we observed in associative memory. To do so, we ran an additional model in which we included each participant’s mean frequency learning accuracy, mean frequency learning accuracy on the last repetition of each item, and explicit report error magnitude as covariates. Here, explicit report error magnitude predicted overall memory performance, χ2(1) = 13.05, *p* < 0.001, and we did not observe main effects of age or quadratic age on memory performance (ps > 0.20). However, we continued to observe a main effect of frequency condition, χ2(1) = 19.65 *p* < 0.001, as well as significant interactions between frequency condition and both linear age χ2(1) = 10.59, *p* = 0.001, and quadratic age χ2(1) = 9.15, *p* = 0.002. Thus, while age differences in initial learning related to overall memory performance, they did not account for age differences in the use of environmental regularities to strategically prioritize memory for valuable information.”

In addition, as suggested by the reviewer, we also included the three covariates as control variables in our mediation analysis. When controlling for online frequency learning and explicit frequency report errors, PFC activity continued to mediate the relation between age and memory difference scores. We have now included these results on p. 16 – 17 of the main text:

“Further, when we included quadratic age, WASI scores, online frequency learning accuracy, online frequency learning accuracy on the final repetition of each item, and mean explicit frequency report error magnitudes as control variables in the mediation analysis, PFC activation continued to mediate the relation between linear age and memory difference scores (standardized indirect effect:.56, 95% confidence interval: [.06, 1.35], *p* = 0.023; standardized direct effect: 1.75, 95% confidence interval: [.12,.3.38], *p* = 0.034).”

We also refer to these analyses when we interpret our findings in our discussion. On p. 23, we write:

“In addition, we continued to observe a robust interaction between age and frequency condition on associative memory, even when controlling for age-related change in the accuracy of both online frequency learning and explicit frequency reports. Thus, though we observed age differences in the learning of environmental regularities and in their influence on subsequent associative memory encoding, our developmental memory effects cannot be fully explained by differences in initial learning.”

We thank the reviewer for this constructive suggestion, as we believe these control analyses strengthen our interpretation of age differences in both the learning and use of environmental regularities to prioritize memory.

4. Please include more information on how the four echo time series were combined. What is considered optimal in this regard?

As noted in the manuscript, we preprocessed our data with fMRIprep, which uses the tedana T2* pipeline (Kundu et al., 2011; Kundu et al., 2013; Kundu et al., 2017) to combine the four echoes. Images acquired at longer delays after the excitation pulse (longer echo times) have higher signal dropout but greater BOLD sensitivity. Thus the ‘optimal’ combination of echoes takes a weighted average of the four echoes that balances signal strength and sensitivity for each voxel. More specifically, tedana first fits a model to estimate both the total signal in each voxel before decay as well as the rate at which the signal in each voxel decays over time (Kundu et al., 2017). Then, using the estimate for the rate of signal decay, tedana combines the signals across the four echoes using a weighted average, where the “weight” of each echo is determined by:TE*e(−TE/T2*)where TE is the echo time and T2* is the rate of signal decay.

By weighting the echoes in this way, the combined data (which we use for all analyses) takes advantage of the signal strength of the earlier echoes and the sensitivity of the later echoes. Because signal decay is modeled separately for each voxel, this method of combining echoes enables differential weighting of images acquired at shorter and longer TEs for different regions of the brain. Our description of the combination of the echoes as “optimal” is meant to reflect the fact that the echoes are weighted differently for different voxels, depending on the rate at which the signal within them decays. We have now clarified this in the manuscript on p. 35, and emphasized to the reader where detailed information on the implementation of the multi-echo combination procedure can be obtained. We have also added text detailing the benefits of the use of multi-echo sequences. Specifically, we now write:

“FMRIPrep uses tedana (for implementation details, see Kundu et al., 2013, 2012) to combine each four-echo time series based on the signal decay rate of each voxel, taking a weighted average of the four echoes that optimally balances signal strength and BOLD sensitivity. This approach enables the acquisition of BOLD data with a higher signal-to-noise ratio, giving us greater sensitivity to detect neural effects of interest (Kundu et al., 2013).”

Reviewer #2 (Recommendations for the authors):Before I start critiquing individual components of the study, I want to say I am quite excited about these results, and believe this a very clever design and interesting neuroimaging findings.1. I found the introduction quite nice, and it really provided a strong foundation for a very timely paper. However, I think more specificity is needed for the predictions based on anatomical targets. For example, the ventral and dorsal lateral PFC are thought to sub-serve quite discrete processes, both of which could be relevant for this study. I think discussing prior work, and the authors own work, with a greater deal of anatomical specificity would help readers better interpret that findings. A similar weakness was in the lack of discussion of where in the prior and work value-related differences emerged. For example there is a lot of discussion of prior work by Davidow and Shohamy, but those all occurred in ventral striatum which does not overlap with the current findings.

We agree with the reviewer that our introduction and discussion would benefit from more anatomical granularity, and we did indeed have a priori predictions about more specific neural regions that might be involved in our task.

First, we expected that both the ventral and dorsal striatum might be responsive to stimulus value across our age range. Prior work has suggested that activity in the ventral striatum often correlates with the intrinsic value of a stimulus, whereas activity in the dorsal striatum may reflect goal-directed action values (Liljeholm and O’Doherty, 2012). In our task, we expected that high-frequency items may acquire intrinsic value during frequency learning that is then reflected in the striatal response to these items during encoding. However, because participants were not rewarded when they encountered these images, but rather incentivized to encode associations involving them, we hypothesized that the dorsal striatum may represent the value of the ‘action’ of remembering each pair. In line with this prediction, the dorsal striatum, and the caudate in particular, have also been shown to be engaged during value-guided cognitive control (Hikosaka et al., 2014; Insel et al., 2017).

We have now revised our introduction to include greater specificity in our anatomical predictions on p. 3:

“When individuals need to remember information associated with previously encountered stimuli (e.g., the grocery store aisle where an ingredient is located), frequency knowledge may be instantiated as value signals, engaging regions along the mesolimbic dopamine pathway that have been implicated in reward anticipation and the encoding of stimulus and action values. These areas include the ventral tegmental area (VTA) and the ventral and dorsal striatum (Adcock et al., 2006; Liljeholm and O’Doherty, 2012; Shigemune et al., 2014).”

Though we initially predicted that encoding of high-value information would be associated with increased activation in both the ventral and dorsal striatum, the activation we observed was largely within the dorsal striatum, and specifically, the caudate. We have now revised our discussion accordingly on p. 26:

“Though we initially hypothesized that both the ventral and dorsal striatum may be involved in encoding of high-value information, the activation we observed was largely within the dorsal striatum, a region that may reflect the value of goal-directed actions (Liljeholm and O’Doherty, 2012). In our task, rather than each stimulus acquiring intrinsic value during frequency learning, participants may have represented the value of the ‘action’ of remembering each pair during encoding.”

Second, while the ventromedial PFC often reflects value, given the control demands of our task, we expected to see greater activity in the dorsolateral PFC, which is often engaged in tasks that require the implementation of cognitive control (Botvinick and Braver, 2015). Thus, we hypothesized that individuals would show increased activation in the dlPFC during encoding of high- vs. low-value information, and that this activation would vary as a function of age. We have now clarified this hypothesis on p. 3:

“Value responses in the striatum may signal the need for increased engagement of the dorsolateral prefrontal cortex (dlPFC) (Botvinick and Braver, 2015), which supports the implementation of strategic control.”

In our discussion, we review disparate findings in the developmental literature and discuss factors that may contribute to these differences across studies. For example, in our discussion of Davidow et al. (2016), we highlight differences between their task design and the present study, focusing on how their task involved immediate receipt of reward at the time of encoding, while our task incentivized memory accuracy. We further note that studies that involve reward delivery at the time of encoding may engage different neural pathways than those that promote goal-directed encoding. Beyond Davidow et al., (2016), there are no other neuroimaging studies that examine the influence of reward on memory across development. Thus, we cannot relate our present neural findings to prior work on the development of value-guided memory. As we note in our discussion (p. 28), “Further work is needed to characterize both the influence of different types of reward signals on memory across development, as well as the development of the neural pathways that underlie age-related change in behavior.“

2. For the frequency learning task, it would be helpful to report on accuracy and RT for the 5th trial only. It would be extremely important to know if children and adults were entering the second phase of the experiment with the same acquired knowledge, or different amounts of acquired knowledge. If this 5th trial accuracy is different across age groups, the authors need to include this measure as a co-variate in all neural analyses, as age differences in biases towards high frequency information may result from not having learned the information rather than not being able to effectively use the information to guide adaptive encoding.

We continued to observe age differences in frequency-learning accuracy and reaction times (RTs) on the fifth and final presentation of each image (see newly added Figure S2). While age differences in RTs may be reflective of processing or motor response speed, we agree with the reviewer that age differences in accuracy likely indicate that children and adults did not necessarily acquire equivalent knowledge of environmental structure prior to encoding associated information.

Indeed, in our original manuscript, we included a model with participants’ frequency *reports* as a predictor of memory performance rather than each item’s true frequency. Not only did participant frequency reports predict associative memory, model comparison revealed that they were actually a better predictor of associative memory than each item’s true frequency. Importantly, we observed a frequency report x age interaction effect, indicating that older participants’ demonstrated greater modulation of memory by their *beliefs* about the structure of the environment. In other words, even when we control for age differences in beliefs about the structure of the environment, older participants continued to demonstrate greater *use* of beliefs about environmental structure to guide memory. Thus, taken together with our frequency-learning results, we believe results from this model demonstrate that there are age differences both in *learning* the structure of the environment, and in *using* learned regularities to guide encoding. Critically, age differences in *using* these learned regularities emerge even when controlling for age differences in learning. We have now clarified and expanded our interpretation of this model in our manuscript discussion on p. 23:

“Importantly, though we observed age-related differences in participants’ learning of the structure of their environment, the strengthening of the relation between frequency report and associative memory with increasing age suggests that age differences in learning cannot fully account for age differences in value guided memory. Even when accounting for individual differences in participants’ explicit knowledge of the structure of the environment, older participants demonstrated a stronger relation between their beliefs about item frequency and associative memory, suggesting that they used their beliefs to guide memory to a greater degree than younger participants.”

Of course, this model only controls for participants *explicit* beliefs about the structure of the environment. As the reviewer notes, there may be age differences in online frequency learning that also influence encoding. Further, our neural analyses did not account for age differences in explicit frequency reports. We have now run additional control analyses to account for the potential influence of individual differences in frequency learning on associative memory. Specifically, for each participant, we computed three metrics: (1) their overall accuracy during frequency learning, (2) their overall accuracy for the last presentation of each item during frequency learning (as suggested), and (3) the mean magnitude of the error in their frequency reports. We then included these metrics as covariates in our memory analyses.

When we include these control variables in our model, we continue to observe a robust effect of frequency condition (*p* < 0.001) as well as robust interactions between frequency condition and linear and quadratic age (*p*s < 0.003). We also observed a main effect of frequency error magnitude on memory accuracy (*p* < 0.001). Here, however, we no longer observe main effects of age or quadratic age on overall memory accuracy. Given the relation we observed between frequency error magnitudes and age, the results from this model suggests that there may be age-related improvements in overall memory that influence *both* memory for associations as well as learning of and memory for item frequencies. The fact that age no longer relates to overall memory when controlling for frequency error magnitudes suggest that age-related variance in memory for item frequencies and memory for associations are strongly related within individuals. Importantly, however, age-related variance in memory for item frequencies *did not* explain age-related variance in the influence of frequency condition on associative memory, suggesting that there are developmental differences in the *use* of knowledge of environmental structure to prioritize valuable information in memory that persist even when controlling for age-related differences in initial *learning* of environmental regularities. Given the importance of this analysis in elucidating the relation between the learning of environmental structure and value-guided memory, we have now updated the results in the main text of our manuscript to include them. On p. 13, we now write:

“Because we observed age-related differences in participants’ online learning of item frequencies and in their explicit frequency reports, we further examined whether these age-differences in initial learning could account for the age differences we observed in associative memory. To do so, we ran an additional model in which we included each participant’s mean frequency learning accuracy, mean frequency learning accuracy on the last repetition of each item, and explicit report error magnitude as covariates. Here, explicit report error magnitude strongly predicted overall memory performance, χ2(1) = 13.05, p < 0.001, and we did not observe main effects of age or quadratic age on memory performance (ps > 0.20). However, we continued to observe a main effect of frequency condition, χ2(1) = 19.65 *p* < 0.001, as well as significant interactions between frequency condition and both linear age χ2(1) = 10.59, *p* = 0.001, and quadratic age χ2(1) = 9.15, *p* = 0.002. Thus, while age differences in initial learning related to overall memory performance, they did not account for age differences in the use of environmental regularities to strategically prioritize memory for valuable information.”

In addition, as suggested by the reviewer, we also included the three covariates as control variables in our mediation. When controlling for online frequency learning and explicit frequency report errors, PFC activity continues to mediate the relation between age and memory difference scores. We have now included these results on p. 16 – 17 of the main text:

“Further, when we included quadratic age, WASI scores, online frequency learning accuracy, online frequency learning accuracy on the final repetition of each item, and mean explicit frequency report error magnitudes as control variables in the mediation analysis, PFC activation continued to mediate the relation between linear age and memory difference scores (standardized indirect effect:.56, 95% confidence interval: [.06, 1.31], *p* = 0.032; standardized direct effect; 1.75, 95% confidence interval: [.11,.3.52], p = 0.030).”

We also refer to these analyses when we interpret our findings in our discussion. Specifically, on p. 23, we write:

“In addition, we continued to observe a robust interaction between age and frequency condition on associative memory, even when controlling for age-related change in the accuracy of both online frequency learning and explicit frequency reports. Thus, though we observed age differences in the learning of environmental regularities and in their influence on subsequent associative memory encoding, our developmental memory effects cannot be fully explained by differences in initial learning.”

We thank the reviewer for this suggestion, as we believe these control analyses strengthen our interpretation of age differences in both the learning and use of environmental regularities to prioritize memory.

3. I think more discussion is warranted on why some neurobehavioral targets show quadratic effects while others show linear. Quadratic effects are often predicted in theoretical models, but from my read on the literature rarely show up in developmental analyses. These data could be leveraged to better understand those models by explaining why some processes are quadratic and others are linear.

We agree with the reviewer that more discussion is warranted here. While many cognitive processes tend to improve with increasing age, the significant interaction between quadratic age and frequency condition on memory accuracy could reflect a number of different patterns of developmental variance. Because quadratic curves are U-shaped, the significant interaction between quadratic age and frequency condition could reflect a peak in value-guided memory in adolescence. However, the combination of linear and quadratic effects can also capture “plateauing” effects, where the influence of age on a particular cognitive process decreases at a particular developmental timepoint. To determine how to interpret the quadratic effect of age on value-guided memory — and specifically, to test for the presence of an adolescent peak — we ran an additional analysis.

To test for an adolescent peak in value-guided memory, we first fit our memory accuracy model *without* any age terms, and then extracted the random slope across frequency conditions for each subject. We then conducted a ‘two lines test’ (Simonsohn, 2018) to examine the relation between age and these random slopes. In brief, the two-lines test fits the data with two linear models — one with a positive slope and one with a negative slope, algorithmically determining the breakpoint in the estimates where the signs of the slopes change. When we analyzed our memory data in this way, we found a robust, positive relation between age and value-guided memory (see newly added Appendix 2 – Figure 3) from childhood to mid-adolescence, that peaked around age 16 (age 15.86). From age ~16 to early adulthood, however, we observed only a marginal negative relation between age and value-guided memory (*p* = 0.0567). Thus, our findings do not offer strong evidence in support of an adolescent peak in value-guided memory — instead, they suggest that improvements in value-guided memory are strongest from childhood to adolescence.

To more clearly demonstrate the relation between age and value-guided memory, we have now included the results of the two-lines test in the Results section of our main text. On p. 12 – 13, we write:

“In line with our hypothesis, we observed a main effect of frequency condition on memory, χ2(1) = 21.51, *p* < 0.001, indicating that individuals used naturalistic value signals to prioritize memory for high-value information. […] Thus, the interaction between frequency condition and quadratic age on memory performance suggests that the biggest age differences in value-guided memory occurred through childhood and early adolescence, with older adolescents and adults performing similarly.”

That said, this developmental trajectory is likely specific to the particular demands of our task. In our previous behavioral study that used a very similar paradigm (Nussenbaum, Prentis, and Hartley, 2020), we observed only a linear relation between age and value-guided memory. Although the task used in our behavioral study was largely similar to the task we employed here, there were subtle differences in the design that may have extended the age range through which we observed improvements in memory prioritization. In particular, in our previous behavioral study, the memory test required participants to select the correct associate from a grid of 20 options (i.e., 1 correct and 19 incorrect options), whereas here, participants had to select the correct associate from a grid of 4 options (1 correct and 3 incorrect options). In our prior work, the need to differentiate the ‘correct’ option from many more foils may have increased the demands on either (or both) memory encoding or memory retrieval, requiring participants to encode and retrieve more specific representations that would be less confusable with other memory representations. By decreasing the task demands in the present study, we may have shifted the developmental curve we observed toward earlier developmental timepoints.

We originally did not emphasize our quadratic findings in the discussion of our manuscript because, given the marginal decrease in memory selectivity we observed from age 16 to age 25 and the different age-related findings across our two studies, we did not want to make strong claims about the specific shape of developmental change. However, we agree with the reviewer that these points are worthy of discussion within the manuscript. We have now amended our discussion on p. 25 accordingly:

“We found that memory prioritization varied with quadratic age, and our follow-up tests probing the quadratic age effect did not reveal evidence for significant age-related change in memory prioritization between late adolescence and early adulthood. However, in our prior behavioral work using a very similar paradigm (Nussenbaum et al., 2020), we found that memory prioritization varied with linear age only. In line with theoretical proposals (Davidow et al., 2018), subtle differences in the control demands between the two tasks (e.g., reducing the number of ‘foils’ presented on each trial of the memory test here relative to our prior study), may have shifted the age range across which we observed differences in behavior, with the more demanding variant of our task showing more linear age-related improvements into early adulthood. In addition, the specific control demands of our task may have also influenced the age at which value-guided memory emerged. Future studies should test whether younger children can modulate encoding based on the value of information if the mnemonic demands of the task are simpler. ”

We thank the reviewer for this helpful suggestion, and believe our additions that expand on the quadratic age effects help clarify our developmental findings.

4. The authors show directly compare the model fits of the mediation models that manipulate directionality.

As noted in the manuscript (p. 16), we found that PFC activity during encoding of pairs involving high- vs. low-frequency items mediated the relation between age and memory differences scores (standardized indirect effect:.07, 95% confidence interval: [.01,.15], *p* = 0.017; standardized direct effect:.15, 95% confidence interval: [-.03,.33], *p* = 0.108), but age did *not* mediate the relation between PFC activity and memory difference scores (standardized indirect effect:.03, 95% confidence interval: [-.007,.09], *p* = 0.13; standardized direct effect;.34, 95% confidence interval: [.14,.54], *p* < 0.001.). Directly comparing the fits of the mediation models does not seem possible — In both cases, the full models include the same dependent variable (memory difference scores) and the same two predictor variables (PFC activation and age). Thus, both models will fit the data equivalently well.

However, to further elucidate the directionality of the effects we observed, we examined the AIC difference between the linear regression testing the main effect of the predictor and the linear regression that included the hypothesized mediator to determine the extent to which the mediator improved model fit. Specifically, to examine the extent to which including PFC activation in the model improved the fit of our regression examining the relation between age and memory difference scores, we computed the model AICs with and without PFC activation included as a predictor.

In this case, adding each subject’s PFC activation reduced the AIC by 9.07 and significantly improved model fit (*F*(1) = 11.39, *p* = 0.001). Including age as a predictor in the regression examining the relation between PFC activation and memory difference scores only reduced the model AIC by.22 and did not significantly improve model fit (*F*(1) = 2.17, *p* = 0.14). Thus, we believe these additional quantitative metrics provide further evidence for the direction of the mediation. These results capture information that is redundant with the mediation analyses included in the manuscript. As such we have chosen not to add them.

5. I found the predictors from the frequency analysis predicting behavior during memory encoding/retrieval to perhaps be the most interesting finding in the paper, especially given that both implicit and explicit measures were predicting memory independently. However, then I was left wanting to know (somewhat desperately!), how much these signals related to the lateral PFC and caudate signals seen during memory encoding. I think this type of analysis would really help make the paper a complete package.

Thank you for this valuable suggestion. We agree that it would be interesting to link frequency-learning behavior to neural activity at encoding. As such, we have now conducted additional analyses to explore these relations.

In the original version of our manuscript, we examined behavior at the *item* level through mixed-effects models, and neural activation during encoding at the *participant* level. Thus, to examine the relation between frequency-learning metrics and neural activation at encoding, we created two additional participant-level metrics. For each participant we computed their average *repetition suppression index*, and a measure of *frequency distance*. The average repetition suppression index reflects the overall extent to which the participant demonstrated repetition suppression in response to the fifth presentation of the high-frequency items, and is computed by averaging each participant’s repetition suppression indices across items. We hypothesized that participants who demonstrated the greatest degree of repetition suppression might be the most sensitive to the difference between the 1- and 5-frequency items, and therefore, show the greatest differences in striatal and PFC activation during encoding of high- vs. low-value information. The *frequency distance* metric reflects the average distance between participants’ explicit frequency reports for items that appeared once and items that appeared five times, and is computed by averaging their explicit frequency reports for items in each frequency condition, and then subtracting the average reports in the low-frequency condition from those in the high-frequency condition. We hypothesized that participants with the largest frequency distances might similarly be the most sensitive to the difference between the 1- and 5-frequency items, and therefore, show the greatest differences in striatal and PFC activation during encoding of high- vs. low-value information.

We first wanted to confirm that the relations we observed between repetition suppression, frequency reports, and age, could also be observed at the participant level. In line with our prior, behavioral analyses, we found that age related to both mean repetition suppression indices (marginally; linear age: *p* = 0.067; quadratic age: *p* = 0.042); and frequency distances (linear and quadratic age: *p*s < 0.001).

In addition, we further tested whether these two metrics related to memory performance. In contrast to our item-level findings, we did not observe a significant relation between repetition suppression indices and memory (*p* = 0.83). We did observe an effect of frequency distance on memory performance. Specifically, we observed significant interactions between frequency distance and age (*p* = 0.014) and frequency distance and quadratic age (*p* = 0.021) on memory difference scores, such that the influence of frequency distance on memory difference scores increased with increasing age from childhood to adolescence.

We next examined how mean repetition suppression indices and frequency distances related to differential neural activation during encoding of high- and low-value pairs. In line with our memory findings, we did not observe any significant relations between mean repetition suppression indices and neural activation in the caudate or prefrontal cortex during encoding (*p*s > 0.15).

Frequency distance did not relate to caudate activation during encoding nor did we observe a frequency distance x age interaction effect (*p*s > 0.16). Frequency distance did, however, relate to differential PFC activation during encoding of high- vs. low-value pairs. Specifically, we observed a main effect of frequency distance on PFC activation (*p* = 0.0012), such that participants whose explicit reports of item frequency, were on average, more distinct across frequency conditions, demonstrated increased PFC activation during encoding of pairs involving high- vs. low-frequency items. Interestingly, when we included frequency distance in our model, we no longer observed a significant effect of age on differential PFC activation, nor did we observe a significant frequency distance x age interaction (*p*s > 0.13). These findings suggest that PFC activation during encoding may have, in part, reflected participants’ beliefs about the structure of the environment, with participants demonstrating stronger differential engagement of control processes across conditions when their representations of the conditions themselves were more distinct.

Finally, we examined how age, frequency distance, and PFC activation related to memory difference scores. Here, even when controlling for both frequency distance and PFC activation, we continued to observe main effects of age and quadratic age on memory difference scores (linear age: *p* = 0.006; quadratic age: *p* = 0.001). In line with our analysis of the relation between frequency reports and memory, these results suggest that age-related variance in value-guided memory may depend on both knowledge of the structure of the environment and use of that knowledge to effectively control encoding.

We have now added these results to our manuscript on p. 13 – 14. We write:

“Given the relations we observed between memory and both repetition suppression and frequency reports, we examined whether they related to neural activation in both our caudate and PFC ROI during encoding. […] Here, we did not observe a significant effect of age on PFC activation (β = -.03, SE = 0.13, p = 0.82), suggesting that age-related variance in PFC activation may be related to age differences in explicit frequency beliefs. Importantly, however, even when we accounted for both PFC activation and frequency distances, we continued to observe an effect of age on memory difference scores (β = 0.56, SE = 0.20, p = 0.006), which, together with our prior analyses, suggest that developmental differences in value-guided memory are not driven solely by age differences in beliefs about the structure of the environment but also depend on the use of those beliefs to guide encoding.”

We have added the full model results to Appendix 3.

Given these results, we have now revised our interpretation of our neural data. Our memory analyses demonstrate that across our age range, we observed age-related differences in *both* the acquisition of knowledge of the structure of the environment and in its *use*. Originally, we interpreted the PFC activation as reflecting the use of learned value to guide memory. However, the strong relation we found between frequency distance and PFC activation suggests that the age differences in PFC activation that we observed may also be related to age differences in knowledge of the structure of the environment that governs when control processes should be engaged most strongly. However, these results must be interpreted cautiously. Participants provided explicit frequency reports *after* they completed the encoding and retrieval tasks, and so explicit frequency reports may have been influenced not only by participants’ memories of online frequency learning, but also by the strength with which they encoded the item and its paired associate, and the experience of successfully retrieving it.

We have now revised our discussion to consider these results. On p. 23, we now write,

“Our neural results further suggest that developmental differences in memory were driven by both knowledge of the structure of the environment and *use* of that knowledge to guide encoding.”

On p. 24, we write,

“The development of adaptive memory requires not only the implementation of encoding and retrieval strategies, but also the flexibility to up- or down-regulate the engagement of control in response to momentary fluctuations in information value (Castel et al., 2007, 2013; Hennessee et al., 2017). Importantly, value-based modulation of lateral PFC engagement during encoding mediated the relation between age and memory selectivity, suggesting that developmental change in both the representation of learned value and value-guided cognitive control may underpin the emergence of adaptive memory prioritization. Prior work examining other neurocognitive processes, including response inhibition (Insel et al., 2017) and selective attention (Störmer et al., 2014), has similarly found that increases in the flexible upregulation of control in response to value cues enhance goal-directed behavior across development (Davidow et al., 2018), and may depend on the engagement of both striatal and prefrontal circuitry (Hallquist et al., 2018; Insel et al., 2017). Here, we extend these past findings to the domain of memory, demonstrating that value signals derived from the structure of the environment increasingly elicit prefrontal cortex engagement and strengthen goal-directed encoding across childhood and into adolescence.”

And on p. 25, we have added an additional paragraph:

“Further, we also demonstrate that in the absence of explicit value cues, the engagement of prefrontal control processes may reflect beliefs about information value that are learned through experience. Here, we found that differential PFC activation during encoding of high- vs. low-value information reflected individual and age-related differences in beliefs about the structure of the environment; participants who represented the average frequencies of the low- and high-frequency items as further apart also demonstrated greater value-based modulation of lateral PFC activation. It is important to note, however, that we collected explicit frequency reports after associative encoding and retrieval. Thus the relation between PFC activation and explicit frequency reports may be bidirectional — while participants may have increased the recruitment of cognitive control processes to better encode information they believed was more valuable, the engagement of more elaborative or deeper encoding strategies that led to stronger memory traces may have also increased participants’ subjective sense of an item’s frequency (Jonides and Naveh-Benjamin, 1987).”

6. Regarding the discussion, I think it would be helpful for the authors to discuss a few features of their data and how they relate to development. The first would be the WASI findings which were quite prominent in most analyses, and in a few showed interactions with age. If this measure is a proxy of executive function, discussing the role of executive function for adaptive memory could help provide a more concrete mechanisms of adaptive memory formation across development.

Thank you for this suggestion. The main focus of our study was to examine age-related change in the influence of learned value signals on memory. Because this study was cross-sectional, one concern was that the children, adolescents, and adults that we recruited may have come from different populations. For example, since we conduct recruitment events at local science fairs and tell participants that they will receive a picture of their brain if they come in, one concern was that the children who signed up to participate may have been particularly excited about science and research, and therefore fundamentally different from the adults we recruited, who may have been more motivated by the monetary compensation. Indeed, we did observe a negative relation between age and age-normed IQ in our sample, suggesting the children had slightly higher IQs for their age relative to adults. Thus, we included WASI scores in all of our analyses to account for these age-related differences in IQ. Our aim in including IQ as a control variable was to partially account for confounding, population-level differences across our age groups, enabling us to more clearly examine the relation between age itself and our neurocognitive processes of interest. We did not intend to examine the role of IQ in motivated memory processes.

We have now clarified why we collected WASI data and used the scores in our analyses in our methods section (p. 7):

“Because this study was cross-sectional, one concern was that the children, adolescents, and adults that we recruited may have come from different populations. Indeed, we observed a significant relation between age and age-normed Wechsler Abbreviated Scale of Intelligence (WASI; Wechsler, 2011) scores in our sample (β = -.60, SE = 0.26, *p* = 0.0238), suggesting the children had slightly higher estimated IQs for their age relative to adults. To account for these age-related differences in reasoning ability, we included age-normed WASI scores as an interacting fixed effect in all analyses. Our aim in including WASI scores as a control variable was to partially account for confounding, population-level differences across our age groups, enabling us to more clearly examine the relation between age itself and our neurocognitive processes of interest.”

That said, as the reviewer notes, we did observe IQ effects across our analyses. However, in our main memory analysis, we observed a main effect of age-normed WASI IQ score on memory accuracy, but no interactions between WASI scores and frequency condition. This pattern of results suggests that individuals with higher IQs were better at forming associations in memory, but not necessarily better at selectively encoding high-value associations. Similarly, we found that individuals with higher WASI scores were also better at learning the structure of the environment, as demonstrated by a main effect of WASI score on frequency report error magnitudes, but the influence of WASI score on frequency report error magnitudes did not vary as a function of age. Taken together, our results are highly consistent with prior studies that have suggested that IQ may relate to learning and memory across development (Deary et al., 2010; Rose et al., 2012).

The reason we have not included a discussion of WASI scores in the manuscript is because our primary goal was to elucidate the relation between age and motivated memory processes. Across our analyses, we observed few interactions between WASI scores and age. We did observe significant interactions between age and WASI scores on differential neural activation within both the caudate and hippocampus during encoding of high- vs. low-value information. We do not have a strong interpretation for these results, so we report them but do not discuss them.

We agree with the reviewer that it would be valuable to identify more concrete mechanisms that drive developmental change in the component processes of adaptive memory. However, as we now clarify in our methods, the purpose of including the WASI was to account for age-related differences in reasoning ability that may confound our interpretation of our age effects. Thus, we do not want to overinterpret our WASI findings.

Additionally, it would be helpful for the authors to discuss the negative findings during retrieval. The fact that these age-related differences in adaptive memory processes are most likely seeming from encoding versus retrieval is not only highly interesting, but are a major prediction that stems from an animal literature on dopaminergic influences on hippocampal-dependent memory.

Most of our hypotheses about age-related change in value-guided memory were about changes in encoding processes. In our whole-brain analyses, we found that neural activation at encoding related to memory difference scores. Neural activation at retrieval, however, did not vary by memory difference scores or by age. Given these findings, we focused our subsequent analyses on encoding.

As the reviewer notes, the animal literature makes interesting theoretical predictions about the role of dopamine in value-guided memory encoding processes. Because we don’t have data that can tie our findings to dopaminergic mechanisms, we refrain from too much speculation. However, we have now revised our discussion to emphasize the specificity of our findings to the encoding phase of the task. We now discuss our retrieval findings in more depth on p. 25:

“During retrieval, we continued to observe increased activation of the caudate and dlPFC for high- vs. low-value pairs. However, this activation did not significantly vary as a function of memory difference scores or age, suggesting that the developmental differences in value-guided memory that we observed were likely driven by age-related change in encoding processes.”

7. It would be helpful to report all behavioral results in the youngest sample only, as the interpretation of the data is quite different if children can or cannot perform these processes.

Thank you for this suggestion. Determining whether the “youngest” participants can or cannot use learned value to influence memory is complicated for several reasons. First, drawing a line to determine who to include in our analysis of the ‘youngest’ participants feels fairly arbitrary — our youngest children included were age 8, but we only have 5 8-year-olds in our sample, and so we would not expect to observe any significant influence of frequency condition within such a small sample. We can include more participants, but by doing so, we then need to add older participants (e.g., 9- and 10-year-olds) to this analysis, potentially muddling what conclusions can be drawn. Thus, we do not believe this question is best answered by statistics — instead, we have chosen to display individual points for each participant that depict their memory performance in Figure 4 on p. 12. Per Reviewer 3’s suggestion, we have also now connected the points belonging to the same participants to more clearly display individual participants’ performance. We believe this plot can give readers the best sense of how participants across our age range performed on the task.

Perhaps more importantly, in line with theoretical accounts of the development of value-guided cognitive control (Davidow, Insel, and Somerville, 2018), we also believe that the specific age at which the ability to use value to guide memory encoding emerges will be highly dependent on the nature of the control demands elicited by the specific task used. For example, younger children may have demonstrated stronger effects of value on memory if they only had to encode two associations (instead of 24), if the stimuli themselves were easier to represent in memory (e.g., child-friendly, nameable objects rather than more abstract images and scenes), or if the value differences were greater (e.g., 1 vs. 10 repetitions instead of 1 vs. 5). Thus, rather than making a claim about the specific age at which motivated memory emerges, we have instead expanded our discussion of this issue on p. 25 – 26 of the manuscript:

“We found that memory prioritization varied with quadratic age, and our follow-up tests probing the quadratic age effect did not reveal evidence for significant age-related change in memory prioritization between late adolescence and early adulthood. However, in our prior behavioral work using a very similar paradigm (Nussenbaum et al., 2020), we found that memory prioritization varied with linear age only. In line with theoretical proposals (Davidow et al., 2018), subtle differences in the control demands between the two tasks (e.g., reducing the number of ‘foils’ presented on each trial of the memory test here relative to our prior study), may have shifted the age range across which we observed differences in behavior, with the more demanding variant of our task showing more linear age-related improvements into early adulthood. In addition, the specific control demands of our task may have also influenced the age at which value-guided memory emerged. Future studies should test whether younger children can modulate encoding based on the value of information if the mnemonic demands of the task are simpler.”

8. Could the authors provide data on differences in behavioral performance for both task content (which could drive motivational differences) or task order (which might lead to practice effects). If notable differences emerge across these factors, I strongly believe they needed to be included as co-variates in ally analyses.

Thank you for raising this important point. During the review process, we also received feedback on our preprint from a colleague who suggested that in addition to including participant random effects in our models, we should also include stimulus random effects (Barr et al., 2013). As noted by the reviewer, task content may influence motivation or memorability — the stimulus-level random effects should account for variance in memory performance that may be driven by differences in how participants encode individual stimuli, beyond any effects of our frequency manipulation. Thus, in our revised manuscript, we have updated our modeling approach accordingly. On p. 33 of our methods section, we now write:

“To determine the random effects structures of our mixed effects models, we began with the maximal model to minimize Type I errors (Barr et al., 2013). We included random participant intercepts and slopes across all fixed effects (except age and IQ) and their interactions. We also included random stimulus intercepts and slopes across all fixed effects and their interactions. Because stimuli were randomly paired during associative encoding and only repeated, on average, around 4 times across participants, our stimulus random effects accounted for individual items (e.g., postcard 1) rather than pairs of items (e.g., postcard 1 and stamp 5). We set the number of model iterations to one million and use the “bobyqa” optimizer. When the maximal model gave convergence errors or failed to converge within a reasonable timeframe (~24 hours), we removed correlations between random slopes and random intercepts, followed by random slopes for interaction effects, followed by random slopes across stimuli. For full details about the fixed- and random-effects structure of all models, see Appendix 3: Full Model Specification and Results.”

Importantly, all of the effects that we reported in the original version of our manuscript hold when accounting for stimulus-level random effects, indicating that our results are robust to individual differences across stimuli.

As the reviewer suggested, we also ran two additional models to test the influence of block order and block type (postcards and stamps vs. pictures and picture frames) on associative memory. We did not observe any significant main effects of or interactions with block order (main effect: χ2(1) = 0.18, *p* = 0.675, all interaction *p* values > 0.20), but we did observe a significant block type x frequency condition interaction effect (*p* = 0.036). This seems to be driven by better memory performance for associations involving low-frequency *pictures* relative to low-frequency *postcards* (see newly added Appendix 2 – Figure 5). Importantly, all the other effects in our model (e.g., frequency condition, age x frequency condition, etc.) hold when we account for block type.

A priori, we planned to collapse data across our two blocks for all analyses; we split our experiment into two blocks to make frequency learning and associative memory encoding easier for participants while still ensuring we had enough trials for our neural analyses to be adequately powered — our experiment was not designed to examine the effects of block type or order. However, given that we observed a frequency condition x block type interaction on associative memory, we also examined each participant’s average β weight (parameter estimate) within our PFC ROI for each block separately. We used fslmeants to extract these parameter estimates, and then examined their relations with both age and memory difference scores, which we also computed separately for each block.

Briefly, when we included block type as a covariate in our analyses, we continued to observe significant effects of age and quadratic age on differential PFC activation across frequency conditions during encoding. Further, we continued to observe a relation between PFC activation and memory difference scores, suggesting that differential engagement of the PFC during encoding of high- vs. low-value associations may, in part, account for individual and developmental differences in value-guided memory.

Thus, while we did observe differences in behavior that related to task content, our main conclusions hold when controlling for them. We have now included these additional analyses of task content in Appendix 2: Supplementary Analyses (Because they span four pages, we have not included them here).

We refer to these analyses in our methods, on p. 33:

“Data were combined across blocks (but we include an analysis of block effects on memory performance in Appendix 2: Supplementary Analyses).”

We thank the reviewer for prompting us to more fully examine different influences on memory performance in our task, and believe that these supplementary results provide greater evidence for our central argument.

Reviewer #3 (Recommendations for the authors):1) Empirical findings directly comparing cross-sectional and longitudinal effects have demonstrated that cross-sectional analyses of age differences do not readily generalize to longitudinal research (e.g., Raz et al., 2005; Raz and Lindenberger, 2012). Formal analyses have demonstrated that proportion of explained age-related variance in cross-sectional mediation models may stem from various factors, including similar mean age trends, within-time correlations between a mediator and an outcome, or both (Lindenberger et al., 2011; see also Hofer, Flaherty, and Hoffman, 2006; Maxwell and Cole, 2007). Thus, the results of the mediation analysis showing that PFC activation explains age-related variance in memory difference scores, cannot be taken to imply that changes in PFC activation are correlated with changes in value-guided memory. While the general limitations of a cross-sectional study are noted in the Discussion of the manuscript, it would be important to discuss the critical limitations of the mediation analysis. While the main conclusions of the paper do not critically depend on this analysis, it would be important to alert the reader to the limited information value in performing cross-sectional mediation analyses of age variance.

Thank you for raising this critical point. We have expanded our discussion to specifically note the limitations of our mediation analysis and to more strongly emphasize the need for future longitudinal studies to reveal how changes in neural circuitry may support the emergence of motivated memory across development. Specifically, on p. 26, we now write:

“One important caveat is that our study was cross-sectional — it will be important to replicate our findings in a longitudinal sample to more directly measure how developmental changes in cognitive control within an individual contribute to changes in their ability to selectively encode useful information. Our mediation results, in particular, must be interpreted with caution as simulations have demonstrated that in cross-sectional samples, variables can emerge as significant mediators of age-related change due largely to statistical artifact (Hofer, Flaherty, and Hoffman, 2006; Lindenberger et al., 2011). Indeed, our finding that PFC activation mediates the relation between age and value-guided memory does not necessarily imply that within an individual, PFC development leads to improvements in memory selectivity. Longitudinal work in which individuals’ neural activity and memory performance is sampled densely within developmental windows of interest is needed to elucidate the complex relations between age, brain development, and behavior (Hofer, Flaherty, and Hoffman, 2006; Lindenberger et al., 2011).”

2) It would be helpful to provide more information on how chance memory performance was handled during data analysis, especially as it is more likely to occur in younger participants. Related to this, please connect the points that belong to the same individual in Figure 3 to facilitate evaluation of individual differences in the memory difference scores.

Thank you for raising this important point. On each memory test trial, participants viewed the item (either a postcard or picture) above images of four possible paired associates (see Figure 1 on p. 6). On each memory test trial, participants had 6 seconds to select one of these items. If participants did not make a response within 6 seconds, that trial was considered ‘missed.’ Missed trials were excluded from behavioral analyses and regressed out in neural analyses. If participants selected the correct associate, memory accuracy was coded as ‘1;’ if they selected an incorrect associate, accuracy was coded as ‘0.’ On each trial, there was 1 correct option and 3 incorrect options. As such, chance-level memory performance was 25%. We have now clarified this on p. 34 and included a dashed line indicating chance-level performance within Figure 4 (formerly Figure 3) on p. 12. In addition, we have also updated Figure 4 to connect the points belonging to the same participants, as suggested by the reviewer.

Out of 90 participants, 2 children performed at or below chance (<= 25% memory accuracy). Interpreting the behavior of the participants who responded to fewer than 12 out of 48 trials correctly is challenging. On the one hand, they might not have remembered anything and responded correctly on these trials due to randomly guessing. On the other hand, they may have implemented an encoding strategy of focusing only on a small number of pairs. Thus, a priori, based on the analysis approach we implemented in our prior, behavioral study (Nussenbaum et al., 2019), we decided to include *all* participants in our memory analyses, regardless of their overall accuracy. However, when we exclude these two participants from our memory analyses, our main findings still hold. Specifically, we continue to observe main effects of frequency condition and age, and interactions between frequency condition and both linear and quadratic age on associative memory accuracy (*p*s < 0.012).

We have now clarified these details about chance-level performance in the methods section of our manuscript on p. 34.

“For our memory analyses, trials were scored as ‘correct’ if the participant selected the correct association from the set of four possible options presented during the memory test, ‘incorrect’ if the participant selected an incorrect association, and ‘missed’ if the participant failed to respond within the 6-second response window. Missed trials were excluded from all analyses. Because participants had to select the correct association from four possible options, chance-level performance was 25%. Two child participants performed at or below chance-level on the memory test. They were included in all analyses reported in the manuscript; however, we report full details of the results of our memory analyses when we exclude these two participants in Appendix 3 (Table 15). Importantly, our main findings remain unchanged.”

In Appendix 3, we include a table with the full results from our memory model without these two participants.

3) I would like to see some consideration of how the different signatures of value learning, repetition suppression and reported item frequency, are related to the observed PFC and caudate effects during memory encoding. Such a discussion would help the reader connect the findings on learning and using information value across development.

Thank you for this valuable suggestion. We agree that it would be interesting to link frequency-learning behavior to neural activity at encoding. As such, we have now conducted additional analyses to explore these relations.

In the original version of our manuscript, we examined behavior at the *item* level through mixed-effects models, and neural activation during encoding at the *participant* level. Thus, to examine the relation between frequency-learning metrics and neural activation at encoding, we created two additional participant-level metrics. For each participant we computed their average *repetition suppression index*, and a measure of *frequency distance*. The average repetition suppression index reflects the overall extent to which the participant demonstrated repetition suppression in response to the fifth presentation of the high-frequency items, and is computed by averaging each participant’s repetition suppression indices across items. We hypothesized that participants who demonstrated the greatest degree of repetition suppression might be the most sensitive to the difference between the 1- and 5-frequency items, and therefore, show the greatest differences in striatal and PFC activation during encoding of high- vs. low-value information. The *frequency distance* metric reflects the average distance between participants’ explicit frequency reports for items that appeared once and items that appeared five times, and is computed by averaging their explicit frequency reports for items in each frequency condition, and then subtracting the average reports in the low-frequency condition from those in the high-frequency condition. We hypothesized that participants with the largest frequency distances might similarly be the most sensitive to the difference between the 1- and 5-frequency items, and therefore, show the greatest differences in striatal and PFC activation during encoding of high- vs. low-value information.

We first wanted to confirm that the relations we observed between repetition suppression, frequency reports, and age, could also be observed at the participant level. In line with our prior, behavioral analyses, we found that age related to both mean repetition suppression indices (marginally; linear age: *p* = 0.067; quadratic age: *p* = 0.042); and frequency distances (linear and quadratic age: *p*s < 0.001).

In addition, we further tested whether these two metrics related to memory performance. In contrast to our item-level findings, we did not observe a significant relation between repetition suppression indices and memory (*p* = 0.83). We did observe an effect of frequency distance on memory performance. Specifically, we observed significant interactions between frequency distance and age (*p* = 0.014) and frequency distance and quadratic age (*p* = 0.021) on memory difference scores, such that the influence of frequency distance on memory difference scores increased with increasing age from childhood to adolescence.

We next examined how mean repetition suppression indices and frequency distances related to differential neural activation during encoding of high- and low-value pairs. In line with our memory findings, we did not observe any significant relations between mean repetition suppression indices and neural activation in the caudate or prefrontal cortex during encoding (*p*s > 0.15).

Frequency distance did not relate to caudate activation during encoding nor did we observe a frequency distance x age interaction effect (*p*s > 0.16). Frequency distance did, however, relate to differential PFC activation during encoding of high- vs. low-value pairs. Specifically, we observed a main effect of frequency distance on PFC activation (*p* = 0.0012), such that participants whose explicit reports of item frequency, were on average, more distinct across frequency conditions, demonstrated increased PFC activation during encoding of pairs involving high- vs. low-frequency items. Interestingly, when we included frequency distance in our model, we no longer observed a significant effect of age on differential PFC activation, nor did we observe a significant frequency distance x age interaction (*p*s > 0.13). These findings suggest that PFC activation during encoding may have, in part, reflected participants’ beliefs about the structure of the environment, with participants demonstrating stronger differential engagement of control processes across conditions when their representations of the conditions themselves were more distinct.

Finally, we examined how age, frequency distance, and PFC activation related to memory difference scores. Here, even when controlling for both frequency distance and PFC activation, we continued to observe main effects of age and quadratic age on memory difference scores (linear age: *p* = 0.006; quadratic age: *p* = 0.001). In line with our analysis of the relation between frequency reports and memory, these results suggest that age-related variance in value-guided memory may depend on both knowledge of the structure of the environment and use of that knowledge to effectively control encoding.

We have now added these results to our manuscript on p. 13 – 14. We write:

“Given the relations we observed between memory and both repetition suppression and frequency reports, we examined whether they related to neural activation in both our caudate and PFC ROI during encoding. […] Importantly, however, even when we accounted for both PFC activation and frequency distances, we continued to observe an effect of age on memory difference scores (β = 0.56, SE = 0.20, p = 0.006), which, together with our prior analyses, suggest that developmental differences in value-guided memory are not driven solely by age differences in beliefs about the structure of the environment but also depend on the use of those beliefs to guide encoding.”

We have added the full model results to Appendix 3.

Given these results, we have now revised our interpretation of our neural data. Our memory analyses demonstrate that across our age range, we observed age-related differences in *both* the acquisition of knowledge of the structure of the environment and in its *use*. Originally, we interpreted the PFC activation as reflecting the use of learned value to guide memory. However, the strong relation we found between frequency distance and PFC activation suggests that the age differences in PFC activation that we observed may also be related to age differences in knowledge of the structure of the environment that governs when control processes should be engaged most strongly. However, these results must be interpreted cautiously. Participants provided explicit frequency reports *after* they completed the encoding and retrieval tasks, and so explicit frequency reports may have been influenced not only by participants’ memories of online frequency learning, but also by the strength with which they encoded the item and its paired associate, and the experience of successfully retrieving it.

We have now revised our discussion to consider these results. On p. 23, we now write,

“Our neural results further suggest that developmental differences in memory were driven by both knowledge of the structure of the environment and *use* of that knowledge to guide encoding.”

On p. 24, we write,

“The development of adaptive memory requires not only the implementation of encoding and retrieval strategies, but also the flexibility to up- or down-regulate the engagement of control in response to momentary fluctuations in information value (Castel et al., 2007, 2013; Hennessee et al., 2017). Importantly, value-based modulation of lateral PFC engagement during encoding mediated the relation between age and memory selectivity, suggesting that developmental change in both the representation of learned value and value-guided cognitive control may underpin the emergence of adaptive memory prioritization. Prior work examining other neurocognitive processes, including response inhibition (Insel et al., 2017) and selective attention (Störmer et al., 2014), has similarly found that increases in the flexible upregulation of control in response to value cues enhance goal-directed behavior across development (Davidow et al., 2018), and may depend on the engagement of both striatal and prefrontal circuitry (Hallquist et al., 2018; Insel et al., 2017). Here, we extend these past findings to the domain of memory, demonstrating that value signals derived from the structure of the environment increasingly elicit prefrontal cortex engagement and strengthen goal-directed encoding across childhood and into adolescence.”

And on p. 25, we have added an additional paragraph:

“Further, we also demonstrate that in the absence of explicit value cues, the engagement of prefrontal control processes may reflect beliefs about information value that are learned through experience. Here, we found that differential PFC activation during encoding of high- vs. low-value information reflected individual and age-related differences in beliefs about the structure of the environment; participants who represented the average frequencies of the low- and high-frequency items as further apart also demonstrated greater value-based modulation of lateral PFC activation. It is important to note, however, that we collected explicit frequency reports after associative encoding and retrieval. Thus the relation between PFC activation and explicit frequency reports may be bidirectional — while participants may have increased the recruitment of cognitive control processes to better encode information they believed was more valuable, the engagement of more elaborative or deeper encoding strategies that led to stronger memory traces may have also increased participants’ subjective sense of an item’s frequency (Jonides and Naveh-Benjamin, 1987).”

4) A point worthy of discussion are the implications of the finding that younger participants demonstrated greater deviations in their frequency reports for the development of value learning, given that frequency reports were found to predict associative memory accuracy.

Thank you for raising this important point. Indeed, one of our main findings is that older participants are better both at *learning* the structure of their environments and also at *using* structured knowledge to strategically prioritize memory. In our original manuscript, we described results of a model that included participants’ explicit frequency reports as a predictor of memory. Model comparison revealed that participants’ frequency reports — which we interpret as reflecting their beliefs about the structure of the environment — predicted memory more strongly than the item’s true frequency. In other words, participants’ beliefs about the structure of the environment (even if incorrect) more strongly influenced their memory encoding than the true structure of the environment. Critically, however, frequency reports interacted with age to predict memory (Figure 8). Even when we accounted for age-related differences in knowledge of the structure of the environment, older participants demonstrated a stronger influence of frequency on memory, suggesting they were better able to use their beliefs to control subsequent associative encoding. We have now clarified our interpretation of this model in our discussion on p. 23:

“Importantly, though we observed age-related differences in participants’ learning of the structure of their environment, the strengthening of the relation between frequency reports and associative memory with increasing age suggests that age differences in learning cannot fully account for age differences in value-guided memory. Even when accounting for individual differences in participants’ explicit knowledge of the structure of the environment, older participants demonstrated a stronger relation between their beliefs about item frequency and associative memory, suggesting that they used their beliefs to guide memory to a greater degree than younger participants.”

As noted by the reviewer, however, our initial memory analysis did not account for age-related differences in participants’ initial, online learning of item frequency, and our neural analyses further did not account for age differences in explicit frequency reports. We have now run additional control analyses to account for the potential influence of individual differences in frequency learning on associative memory. Specifically, for each participant, we computed three metrics: (1) their overall accuracy during frequency learning, (2) their overall accuracy for the last presentation of each item during frequency learning (as suggested by Reviewer 2), and (3) the mean magnitude of the error in their frequency reports. We then included these metrics as covariates in our memory analyses.

When we include these control variables in our model, we continue to observe a robust effect of frequency condition (*p* < 0.001) as well as robust interactions between frequency condition and linear and quadratic age (*p*s < 0.003) on associative memory accuracy. We also observed a main effect of frequency error magnitude on memory accuracy (*p* < 0.001). Here, however, we no longer observe main effects of age or quadratic age on overall memory accuracy. Given the relation we observed between frequency error magnitudes and age, the results from this model suggests that there may be age-related improvements in overall memory that influence *both* memory for associations as well as learning of and memory for item frequencies. The fact that age no longer relates to overall memory when controlling for frequency error magnitudes suggest that age-related variance in memory for item frequencies and memory for associations are strongly related within individuals. Importantly, however, age-related variance in memory for item frequencies *did not* explain age-related variance in the influence of frequency condition on associative memory, suggesting that there are developmental differences in the *use* of knowledge of environmental structure to prioritize valuable information in memory that persist even when controlling for age-related differences in initial *learning* of environmental regularities. Given the importance of this analysis in elucidating the relation between the learning of environmental structure and value-guided memory, we have now updated the results in the main text of our manuscript to include them. Specifically, on p. 13, we now write:

“Because we observed age-related differences in participants’ online learning of item frequencies and in their explicit frequency reports, we further examined whether these age differences in initial learning could account for the age differences we observed in associative memory. To do so, we ran an additional model in which we included each participant’s mean frequency learning accuracy, mean frequency learning accuracy on the last repetition of each item, and explicit report error magnitude as covariates. Here, explicit report error magnitude predicted overall memory performance, χ2(1) = 13.05, *p* < 0.001, and we did not observe main effects of age or quadratic age on memory performance (ps > 0.20). However, we continued to observe a main effect of frequency condition, χ2(1) = 19.65 *p* < 0.001, as well as significant interactions between frequency condition and both linear age χ2(1) = 10.59, *p* = 0.001, and quadratic age χ2(1) = 9.15, *p* = 0.002. Thus, while age differences in initial learning related to overall memory performance, they did not account for age differences in the use of environmental regularities to strategically prioritize memory for valuable information.”

In addition, as suggested by the reviewer, we also included the three covariates as control variables in our mediation analysis. When controlling for online frequency learning and explicit frequency report errors, PFC activity continued to mediate the relation between age and memory difference scores. We have now included these results on p. 16 – 17 of the main text:

“Further, when we included quadratic age, WASI scores, online frequency learning accuracy, online frequency learning accuracy on the final repetition of each item, and mean explicit frequency report error magnitudes as control variables in the mediation analysis, PFC activation continued to mediate the relation between linear age and memory difference scores (standardized indirect effect:.56, 95% confidence interval: [.06, 1.35], *p* = 0.023; standardized direct effect; 1.75, 95% confidence interval: [.12,.3.38], *p* = 0.034).”

We also refer to these analyses when we interpret our findings in our discussion. On p. 23, we write:

“In addition, we continued to observe a robust interaction between age and frequency condition on associative memory, even when controlling for age-related change in the accuracy of both online frequency learning and explicit frequency reports. Thus, though we observed age differences in the learning of environmental regularities and in their influence on subsequent associative memory encoding, our developmental memory effects cannot be fully explained by differences in initial learning.”

We thank the reviewer for this constructive suggestion, as we believe these control analyses strengthen our interpretation of age differences in both the learning and use of environmental regularities to prioritize memory.

5) It would be helpful to include (supplementary) figures accompanying the behavioral results from the frequency learning phase.

We have now included a supplementary figure (Appendix 2 – Figure 2) showing how accuracy and reaction times during frequency learning vary across age and item appearance counts.

6) Supplementary Figure S2 – providing an (additional) plot of the estimated age effects would help match the displayed results better to the description in the main text.

Thank you for this suggestion. We now include a new figure in the main text of manuscript (Figure 3 on p. 10) that shows neural activation in the parahippocampal cortex in response to *all* item repetitions, as well as the relation between age and repetition suppression, as suggested.